# CRITICAL POINTS AND CONVERGENCE ANALYSIS OF GENERATIVE DEEP LINEAR NETWORKS TRAINED WITH BURES-WASSERSTEIN LOSS

## ABSTRACT

We consider a deep matrix factorization model of covariance matrices trained with the Bures-Wasserstein distance. While recent works have made important advances in the study of the optimization problem for overparametrized low-rank matrix approximation, much emphasis has been placed on discriminative settings and the square loss. In contrast, our model considers another interesting type of loss and connects with the generative setting. We characterize the critical points and minimizers of the Bures-Wasserstein distance over the space of rank-bounded matrices. For low-rank matrices the Hessian of this loss can blow up, which creates challenges to analyze convergence of optimizaton methods. We establish convergence results for gradient flow using a smooth perturbative version of the loss and convergence results for finite step size gradient descent under certain assumptions on the initial weights.

## 1 INTRODUCTION

We investigate generative deep linear networks and their optimization using the Bures-Wasserstein distance. More precisely, we consider the problem of approximating a target Gaussian distribution with a deep linear neural network generator of Gaussian distributions by minimizing the Bures-Wasserstein distance. This problem is of interest in two important ways. First, it pertains to the optimization of deep linear networks for a type of loss that is qualitatively different from the well-studied and very particular square loss. Second, it can be regarded as a simplified but instructive instance of the parameter optimization problem in generative networks, specifically Wasserstein generative adversarial networks, which are currently not as well understood as discriminative networks.

The optimization landscapes and the properties of parameter optimization procedures for neural networks are among the most puzzling and actively studied topics in theoretical deep learning (see, e.g. Mei et al., 2018; Liu et al., 2022). Deep linear networks, i.e., neural networks having the identity as activation function, serve as a simplified model for such investigations (Baldi & Hornik, 1989; Kawaguchi, 2016; Trager et al., 2020; Kohn et al., 2022; Bah et al., 2021). The study of linear networks has guided the development of several useful notions and intuitions in the theoretical analysis of neural networks, from the absence of bad local minima to the role of parametrization and overparametrization in gradient optimization (Arora et al., 2018; 2019a;b). Many previous works have focused on discriminative or autoregressive settings and have emphasized the square loss. Although the square loss is indeed a popular choice in regression tasks, it interacts in a very special way with the particular geometry of linear networks (Trager et al., 2020). The behavior of linear networks optimized with different losses has also been considered in several works (Laurent & Brecht, 2018; Lu & Kawaguchi, 2017; Trager et al., 2020) but is less well understood.

The Bures-Wasserstein distance was introduced by Bures (1969) to study Hermitian operators in quantum information, particularly density matrices. It induces a metric on the space of positive semi-definite matrices. The Bures-Wasserstein distance corresponds to the 2-Wasserstein distance between two centered Gaussian distributions (Bhatia et al., 2019). Wasserstein distances enjoy several properties, e.g. they remain well defined between disjointly supported measures and have duality formulations that allow for practical implementations (Villani, 2003), that make them good candidates and indeed popular choices of a loss for learning generative models, with a well-known case

being the Wasserstein Generative Adversarial Networks (GANs) (Arjovsky et al., 2017). While the 1-Wasserstein distance has been most commonly used in this context, the Bures-Wasserstein distance has also attracted much interest, e.g. in the works of Muzellec & Cuturi (2018); Chewi et al. (2020); Mallasto et al. (2022), and has also appeared in the context of linear quadratic Wasserstein generative adversarial networks (Feizi et al., 2020). A 2-Wasserstein GAN is a minimum 2-Wasserstein distance estimator expressed in Kantorovich duality (see details in Appendix B). This model can serve as an attractive platform to develop the theory particularly when the inner problem can be solved in closed-form. Such a formula is available when comparing pairs of Gaussian distributions. In the case of centered Gaussians this corresponds precisely to the Bures-Wasserstein distance. Strikingly, even in this simple case, the optimization properties of the corresponding problem are not well understood; which we aim to address in the present work.

## 1.1 CONTRIBUTIONS

We establish a series of results on the optimization of deep linear networks trained with the Bures-Wasserstein loss, which we can summarize as follows.

- We obtain an analogue of the Eckart-Young-Mirsky theorem characterizing the critical points and minimizers of the Bures-Wasserstein distance over matrices of a given rank (Theorem 4.2).

- To circumvent the non-smooth behaviour of the Bures-Wasserstein loss when the matrices drop rank, we introduce a smooth perturbative version (Definition 5 and Lemma 3.3), and characterize its critical points and minimizers over rank-constrained matrices (Theorem 4.4) and link them to the critical points on the parameter space (Proposition 4.5).

- For the smooth Bures-Wasserstein loss, in Theorem 5.6 we show exponential convergence of the gradient flow assuming balanced initial weights (Definition 2.1) and a uniform margin deficiency condition (Definition 5.2).

- For the Bures-Wasserstein loss and its smooth version, in Theorem 5.7 we show convergence of gradient descent provided the step size is small enough and assuming balanced initial weights.

## 1.2 RELATED WORKS

**Low rank matrix approximation**   The function space of a linear network corresponds to $n \times m$ matrices of rank at most $\underline{d}$, the lowest width of the network. Hence optimization in function space is closely related to the problem of approximating a given data matrix by a low-rank matrix. When the approximation error is measured in Frobenius norm, Eckart & Young (1936) characterized the optimal bounded-rank approximation of a given matrix in terms of its singular value decomposition. Mirsky (1960) obtained the same characterization for the more general case of unitary invariant matrix norms, which include the Euclidean operator norm and the Schatten-$p$ norms. There are generalizations to certain weighted norms (Ruben & Zamir, 1979; Dutta & Li, 2017). However, for general norms the problem is known to be difficult (Song et al., 2017; Gillis & Vavasis, 2018; Gillis & Shitov, 2019).

**Loss landscape of deep linear networks**   For the square loss, the optimization landscape of linear networks has been studied in numerous works. The pioneering work of Baldi & Hornik (1989) showed, focusing on the two-layer case, that there is a single minimum (up to a trivial parametrization symmetry) and all other critical points are saddle points. Kawaguchi (2016) obtained corresponding results for deep linear nets and showed the existence of bad saddles (with no negative Hessian eigenvalues) for networks with more than three layers. Chulhee et al. (2018) found sets of parameters such that any critical point in this set is a global minimum and any critical point outside is a saddle. Variations include the study of critical points for different types of architectures, such as deep linear residual networks (Hardt & Ma, 2017) and deep linear convolutional networks (Kohn et al., 2022).

For losses different from the square loss there are also several results. Laurent & Brecht (2018) showed that deep linear nets with no bottlenecks have no local minima that are not global for arbitrary convex differentiable loss. Lu & Kawaguchi (2017) showed that if the loss is such that any local minimizer in parameter space can be perturbed to an equally good minimizer with full-rank factor matrices, then all local minima in parameter space are local minima in function space. Trager

et al. (2020) found that for linear networks with arbitrarily rank-constrained function space, only for the square loss does one have the nonexistence of non-global local minima. However, for arbitrary convex losses, non-global local minima when they exist are always pure, meaning that they correspond to local minima in function space.

**Optimization dynamics of deep linear networks** Saxe et al. (2014) studied the learning dynamics of deep linear networks under different classes of initial conditions. Arora et al. (2019b) obtained a closed-form expression for the parametrization along time in a deep linear network for the square loss. Notably, the authors found that solutions with a lower rank are preferred as the depth of the network increases. Arora et al. (2018) derive several invariances of the flow and compare the dynamics in parameter and function space. For the square loss Arora et al. (2019a) proved linear convergence for linear networks with no bottlenecks, approximately balanced initial weights, and initial loss smaller than for any rank-deficient solution. A detailed analysis of the dynamics in the shallow case with square loss was conducted by Tarmoun et al. (2021); Min et al. (2021) including symmetric factorization. The role on inbalance was remarked in those works. For the deep case, also focusing on the square loss, Bah et al. (2021) showed the gradient flow converges to a critical point and a global minimizer on the manifold of fixed rank matrices of some rank. More recently, Nguegnang et al. (2021) extended this analysis to obtain corresponding results for gradient descent.

**Bures-Wasserstein distance** Chewi et al. (2020) studied the convergence of gradient descent algorithms for the Bures-Wasserstein barycenter, proving linear rates of convergence for the gradient descent. In contrast to our work, they consider a Polyak-Łojasiewicz inequality derived from the optimal transport theory to circumvent the non geodesical convexity of the barycenter. In the same vein, Muzellec & Cuturi (2018) exploit properties of optimal transport theory to optimize the distance between two elliptical distributions. To avoid rank deficiency, they perturbed the diagonal elements of the covariance matrix by a small parameter. Feizi et al. (2020) characterized the optimal solution of a 2-Wasserstein GAN with rank-$k$ linear generator as the $k$-PCA solution. We will obtain an analogue result in our settings, along with a description of critical points.

## 1.3 NOTATIONS

We adopt the following notations throughout the paper. For any $n \in \mathbb{N}$, denote $[n] = \{1, 2, \ldots, n\}$. Let $\mathcal{S}(n)$ be the spaces of real symmetric matrices of size $n$. We denote by $\mathcal{S}_+(n)$ (resp. $\mathcal{S}_{++}(n)$) the space of real symmetric positive semi-definite (resp. definite) matrices of size $n$. Given $k \leqslant n$, the set of rank $k$ positive semi-definite matrices is denoted by $\mathcal{S}_+(k, n)$. We use $\mathcal{M}_k$ (resp. $\mathcal{M}_{\leqslant k}$) to denote the set of matrices of rank exactly $k$ (resp. of rank at most $k$), with the format of the matrix understood from context. The scalar product between two matrices $A, B \in \mathbb{R}^{n \times m}$ is $\langle A, B \rangle = \operatorname{tr} A^\top B$, and its associated (Frobenius) norm is noted $\|\cdot\|_F$. The identity matrix of size $n$ will be denoted $I_n$, or $I$ when $n$ is clear. For a (Fréchet) differentiable function $f \colon X \to Y$, we denote its differential at $x \in X$ in the direction $v$ by $\mathrm{d}f(x)[v]$. Finally, $\operatorname{Crit}(f)$ gives the set of critical points of $f$, i.e. the set of points at which the differential of $f$ is $0$.

## 2 LINEAR NETWORKS AND THEIR GRADIENT DYNAMICS

We consider a linear network with $d_0$ inputs and $N$ layers of widths $d_1, \ldots, d_N$, which is a model of linear functions of the form

$$x \mapsto W_N \cdots W_1 x,$$

parametrized by the weight matrices $W_j \in \mathbb{R}^{d_j \times d_{j-1}}$, $j \in [N]$. We will denote the tuple of weight matrices by $\overrightarrow{W} = (W_1, \ldots, W_N)$ and the space of all such tuples by $\Theta$. This is the *parameter space of our model*. To slightly simplify the notation we will also denote the input and output dimensions by $m = d_0$ and $n = d_N$, respectively, and write $W = W_N \cdots W_1$ for the end-to-end matrix. For any $1 \leqslant i \leqslant j \leqslant N$, we will also write $W_{j:i} = W_j \cdots W_i$ for the matrix product of layer $i$ up to $j$. We note that the represented function is linear in the network input $x$, but the parametrization is not linear in the parameters $\overrightarrow{W}$. We denote the parametrization map by

$$\mu \colon \Theta \to \mathbb{R}^{d_N \times d_0}; \; \overrightarrow{W} = (W_1, \ldots, W_N) \mapsto W_{N:1} = W_N \cdots W_1.$$

The *function space* of the network is the set of linear functions it can represent, corresponds to the set of possible end-to-end matrices, which are the $n \times m$ matrices of rank at most $\underline{d} := \min\{d_0, \ldots, d_N\}$. When $\underline{d} = \min\{d_0, d_N\}$, the function space is a vector space, but otherwise, when there is a bottleneck so that $\underline{d} < d_0, d_N$, it is a non-convex subset of $\mathbb{R}^{m \times n}$ determined by polynomial constraints, namely the vanishing of the $(\underline{d} + 1) \times (\underline{d} + 1)$ minors.

Next we collect a few results on the gradient dynamics of linear networks for general differentiable losses, which have been established in previous works even when in some cases the focus was on the on the square loss (Kawaguchi, 2016; Bah et al., 2021; Chitour et al., 2022; Arora et al., 2018). In the interest of conciseness, here we only provide the main takeaways and defer a more detailed discussion to Appendix C. For the remainder of this section let $L^1 : \mathbb{R}^{n \times m} \to \mathbb{R}$ be *any* differentiable loss and let $L^N$ be defined through the parametrization $\mu$ as $L^N(\overrightarrow{W}) = L^1 \circ \mu(\overrightarrow{W})$. For any such loss, the gradient flow $t \mapsto \overrightarrow{W}(t)$ is defined by

$$\frac{\mathrm{d}\overrightarrow{W}(t)}{\mathrm{d}t} = -\nabla L^N(\overrightarrow{W}(t)) \iff \forall j \in [N], \quad \frac{\mathrm{d}W_j(t)}{\mathrm{d}t} = -\nabla_{W_j} L^N(W_1(t), \ldots, W_N(t)). \quad \text{(GF)}$$

This governs the evolution of the parameters during gradient minimization of the loss. Further, we observe that the gradient of $L^N$ is given by

$$\nabla_{W_j} L^N(W_1, \ldots, W_N) = W_{j+1}^\top \cdots W_N^\top \nabla L^1(W) W_1^\top \cdots W_{j-1}^\top \quad \text{for all } j \in \{1, \ldots, N\}. \quad (1)$$

As it turns our, the gradient flow dynamics preserves the difference of the Gramians of subsequent layer weight matrices, which are thus invariants of the gradient flow,

$$\frac{\mathrm{d}}{\mathrm{d}t}(W_{j+1}(t)^\top W_{j+1}(t)) = \frac{\mathrm{d}}{\mathrm{d}t}(W_j(t) W_j(t)^\top).$$

The important notion of *balancedness* for the weights of linear networks was first introduced by Fukumizu (1998) in the shallow case and generalized to the deep case by Du et al. (2018). This is useful in particular as a way of removing the redundancy of the parametrization when investigating the dynamics in function space and has been considered in numerous works.

**Definition 2.1** (Balanced weights). The weights $W_1, \ldots, W_N$ are said to be *balanced* if, for all $j \in [N-1]$, $W_j W_j^\top = W_{j+1}^\top W_{j+1}$.

Assuming balanced initial weights, if the flow of each $W_j$ is defined and bounded, then the rank of the end-to-end matrix $W$ remains constant during training (Bah et al., 2021, Proposition 4.4). Moreover, the products $W_{N:1} W_{N:1}^\top$ and $W_{N:1}^\top W_{N:1}$ can be written in a concise manner; namely, we have $W_{N:1} W_{N:1}^\top = (W_N W_N^\top)^N$ and $W_{N:1}^\top W_{N:1} = (W_1^\top W_1)^N$, which simplifies many computations.

**Remark 2.2.** In order to relax the balanced initial weights assumption, some works also consider approximate balancedness (Arora et al., 2019a), which requires only that there exists $\delta > 0$ such that $\|W_j W_j^\top - W_{j+1}^\top W_{j+1}\|_F \leq \delta$ for $j \in [N-1]$. We will use exactly balanced initialization in our proofs, but they would also go through by invoking approximate balancedness. Another alternative initialization has been proposed by Yun et al. (2021). We defer such extensions to future work favoring here the discussion of the Bures-Wasserstein loss.

## 3 WASSERSTEIN GENERATIVE LINEAR NETWORK

### 3.1 THE BURES-WASSERSTEIN LOSS

The Bures-Wasserstein (BW) distance is defined on the space of positive semi-definite matrices (or covariance space) $\mathcal{S}_+(n)$. Here we collect some of the key properties and discuss the gradient, and refer the reader to Bhatia et al. (2019) for further details on this interesting distance.

**Definition 3.1** (Bures-Wasserstein distance). Given two symmetric positive semidefinite matrices $\Sigma_0, \Sigma \in \mathcal{S}_+(n)$, the squared Bures-Wasserstein distance between $\Sigma_0$ and $\Sigma$ is defined as

$$\mathcal{B}^2(\Sigma, \Sigma_0) = \mathrm{tr}\left(\Sigma + \Sigma_0 - 2(\Sigma_0^{1/2} \Sigma \Sigma_0^{1/2})^{1/2}\right). \quad (2)$$

(Kroshnin et al., 2021, Lemma A.3) shows that the square root is differentiable on the set of positive definite matrices and as a consequence we can differentiate the BW distance at $\Sigma_0, \Sigma \in \mathcal{S}_{++}(n)$.

However, the mapping $\Sigma \mapsto \mathcal{B}^2(\Sigma, \Sigma_0)$ is not differentiable over all of $\mathbb{R}^{n \times n}$. Indeed, letting $\Gamma Q \Gamma^\top$ be a spectral decomposition of $\Sigma_0^{1/2} \Sigma \Sigma_0^{1/2}$, (2) can be written as

$$\mathcal{B}^2(\Sigma, \Sigma_0) = \|\Sigma^{1/2}\|_F^2 + \|\Sigma_0^{1/2}\|_F^2 - 2 \operatorname{tr} Q^{1/2}. \tag{3}$$

Due to the square root on $Q$, the map $\Sigma \mapsto \mathcal{B}^2(\Sigma, \Sigma_0)$ is not differentiable when the rank of $\Sigma_0^{1/2} \Sigma \Sigma_0^{1/2}$, i.e. the number of positive eigenvalues of $Q$, changes. More specifically, while one can compute the gradient over the set of matrices of rank $k$ for any given $k$, the norm of the gradient blows up if the matrix changes rank. The gradient of $\mathcal{B}^2$ restricted to the set of full-rank matrices is given in Appendix B.

## 3.2 Linear Wasserstein GAN

The distance defined in (2) corresponds to the 2-Wasserstein distance between two zero-centered Gaussians and can be used as a loss for training models of Gaussian distributions and in particular generative linear networks. Recall that zero-centered Gaussian distributions are completely specified by their covariances. Given a bias-free linear network and a latent Gaussian probability measure $\mathcal{N}(0, I_m)$, a linear network generator computes a push-forward of the latent distribution, which is again a Gaussian distribution. If $Z \sim \mathcal{N}(0, I_m)$ and $X = WZ$, then

$$X \sim \mathcal{N}(0, WW^\top) =: \nu,$$

Given a target distribution $\nu_0 = \mathcal{N}(0, \Sigma_0)$ or simply the covariance matrix $\Sigma_0$ (which may be a sample covariance matrix), one can select $W$ by minimizing $\mathcal{B}^2(WW^\top, \Sigma_0) = \mathcal{W}_2^2(\nu, \nu_0)$ so that the network approximates the distribution $\mathcal{N}(0, \Sigma_0)$. We will denote the map that takes the end-to-end matrix $W$ to the covariance matrix $WW^\top$ by $\pi \colon \mathbb{R}^{n \times m} \to \mathbb{R}^{n \times n}$; $W \mapsto WW^\top$.

**Loss in covariance, function, and parameter spaces**   We consider the following losses, which differ only on the choice of the search variable, taking a function space or a parameter space view.

- First, we denote the loss over covariance matrices $\Sigma \in \mathcal{S}_+(n)$ as $\tilde{L} \colon \Sigma \mapsto \mathcal{B}^2(\Sigma, \Sigma_0)$ .
- Secondly, given $\pi \colon W \mapsto WW^\top \in \mathcal{S}_+(n)$ , we define the loss in function space, i.e., over end-to-end matrices $W \in \mathbb{R}^{n \times m}$ as $L^1 \colon W \mapsto \tilde{L} \circ \pi(W)$. This is given by

$$\forall W \in \mathbb{R}^{n \times m}, \quad L^1(W) = \operatorname{tr}\left(WW^\top + \Sigma_0 - 2(\Sigma_0^{1/2} WW^\top \Sigma_0^{1/2})^{1/2}\right). \tag{4}$$

  This loss is not convex on $\mathbb{R}^{n \times m}$, as can be seen in the one-dimensional case.

- Lastly, for a tuple of weight matrices $\overrightarrow{W} = (W_1, \ldots, W_N)$, we compose $L^1$ with the parametrization map $\mu \colon \overrightarrow{W} \mapsto W_{N:1}$, to define the loss in parameter space as $L^N \colon \overrightarrow{W} \mapsto \tilde{L} \circ \pi \circ \mu(\overrightarrow{W})$, for $\overrightarrow{W} \in \Theta$. Observe that this is, again, a non-convex loss.

Thus, for $\overrightarrow{W} \in \Theta$, $L^N(\overrightarrow{W}) = L^1(\mu(\overrightarrow{W})) = \tilde{L}(\pi \circ \mu(\overrightarrow{W})) = \mathcal{B}^2(\pi \circ \mu(\overrightarrow{W}), \Sigma_0)$. While the gradient flow (GF) is defined on the parameters $\overrightarrow{W}$, the covariance space perspective is useful since it leads to a convex objective function, even if this may be subject to non-convex constraints. One of our goals will be to translate properties between $\tilde{L}$, $L^1$, and $L^N$.

**Smooth perturbative loss**   As mentioned before, the Bures-Wasserstein loss is non-smooth at matrices with vanishing singular values. In turn, the usual analysis tools to prove uniqueness and convergence of the gradient flow do not apply for this loss. Therefore, we introduce a smooth perturbative version. Consider the perturbation map $\varphi_\tau \colon \Sigma \mapsto \Sigma + \tau I_n$, where $\tau > 0$ plays the role of a regularization strength. Then the perturbative loss on the covariance space can be expressed as $\tilde{L}_\tau = \tilde{L} \circ \varphi_\tau$, and the perturbative loss on function space as $L_\tau^1 = \tilde{L}_\tau \circ \pi$. More explicitly, we let

$$L_\tau^1(W) = \operatorname{tr}\left(WW^\top + \tau I_n + \Sigma_0 - 2(\Sigma_0^{1/2}(WW^\top + \tau I_n)\Sigma_0^{1/2})^{1/2}\right). \tag{5}$$

This function is smooth enough to apply usual convergence arguments for the gradient flow. Likewise, $L_\tau^N := \tilde{L}_\tau \circ \pi \circ \mu$ is well-defined and smooth on $\Theta$.

**Remark 3.2.** The perturbative loss (5) is differentiable. Many results from Bah et al. (2021) can be carried over for the differentiable Bures-Wasserstein loss. For example, the uniform boundedness of the end-to-end matrix holds, $\|W(t)\| \leqslant \sqrt{2L^1(W(0)) + \operatorname{tr}\Sigma_0}$. Similar observations may apply for the case of $L^1$ in the case that the matrix $WW^\top$ remains positive definite throughout training, in which case the gradient flow remains well defined and the loss is monotonically decreasing. We expand on this in Appendix C.

The next lemma, proved in Appendix B.4, provides a quantitative bound for the difference between the original and the perturbative loss. For this lemma, we use the fact that the rank is constant. To compare the two losses, we fix the parameters to be the same. Recall that $\Sigma_\tau = WW^\top + \tau I$.

**Lemma 3.3.** *Let $W \in \mathbb{R}^{n \times m}$, $\tau > 0$, and let $\Sigma_\tau = WW^\top + \tau I_n$. Assume that $\operatorname{rank}(WW^\top) = r$, $\operatorname{rank}\Sigma_0 = n$, $L^1(W)$ is given by (4), and $\tilde{L}(\Sigma_\tau)$ is given by (5). Then*

$$|\tilde{L}(\Sigma_\tau) - L^1(W)| \leq \tau n + r\tau^{1/2}. \tag{6}$$

We observe that the upper bound (6) is tight, since it goes to zero as $\tau$ goes to zero.

## 4 CRITICAL POINTS

In this section, we characterize the critical points of the Bures-Wasserstein loss restricted to matrices of a given rank. The proofs of results in this section are given in Appendix D.

For $k \in \mathbb{N}$, denote by $\mathcal{M}_k$ the manifold of rank-$k$ matrices, i.e.

$$\mathcal{M}_k = \{W \in \mathbb{R}^{n \times m} \mid \operatorname{rank} W = k\}.$$

Similarly, denote $\mathcal{M}_{\leqslant k}$ the set of matrices of rank at most $k$. The format $n \times m$ of the matrices is to be inferred from the context. The manifold $\mathcal{M}_k$ is viewd as an embedded submanifold of the linear space $(\mathbb{R}^{n \times m}, \langle \cdot, \cdot \rangle_F)$, with codimension $(n - k)(m - k)$ (Boumal 2022, §2.6; Uschmajew & Vandereycken 2020, §9.2.2). Given a function $f \colon \mathbb{R}^{n \times m} \to \mathbb{R}$, its *restriction* on $\mathcal{M}_k$ is denoted by $f|_{\mathcal{M}_k} \colon \mathcal{M}_k \ni W \mapsto f(W)$. Even if a function $f$ is not differentiable over all of $\mathbb{R}^{n \times m}$, its restriction on $\mathcal{M}_k$ may be differentiable.

**Definition 4.1.** Let $\mathcal{M}$ be a smooth manifold. Let $f \colon \mathbb{R}^{n \times m} \to \mathbb{R}$ be any function such that its restriction on $\mathcal{M}$ is differentiable. A point $W \in \mathcal{M}$ is said to be a critical point for $f|_{\mathcal{M}}$ if the differential of $f|_{\mathcal{M}}$ at $W$ is the zero function, $\mathrm{d}f|_{\mathcal{M}}(W) = 0$.

### 4.1 CRITICAL POINTS OF $L^1$ OVER $\mathcal{M}_k$

Given a matrix $A \in \mathbb{R}^{n \times n}$ and a set $\mathcal{J}_k \subseteq [n]$ of $k$ indices, we denote by $A_{\mathcal{J}_k} \in \mathbb{R}^{n \times k}$ the submatrix of $A$ consisting of the columns with index in $\mathcal{J}_k$. If the matrix $A$ is diagonal, we denote by $\bar{A}_{\mathcal{J}_k} \in \mathbb{R}^{k \times k}$ the diagonal submatrix which extracts the rows and columns with index in $\mathcal{J}_k$. The following result characterizes the critical points of the loss in function space. It can be regarded as a type of Eckart-Young-Mirsky result for the case of the Bures-Wasserstein loss.

**Theorem 4.2** (Critical points of $L^1$). *Assume $\Sigma_0$ has $n$ distinct, positive eigenvalues. Let $\Sigma_0 = \Omega\Lambda\Omega^\top$ be a spectral decomposition of $\Sigma_0$ (so $\Omega \in U(n)$). Let $k \in [\min\{n, m\}]$. A matrix $W^* \in \mathcal{M}_k$ is a critical point of $L^1|_{\mathcal{M}_k}$ if and only if $W^* = \Omega_{\mathcal{J}_k}\bar{\Lambda}_{\mathcal{J}_k}^{1/2}V^\top$ for some $\mathcal{J}_k \subseteq [n]$ and $V \in \mathbb{R}^{m \times k}$ with $V^\top V = I_k$. In particular, the number of critical points is $\binom{n}{k}$. The minimum is attained for $\mathcal{J}_k = [k]$. In particular, $\inf_{\mathcal{M}_k} L^1(W) = \min_{\mathcal{M}_k} L^1(W)$ and $\min_{\mathcal{M}_k} L^1(W) = \min_{\mathcal{M}_{\leqslant k}} L^1(W)$.*

The proof relies on an expression of the gradient $\nabla L^1|_{\mathcal{M}_k}$ (see Lemma D.3) and evaluating its zeros. The value of the loss evaluated at these critical points allows to rank them and identify those that are minimal.

**Remark 4.3.** Interestingly, the critical points and the minimizer of $L^1$ characterized in the above result agree with those of the square loss (Eckart & Young, 1936; Mirsky, 1960). Nonetheless, we observe that (2) is only defined for positive semidefinite matrices. Hence the notion of unitary invariance considered by Mirsky (1960) here only makes sense for left and right multiplication by

the same matrix. Moreover, while we can establish unitary invariance for a variational extension of the distance, this still is not a norm in the sense that there is no function $B \colon \mathbb{R}^{n \times n} \to \mathbb{R}$ such that $\mathcal{B}(\Sigma, \Sigma_0) = B(\Sigma - \Sigma_0)$, and hence it does not fall into the framework of Mirsky (1960). We offer more details about this in Appendix B.

## 4.2 Critical points of the perturbative loss

For the critical points of the perturbative loss $L_\tau^1(W)$ we obtain the following results.

**Theorem 4.4** (Critical points of $L_\tau^1$). *Let $\Sigma_0 = \Omega \Lambda \Omega^\top$ be a spectral decomposition of $\Sigma_0$. A point $W^* \in \mathcal{M}_k$ is a critical point for $L_\tau^1$ if and only if there exists a subset $\mathcal{J}_k \subseteq [n]$ and a semi-orthogonal matrix $V \in \mathbb{R}^{n \times k}$ (i.e., so that $V^\top V = I$) such that $W^* = \Omega_{\mathcal{J}_k} (\bar{\Lambda}_{\mathcal{J}_k} - \tau I_k)^{1/2} V^\top$. The (unique) minimum over $\mathcal{M}_{\leqslant k}$ is attained for $\mathcal{J}_k = [k]$*

Note that the above characterization of the critical points imposes an upper bound on $\tau$. In other words, for a given $W^*$ to be a critical point, one must have that $\tau < \lambda_j$ for all $j \in \mathcal{J}_k$, because the eigenvalues of $\bar{\Lambda}_{\mathcal{J}_k} - \tau I_k$ are positive.

In order to link the critical points in parameter space to the critical points in the function space, we appeal to the correspondence drawn in Trager et al. (2020, Propositions 6 and 7). For the Bures-Wasserstein loss, this allows to conclude the following.

**Proposition 4.5.** *Assume a full-rank target $\Sigma_0$, with distinct, decreasing eigenvalues, and spectral decomposition $\Sigma_0 = \Omega \Lambda \Omega^\top$. Let $\tau \in (0, \lambda_n]$. If $\overrightarrow{W}^* \in \mathrm{Crit}(L_\tau^N)$, then $\Sigma^* = \pi(\mu(\overrightarrow{W}^*))$ is a critical point of the loss $\tilde{L}_\tau|_{\mathcal{S}_+(k,n)}$, where $k = \mathrm{rank}\,\Sigma^*$. Moreover, if $k = \underline{d}$, then $\overrightarrow{W}$ is a local minimizer for the loss $L_\tau^N$ if and only if $\Sigma^* = \pi(\mu(\overrightarrow{W}^*))$ is a local minimizer, and therefore the global minimizer, of the loss $\tilde{L}_\tau|_{\mathcal{S}_+(\underline{d},n)}$. In this case, $\Sigma_\tau^* = \Sigma^* + \tau I_n$ is the $\tau$-best $\underline{d}$-rank approximation of the target in the covariance space, in the sense that $\Sigma_\tau^* = \Omega \begin{pmatrix} \Lambda_{[\underline{d}]} & \\ & \tau \end{pmatrix} \Omega^\top$.*

Proposition 4.5 ensures that, under the assumption that the solution of the gradient flow is a (local) minimizer and has the highest possible rank $\underline{d}$ given the network architecture, the solution in the covariance space is the best $\underline{d}$-rank approximation of the target in the sense of the $\tau$-smoothed Bures-Wasserstein distance.

**Remark 4.6.** Under the balancedness assumption, one can show that the rank of the end-to-end matrix does not drop during training (Bah et al., 2021, Proposition 4.4), and that one escape almost surely the strict saddle points (Bah et al., 2021, Theorem 6.3). If the initialization of the network has rank $\underline{d}$, the matrices $W(t)$, $t > 0$, maintain rank $\underline{d}$ throughout training. There can be issues in the limit, since $\mathcal{M}_d$ is not closed. Proving that the limit point also belongs to $\mathcal{M}_d$ is an interesting open problem that we leave for future work.

## 5 Convergence analysis

The Bures-Wasserstein distance can be viewed through the lens of the Procrustes metric (Dryden et al., 2009; Masarotto et al., 2019). In fact, it can be obtained by the following minimization problem.

**Lemma 5.1** (Bhatia et al. 2019, Theorem 1). *For $\Sigma$, $\Sigma_0 \in \mathcal{S}_+(n)$,*

$$\mathcal{B}^2(\Sigma, \Sigma_0) = \min_{U \in U(n)} \|\Sigma^{1/2} - \Sigma_0^{1/2} U\|_F^2, \tag{7}$$

*where $U(n)$ denotes the $n \times n$ orthogonal matrix group. Moreover, the minimizer $\bar{U}$ occurs in the polar decomposition of $\Sigma^{1/2} \Sigma_0^{1/2}$.*

We emphasize that in the above description of the Bures-Wasserstein distance, the minimizer $\bar{U}$ depends on $W$, so that $\mathcal{B}^2$ fundamentally differs from a squared Frobenius norm. Moreover, the square root on $\Sigma_\theta$ can lead to singularities when differentiating the loss. Nonetheless, the expression (7) can be used to avoid such singularities, by leveraging the following deficiency margin concept.

**Definition 5.2** (Modified deficiency margin). Given a target matrix $\Sigma_0 \in \mathbb{R}^{n \times n}$ and a positive constant $c > 0$, we say that $\Sigma_\theta \in \mathbb{R}^{n \times n}$ has a modified deficiency margin $c$ with respect to $\Sigma_0$ if

$$\min_{U \in U(n)} \|\Sigma_\theta^{1/2} - \Sigma_0^{1/2} U\|_F \leq \sigma_{\min}(\Sigma_0^{1/2}) - c. \tag{8}$$

With a slight abuse of denomination, we will say that $W$ has a uniform deficiency margin if $WW^\top$ has one. This deficiency margin idea can be traced back to Arora et al. (2019a). Note that we can write $\sqrt{WW^\top} = \Sigma_\theta^{1/2}$, and this square root can be realized by Cholesky decomposition. Notice that if we initialize close to the target then the above bound (8) holds trivially. In fact, if the initial condition $W(0)$ satisfies the uniform deficient margin, then we have that the least singular value of $W(k)$ remains bounded below by $c$ for all times $k \geq 0$, for the gradient flow or gradient descent with decreasing $L^N$:

**Lemma 5.3.** *Suppose $W(0)W(0)^\top$ has a modified deficiency margin $c$ with respect to $\Sigma_0$. Then*

$$\sigma_{\min}\left(\sqrt{W(k)W(k)^\top}\right) \geq c, \quad \text{for } k \geq 0. \tag{9}$$

The proof of this and all results in this section are provided in Appendix E. We note that, while the modified margin deficiency assumption is sufficient for Lemma 5.3 to hold, it is by no means necessary. We will assume that the modified margin deficiency assumption holds for the simplicity of exposition, but the gradient flow analysis in the next paragraph only requires the less restrictive Lemma 5.3 to hold.

**Convergence of the gradient flow for the smooth loss**   Because we cannot exclude the possibility that the rank of $WW^\top$ drops along the gradient flow of the BW loss, we consider the smooth perturbation as a way to avoid singularities. We consider the gradient flow (GF) for the perturbative loss (5). The gradient of (5) is given by

$$\nabla L_\tau^1(W) = 2\Big(W - \Sigma_0^{1/2}\big(\Sigma_0^{1/2}(WW^\top + \tau I_n)\Sigma_0^{1/2}\big)^{-1/2}\Sigma_0^{1/2}W\Big).$$

On the other hand, we may denote $\Sigma_\tau := WW^\top + \tau I_n$ as a regularized covariance matrix, and express the $L^1$ loss in terms of the optimal transport plan between $\Sigma_\tau$ and $\Sigma_0$ (Kroshnin et al., 2021). We have

$$\begin{aligned}
\tilde{L}(\Sigma_\tau) &= \operatorname{tr}\left(\Sigma_\tau + \Sigma_0 - 2(\Sigma_0^{1/2}\Sigma_\tau\Sigma_0^{1/2})^{1/2}\right) = \big\|\big(T_{\Sigma_\tau}^{\Sigma_0} - I\big)\Sigma_\tau\big\|_F^2 \\
&= \operatorname{tr}\big(T_{\Sigma_\tau}^{\Sigma_0} - I\big)\Sigma_\tau \operatorname{tr}\big(T_{\Sigma_\tau}^{\Sigma_0} - I\big),
\end{aligned} \tag{10}$$

where $T_{\Sigma_\tau}^{\Sigma_0} = \Sigma_0^{1/2}\big(\Sigma_0^{1/2}\Sigma_\tau\Sigma_0^{1/2}\big)^{-1/2}\Sigma_0^{1/2} = \Sigma_\tau^{-1/2}\big(\Sigma_\tau^{1/2}\Sigma_0\Sigma_\tau^{1/2}\big)^{1/2}\Sigma_\tau^{-1/2}$.

The perturbation $\tau I_n$ ensures strict convexity as shown in the following result.

**Lemma 5.4.** *The function $\Sigma_\tau \mapsto \tilde{L}(\Sigma_\tau)$ is strictly convex on $\mathcal{S}_{++}(n)$.*

*Proof.* First we observe that the function $f(X) = \operatorname{tr} X^{1/2}$ is strictly concave on $\mathcal{S}_{++}(n)$; for details we refer the reader to Bhatia et al. (2019, Theorem 7). As a result, the function

$$\Sigma_\tau \mapsto \tilde{L}(\Sigma_\tau) = \operatorname{tr}\Sigma_0 + \operatorname{tr}\Sigma_\tau - 2\operatorname{tr}\big(\Sigma_0^{1/2}\Sigma_\tau\Sigma_0^{1/2}\big)^{1/2}$$

is convex on $\mathcal{S}_{++}(n)$. Then $\Sigma_\tau \mapsto \operatorname{tr}\big(\Sigma_0^{1/2}\Sigma_\tau\Sigma_0^{1/2}\big)^{1/2}$ is an injective linear map since $\Sigma_\tau$ is positive definite matrix. This means that $\tilde{L}$ is strictly convex. $\square$

What's more, the loss $\tilde{L}$ is strongly-convex on $\mathcal{S}_{++}(n)$, as stated in the next lemma.

**Lemma 5.5.** *The Hessian of the loss $\tilde{L}$ satisfies $\nabla_{\Sigma_\tau}^2 \tilde{L}(\Sigma_\tau) \succcurlyeq K \cdot I_n$ for any $\Sigma_\tau \in \mathcal{S}_{++}(n)$, with $K = \frac{\sqrt{\tau \lambda_{\min}(\Sigma_0)}}{2C_0^2}$, where $C_0 = 2(\tilde{L}_\tau(\Sigma(0)) + \operatorname{tr}\Sigma_0)$.*

Let us denote the minimizer of the perturbative loss $\tilde{L}(\Sigma_\tau)$ by $\Sigma_\tau^*$. Let $\Delta_0^* = \Sigma_\tau(0) - \Sigma_\tau^*$ be the distance of the initialization from the optimal solution. Equipped with the strict convexity by Lemma 5.4, we are ready to show that the gradient flow has convergence rate $O(e^{-\tilde{K}_{c,N}Kt})$, where $K$ is the constant from the Hessian bound given by Lemma E.5, and $\tilde{K}_{c,N}$ is a constant which depends on the modified margin deficiency and the depth of the linear neural network. Recall that $\Sigma_\tau = W_{N:1}W_{N:1}^\top + \tau I_n$, so we prove convergence of gradient flow for the parametrization $\overrightarrow{W}$.

**Theorem 5.6** (Convergence of gradient flow). *Assume both balancedness (Definition 2.1) and the modified uniform deficiency margin conditions (Definition 5.2). Then the gradient flow* (GF) *converges as*

$$\tilde{L}(\Sigma_\tau(t)) - \tilde{L}(\Sigma_\tau^*) \leq e^{-8Nc^{\frac{2(2N-1)}{N}}Kt}\Delta_0^*, \tag{11}$$

*where* $K = \frac{\sqrt{\tau\lambda_{\min}(\Sigma_0)}}{2C_0^2}$ *is the strong convexity parameter from Lemma 5.5, with* $C_0 = 2(\tilde{L}(\Sigma_\tau(0)) + \mathrm{tr}(\Sigma_0))$.

**Convergence of gradient descent for the BW loss**    Assuming that the initial end-to-end matrix $W$ have a uniform deficiency margin, we can establish the following convergence result for gradient descent with finite step sizes, which is valid for the perturbed loss and also for the non-perturbed original loss. Given an initial value $\overrightarrow{W}(0)$, we consider the gradient descent iteration

$$\overrightarrow{W}(k+1) = \overrightarrow{W}(k) - \eta\nabla L^N(\overrightarrow{W}(k)), \quad k = 0, 1, \ldots, \tag{GD}$$

where $\eta > 0$ is the learning rate or step size and the gradient is given by (1).

**Theorem 5.7** (Convergence of gradient descent). *Assume that the initial values* $W_i(0)$, $1 \leq i \leq N$, *are balanced and are so that* $W(0) = W_{N:1}(0)$ *has uniform deficiency margin* $c$. *If the learning rate* $\eta > 0$ *satisfies*

$$\eta \leq \min\left\{\frac{c^2}{8M\sqrt{L^1(W(0))}}, \frac{Nc^{\frac{2(N-1)}{N}}}{2\Delta}, \frac{1}{4Nc^{\frac{2(N-1)}{N}}}\right\},$$

*where* $\Delta = \frac{2^{N+1}}{c^{2N}}N^2M^{(4N-3)/N}\lambda_{\max}^{1/2}(\Sigma_0) + 8N(N-1)M^{(3N-4)/N}\left(M^{1/N} + \|\Sigma_0^{1/2}\|_F\right)$, *and* $M = \sqrt{2\left(L^1(W(0)) + \|\Sigma_0^{1/2}\|_F^2\right)}$, *then, for any* $\epsilon > 0$, *one can achieve* $\epsilon$ *loss by the gradient descent* (GD) *at iteration*

$$k \geq \frac{1}{2\eta Nc^{\frac{2(N-1)}{N}}}\log\left(\frac{L^1(W(0))}{\epsilon}\right).$$

**Remark 5.8.** Our Theorems 5.6 and 5.7 show that the depth of the network can accelerate the convergence of the gradient algorithms.

## 6    CONCLUSION

In this work, we studied the training of generative linear neural networks using the Bures-Wasserstein distance. We characterized the critical points and minimizers of this loss in function space or over the set of matrices of fixed rank $k$. We introduced a smooth approximation of the BW loss obtained by regularizing the covariance matrix and also characterized its critical points in function space. Furthermore, under the assumption of balanced initial weights satisfying a uniform deficiency margin condition, we established a convergence guarantee to the global minimizer for the gradient flow of the perturbative loss with exponential rate of convergence. Finally, we also considered the finite-step size gradient descent optimization and established a linear convergence result for both the original and the perturbed loss, provided the step size is small enough depending on the uniform margin deficiency condition. We collect our results in Table 1. These results contribute to the ongoing efforts to better characterize the optimization problems that arise in learning with deep neural networks beyond the commonly considered discriminative settings with the square loss.

In future work, it would be interesting to refine our characterization of critical points of the Bures-Wasserstein loss in parameter space. Moreover, the uniform margin deficiency condition that we invoked in order to establish our convergence results constrains the parametrization to be of full rank. Relaxing this assumption is an interesting endeavor to pursue.

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

| Loss | Parametrization | Critical points | Initialization | Convergence |
|------|-----------------|-----------------|----------------|-------------|
| $L^1$ | $W_{N:1}$ | $\Omega_{\mathcal{J}_k}\bar{\Lambda}_{\mathcal{J}_k}^{1/2}V^\top$ | — | — |
| $L^1_\tau$ | $W_{N:1}$ | $\Omega_{\mathcal{J}_k}(\bar{\Lambda}_{\mathcal{J}_k}-\tau I_k)^{1/2}V^\top$ | — | — |
| $\tilde{L}_\tau$ | $\Sigma_\tau$ | $\Sigma_0$ | Balanced, MDM | GF: Exponential |
| $L^1$ | $W_{N:1}$ | $\Omega\Lambda V^\top$ | Balanced, MDM | GD: $O(\log(1/\epsilon))$ |

Table 1: Summary of the results. The target is assumed full rank, with distinct eigenvalues, and spectral decomposition $\Sigma_0 = \Omega\Lambda\Omega^\top$. The end-to-end matrix is $W_{N:1}$, and the regularized covariance is $\Sigma_\tau$. $V \in \mathbb{R}^{m \times k}$ is any semi-orthogonal matrix, and $\mathcal{J}_k \subset [n]$ is an index set. Balanced stands for balanced weights (Definition 2.1), MDM stands for modified deficiency margin (Definition 5.2).

## APPENDIX

The appendix is organized as follows.

- Appendix A presents a table summarizing the different geometrical and convergence results.
- Appendix B discusses the background on the Bures-Wasserstein Loss and related Optimal Transport topics.
- Appendix C presents some general properties of a linear neural network and classical results on convergence in the parameter space.
- Appendix D gathers the proofs that were omitted in Section 4.
- Finally, Appendix E presents the proofs from Section 5.

## A    SUMMARY OF THE RESULTS

Table 1 presents a summary of the different results obtained in this paper. Note that, even if the different losses are expressed in the function or the covariance spaces, the gradient flow and gradient descent algorithm are performed on the parameter space $\Theta$.

## B    BACKGROUND ON THE BURES-WASSERSTEIN DISTANCE

### B.1    DEFINITION OF $\mathcal{W}_2^2$

The Bures-Wasserstein distance has a natural connection with the 2-Wasserstein distance on a metric space. In the case of zero-centered Gaussian measures, those distances are identical. We present here a more general definition of the 2-Wasserstein distance, which enjoys desirable properties.

Given a metric space $(\mathcal{X}, \|\cdot\|)$, the 2-Wasserstein distance is a well-established metric on the space of quadratically integrable probability measures $\mathcal{P}_2(\mathcal{X})$.

**Definition B.1** (2-Wasserstein distance). Given two quadratically integrable measures $(\nu, \nu_0) \in (\mathcal{P}_2(\mathcal{X}))^2$ the 2-Wasserstein distance is defined as the following minimization problem

$$\mathcal{W}_2^2(\nu, \nu_0) = \inf_{\pi \in \Pi(\nu, \nu_0)} \int \|x - y\|^2 \, \mathrm{d}\pi(x, y), \tag{12}$$

where $\Pi(\nu, \nu_0)$ is the set of fixed marginals distributions: $\Pi(\nu, \nu_0) = \{\pi \in \mathcal{P}_2(\mathcal{X} \times \mathcal{X}) \mid \pi_1 = \nu, \pi_2 = \nu_0\}$, with $\pi_i$ the marginal of $\pi$ along the $i$th variable.

It is known that this distance metrizes the weak convergence on the space $\mathcal{P}_2$, see e.g. (Villani, 2008, Theorem 6.9), and can therefore be leveraged when designing a system that relies on comparing probability distributions such as a GAN. On the other hand, the computational burden of such a loss can quickly become prohibitive (Pele & Werman, 2009). In a very few cases, efficient computations can be done for the loss (12). This constrasts with a usual WGAN, which was first introduced by Arjovsky et al. (2017), where the loss is computed using a neural network, based on the dual expression of the (1-)Wasserstein distance.

Indeed, between two Gaussian measures, the 2-Wasserstein distance has a closed-form expression or a closed-form expression for the discriminator so that adversarial training is not needed. We will consider two centered Gaussian distributions, which are described by their covariance matrices. In the case of centered Gaussian distributions, the 2-Wasserstein distance reduces to the Bures-Wasserstein distance between the covariance matrices $\Sigma_0$ and $\Sigma$ (Dowson & Landau, 1982):

**Lemma B.2.** *If $\nu = \mathcal{N}(\boldsymbol{m}, \Sigma)$ and $\nu_0 = \mathcal{N}(\boldsymbol{m}_0, \Sigma_0)$, then*

$$\mathcal{W}_2^2(\nu, \nu_0) = \|\boldsymbol{m} - \boldsymbol{m}_0\|^2 + \mathcal{B}^2(\Sigma, \Sigma_0)$$

It is well known (see Kantorovitch (1958) or (Villani, 2003, Theorem 1.3) or (Villani, 2008, Theorem 5.10)) that the squared 2-Wasserstein distance has the following dual expression, also known as the Kantorovich duality:

$$\mathcal{W}_2^2(\nu_0, \nu_\theta) = \sup_{(f,g) \in \mathcal{L}^1(\nu_\theta) \times \mathcal{L}^1(\nu_0)} \left\{ \int f(x) \, \mathrm{d}\nu_\theta(x) + \int g(x) \, \mathrm{d}\nu_0(x) \right. \tag{13}$$
$$\left. \mid \forall (x, y), \ f(x) + g(y) \leqslant \|x - y\|^2 \right\},$$

where $\mathcal{L}^1(\nu)$ is the set of the integrable functions with respect to a measure $\nu$. Therefore, the dual variables $f$ and $g$ are required to be integrable with respect to the source and target measures, and to fulfil the cost inequality.

**Remark B.3.** In the context of GANs it is common to consider the 1-Wasserstein distance with cost given by the distance $\|x - y\|$, which has a dual expression, referred to as the Kantorovich-Rubinstein formula (Villani, 2008, §6.2) that allows for a more tractable computation in practice, with for instance only one dual variable. Nonetheless, there is no closed-form solution known when $c(x, y) = \|x - y\|$.

### B.2 FURTHER PROPERTIES OF THE BW LOSS

In this section, we provide further background on the Bures-Wasserstein distance. First, we show that, except in some particular cases (Lemma B.4), the Bures-Wasserstein distance between two covariance matrices is not translation invariant (Lemma B.5).

**Lemma B.4.** *In the case that $\Sigma_0$ and $\Sigma$ commute, the Bures-Wasserstein distance reduces to the Hellinger distance:*

$$\Sigma_0 \Sigma = \Sigma \Sigma_0 \quad \implies \quad \mathcal{B}^2(\Sigma, \Sigma_0) = \|\Sigma^{1/2} - \Sigma_0^{1/2}\|_F^2.$$

*Proof.* It simply follows from the fact that, if $\Sigma$ and $\Sigma_0$ commute, so do $\Sigma^{1/2}$ and $\Sigma_0^{1/2}$, so that $\Sigma_0^{1/2} \Sigma \Sigma_0^{1/2} = (\Sigma_0^{1/2} \Sigma^{1/2})^2$ and

$$\mathrm{tr}\,((\Sigma^{1/2})^2 + (\Sigma_0^{1/2})^2 - 2(\Sigma_0^{1/2} \Sigma^{1/2})) = \mathrm{tr}\,((\Sigma^{1/2} - \Sigma_0^{1/2})(\Sigma^{1/2} - \Sigma_0^{1/2})^\top)$$
$$= \|\Sigma^{1/2} - \Sigma_0^{1/2}\|_F^2$$

as claimed. $\qquad\square$

From this, one remarks that the problem of minimizing the BW distance between two covariance matrices that commute does fall under the framework of the Eckart-Young-Mirsky theorem if the optimization variable is $\Sigma^{1/2} = (WW^\top)^{1/2}$, as it reduces to a problem cast with the square loss. Nonetheless, in the case where $\Sigma$ and $\Sigma_0$ do not commute, we do not have such a correspondence, as in general, the BW distance is not translation invariant, neither when considered as a function of $(\Sigma, \Sigma_0)$ nor when considered as a function of $(\Sigma^{1/2}, \Sigma_0^{1/2})$.

**Lemma B.5** (BW is not translation invariant). *There exist positive semidefinite matrices $(\Sigma, \Sigma_0) \in \mathcal{S}_+(n) \times \mathcal{S}_+(n)$ and a translation $T \in \mathcal{S}_+(n)$, such that $\mathcal{B}^2(\Sigma + T, \Sigma_0 + T) \neq \mathcal{B}^2(\Sigma, \Sigma_0)$. The same statement also holds for the loss $\mathcal{L}$ defined on the matrix square roots, $\mathcal{L}(\Sigma^{1/2}, \Sigma_0^{1/2}) = \mathcal{B}^2(\Sigma, \Sigma_0)$.*

*Proof.* For the first part of the statement, taking

$$\Sigma = \begin{pmatrix} 1 & 0 \\ 0 & 1 \end{pmatrix}, \quad \Sigma_0 = \begin{pmatrix} 1 & 0 \\ 0 & 2 \end{pmatrix}, \quad T = \begin{pmatrix} t & 0 \\ 0 & t \end{pmatrix}, \quad t > 0,$$

then $\mathcal{B}^2(\Sigma + T, \Sigma_0 + T) - \mathcal{B}^2(\Sigma, \Sigma_0) = \left(\sqrt{2+t} - \sqrt{1+t}\right)^2 - \left(\sqrt{2} - 1\right)^2$, which is non-zero. For the second part of the statement, if

$$\Sigma_0^{1/2} = \begin{pmatrix} 1 & 0 \\ 0 & 2 \end{pmatrix}, \quad \Sigma^{1/2} = \begin{pmatrix} 1 & 1 \\ 1 & 2 \end{pmatrix}, \quad T = \begin{pmatrix} 1 & 0 \\ 0 & 1 \end{pmatrix},$$

one computes

$$\begin{aligned}
\mathcal{L}(\Sigma^{1/2}, \Sigma_0^{1/2}) =& \|\Sigma^{1/2}\|_F^2 + \|\Sigma_0^{1/2}\|_F^2 - 2\operatorname{tr}\left(\Sigma_0^{1/2}\Sigma\Sigma_0^{1/2}\right)^{1/2} \\
=& 12 - 2\operatorname{tr}\begin{pmatrix} 2 & 6 \\ 6 & 20 \end{pmatrix}^{1/2}
\end{aligned}$$

and

$$\begin{aligned}
\mathcal{L}(\Sigma^{1/2} + T, \Sigma_0^{1/2} + T) =& \|\Sigma^{1/2} + T\|_F^2 + \|\Sigma_0^{1/2} + T\|_F^2 \\
& - 2\operatorname{tr}\left((\Sigma_0^{1/2} + T)(\Sigma^{1/2} + T)(\Sigma^{1/2} + T)(\Sigma_0^{1/2} + T)\right)^{1/2} \\
=& 28 - 2\operatorname{tr}\begin{pmatrix} 20 & 30 \\ 30 & 90 \end{pmatrix}^{1/2},
\end{aligned}$$

which gives the difference $\mathcal{L}(\Sigma^{1/2} + T, \Sigma_0^{1/2} + T) - \mathcal{L}(\Sigma^{1/2}, \Sigma_0^{1/2}) \approx 0.121229 \neq 0.$ $\qquad\square$

Lemma B.5 therefore implies that, in the general case, one cannot express the Bures-Wasserstein distance (either on the covariance or on their square roots) as a norm of a difference (otherwise, the loss would be translation invariant). This hinders a direct application of the Eckart-Young-Mirsky theorem, where the problem is cast as $\min_X \|A - X\|$ with a fixed $A$ for some unitary invariant norm. Even if this is not possible in general, a similar expression exists, relying on the following variational formulation.

**Lemma B.6.** *The Bures-Wasserstein distance between two covariance matrices $\Sigma_0$ and $\Sigma$ on $\mathcal{S}_+^n$ coincides with the variational formulation (7),*

$$\min_{U \in U(n)} \|\Sigma^{1/2} - U\Sigma_0^{1/2}\|_F^2 = \operatorname{tr}\left(\Sigma_0 + \Sigma - 2(\Sigma_0^{1/2}\Sigma\Sigma_0^{1/2})\right). \tag{14}$$

*Proof.* We write

$$\min_{U \in U(n)} \|\Sigma^{1/2} - U\Sigma_0^{1/2}\|_F^2 = \operatorname{tr}\left(\Sigma_0^{1/2} - U\Sigma_0^{1/2}\right)^\top \left(\Sigma_0^{1/2} - U\Sigma_0^{1/2}\right).$$

Let $\Sigma^{1/2}\Sigma_0^{1/2} = V_1 R^{1/2} V_2$ be the singular value decomposition for $V_1, V_2$ unitary, and $R = \left(\Sigma^{1/2}\Sigma_0\right)^\top \left(\Sigma^{1/2}\Sigma_0^{1/2}\right) = \Sigma_0^{1/2}\Sigma\Sigma_0^{1/2}$. Therefore $\operatorname{tr}\left(U^\top \Sigma^{1/2}\Sigma_0^{1/2}\right) = \operatorname{tr}\left(V_1 U^\top R^{1/2}\right)$ is maximized when $V_1 U^\top V_2$ is the identity. Thus, we get (14) and the proof is complete. $\qquad\square$

**Lemma B.7.** *For any symmetric matrix $C \in \mathcal{S}^n$, for any matrices $(A, B) \in \left(\mathbb{R}^{n \times n}\right)^2$, one has*

$$\operatorname{tr}(CAB^\top) = \operatorname{tr}(CBA^\top). \tag{15}$$

*Proof.*

$$\operatorname{tr}(CAB^\top) = \operatorname{tr}(C^\top AB^\top) = \operatorname{tr}(BA^\top C) = \operatorname{tr}(CBA^\top).$$

$\qquad\square$

### B.3 GRADIENT OF THE BURES-WASSERSTEIN LOSS

We give here the gradient of the squared-Bures-Wasserstein distance between two full-rank covariance matrices.

**Lemma B.8** (Gradient of $\mathcal{B}^2$ for full-rank matrices). *Suppose $\Sigma, \Sigma_0 \in \mathcal{S}_{++}(n)$. Then the gradient of $\mathcal{B}^2$ is given by*

$$\nabla_\Sigma \mathcal{B}^2(\Sigma, \Sigma_0) = I - \Sigma_0^{1/2}(\Sigma_0^{1/2}\Sigma\Sigma_0^{1/2})^{-1/2}\Sigma_0^{1/2}. \tag{16}$$

The proof of this Lemma is given in Appendix B.4. The right hand side of (16) is the optimal transport plan between two centered Gaussian distributions (Bhatia et al. 2019; Muzellec & Cuturi 2018, eq. 7), whose Fréchet differentiability has been explored by Kroshnin et al. (2021, Lemma A.2). This is a formulation that we use in the computation of upper bounds for the Hessian in Appendix E.1.

### B.4 PROOFS OF SECTION 3

In this section, we provide the proofs of Section 3.

*Proof of Lemma B.8.* Recall the BW distance is given by $\mathcal{B}^2(\Sigma, \Sigma_0) = \operatorname{tr}\Sigma + \operatorname{tr}\Sigma_0 - 2\operatorname{tr}\Sigma_0^{1/2}\Sigma\Sigma_0^{1/2}$. The gradient of the BW is given by

$$\nabla_\Sigma \mathcal{B}^2(\Sigma, \Sigma_0) = I - 2\nabla_\Sigma \operatorname{tr}\left(\Sigma_0^{1/2}\Sigma\Sigma_0^{1/2}\right)^{1/2} \tag{17}$$

Since $\Sigma, \Sigma_0$ are positive definite, we differentiate the $f(\Sigma, \Sigma_0) = \operatorname{tr}\left(\Sigma_0^{1/2}\Sigma\Sigma_0^{1/2}\right)^{1/2}$ with respect to $\Sigma$

$$
\begin{aligned}
\nabla_\Sigma f(\Sigma, \Sigma_0) &= \left(\partial_\Sigma \left(\Sigma_0^{1/2}\Sigma\Sigma_0^{1/2}\right)^{1/2}\right)^\top I \\
&= \left(\left(\left(\Sigma_0^{1/2}\Sigma\Sigma_0^{1/2}\right)^{1/2} \otimes I + I \otimes \left(\Sigma_0^{1/2}\Sigma\Sigma_0^{1/2}\right)^{1/2}\right)^{-1} \partial_\Sigma \left(\Sigma_0^{1/2}\Sigma\Sigma_0^{1/2}\right)\right)^\top I \\
&= \left(\Sigma_0^{1/2} \otimes \Sigma_0^{1/2}\right)\left(\left(\Sigma_0^{1/2}\Sigma\Sigma_0^{1/2}\right)^{1/2} \otimes I + I \otimes \left(\Sigma_0^{1/2}\Sigma\Sigma_0^{1/2}\right)^{1/2}\right)^{-1} I \\
&= \frac{1}{2}\left(\Sigma_0^{1/2} \otimes \Sigma_0^{1/2}\right)\left(\Sigma_0^{1/2}\Sigma\Sigma_0^{1/2}\right)^{-1/2} \\
&= \frac{1}{2}\Sigma_0^{1/2}\left(\Sigma_0^{1/2}\Sigma\Sigma_0^{1/2}\right)^{-1/2}\Sigma_0^{1/2}.
\end{aligned}
\tag{18}
$$

Substituting the above expression to (17) we get

$$\nabla_\Sigma \mathcal{B}^2(\Sigma, \Sigma_0) = I - \Sigma_0^{1/2}\left(\Sigma_0^{1/2}\Sigma\Sigma_0^{1/2}\right)^{-1/2}\Sigma_0^{1/2}. \tag{19}$$

$\square$

*Proof of Lemma 3.3.* First note that the difference between the original loss and the perturbative loss is given by

$$
\begin{aligned}
|\tilde{L}(\Sigma_\tau) - L^1(W)| &= \left|\tau n - 2\operatorname{tr}\left(\left(\Sigma_0^{1/2}\Sigma_\tau\Sigma_0^{1/2}\right)^{1/2} - \left(\Sigma_0^{1/2}WW^\top\Sigma_0^{1/2}\right)^{1/2}\right)\right| \\
&\leq \tau n + 2\operatorname{tr}\left(\left(\Sigma_0^{1/2}\Sigma_\tau\Sigma_0^{1/2}\right)^{1/2} - \left(\Sigma_0^{1/2}WW^\top\Sigma_0^{1/2}\right)^{1/2}\right)
\end{aligned}
\tag{20}
$$

Let the singular value decomposition of $(\Sigma_0^{1/2}\Sigma_\tau\Sigma_0^{1/2})^{1/2} = Q\Lambda_\tau^{1/2}Q^\top$, where

$$
\Lambda_\tau = \begin{pmatrix} \lambda_1 + \tau & & \\ & \ddots & \\ & & \lambda_r + \tau \end{pmatrix},
$$

$\lambda_1 > \lambda_2 > \ldots > \lambda_r$, and $r = \text{rank}(WW^\top)$. Similarly, we get the singular value decomposition of $(\Sigma_0^{1/1} WW^\top \Sigma_0^{1/2})^{1/2} = U\Lambda^{1/2}V^\top$, where

$$\Lambda = \begin{pmatrix} \lambda_1 & & \\ & \ddots & \\ & & \lambda_r \end{pmatrix}.$$

We know that $\text{tr}\,((\Sigma_0^{1/2}\Sigma_\tau \Sigma_0^{1/2})^{1/2}) = \text{tr}\,(\Lambda_\tau^{1/2}) = \sum_{i=1}^r (\lambda_i + \tau)^{1/2}$ since the Frobenius norm is unitary invariant. Likewise we get $\text{tr}\,((\Sigma_0^{1/2} WW^\top \Sigma_0^{1/2})^{1/2}) = \text{tr}\,(\Lambda^{1/2}) = \sum_{i=1}^r \lambda_i^{1/2}$. Next observe that the eigenvalues are distinct and in descending order. This means that we can upper bound the eigenvalues as,

$$\sum_{i=1}^r (\lambda_i + \tau)^{1/2} \le \sum_{i=1}^r \lambda_i^{1/2} + r\tau^{1/2}.$$

Therefore, we get back to Lemma 3.3 and get that

$$|\tilde{L}(\Sigma_\tau) - L^1(W)| \le \tau n + 2r\tau^{1/2}.$$

$\square$

## C  GENERAL RESULTS FOR LINEAR NETWORKS

This section deals with general properties of linear networks and their convergence in parameter space. We first recall well-known results that hold for any differentiable loss $L^1$ and its parametrization $L^N = L^1 \circ \mu$.

**Lemma C.1** (Gradient flow, Bah et al. 2021, Lemma 2.1). *For any differentiable loss $L^1$, and parametrization $L^N = L^1 \circ \mu$, such that $\mu(W_1, \ldots, W_N) = W_N \cdots W_1$, one has*

*1. For all $j \in [N]$,*

$$\nabla_{W_j} L^N(W_1, \ldots, W_N) = W_{j+1}^\top \cdots W_N^\top \nabla L^1(W) W_1^\top \cdots W_{j-1}^\top. \tag{21}$$

*2. Assume each of the $W_i(t)$ satisfies the flow (GF). Then, the product $W_{N:1} = W_N \cdots W_1$ satisfies*

$$\frac{dW(t)}{dt} = -\sum_{j=1}^N W_N \cdots W_{j+1} W_{j+1}^\top \cdots W_N^\top \nabla L^1(W) W_1^\top \cdots W_{j-1}^\top W_{j-1} \cdots W_1. \tag{22}$$

*3. For all $j \in [N]$, and all $t \geqslant 0$, we have that*

$$\frac{d}{dt}(W_{j+1}^\top(t) W_{j+1}(t)) = \frac{d}{dt}(W_j(t) W_j^\top(t)) \tag{23}$$

*4. If $W_1(0), \ldots, W_N(0)$ are balanced, then for all $t \geq 0$, $W_{j+1}^\top(t) W_{j+1}(t) = W_j(t) W_j^\top(t)$, and*

$$R(t) := \frac{dW(t)}{dt} + \sum_{j=1}^N (W(t)W^\top(t))^{\frac{N-j}{N}} \nabla L^1(W)(W^\top(t)W(t))^{\frac{j-1}{N}} = 0. \tag{24}$$

The BW loss satisfies the Łojasiewicz inequality. Indeed, the following equality can be computed.

**Lemma C.2.** *For any $W \in \mathbb{R}^{d_N \times d_0}$, and for the loss $L^1$ defined in (4), we have*

$$\|\nabla_W L^1(W)\|_F^2 = 4L^1(W). \tag{25}$$

*Proof.* This equality can be obtained directly by computation. Since

$$\nabla L^1(W) = 2W - 2\Sigma_0^{1/2}(\Sigma_0^{1/2} WW^\top \Sigma_0^{1/2})^{-1/2}\Sigma_0^{1/2}W, \tag{26}$$

we have

$$\|\nabla_W L^1(W)\|_F^2$$
$$= 4\,\mathrm{tr}\left(\left(W - \Sigma_0^{1/2}(\Sigma_0^{1/2}WW^\top\Sigma_0^{1/2})^{-1/2}\Sigma_0^{1/2}W\right)\left(W^\top - W^\top\Sigma_0^{1/2}(\Sigma_0^{1/2}WW^\top\Sigma_0^{1/2})^{-1/2}\Sigma_0^{1/2}\right)\right)$$
$$= 4\,\mathrm{tr}(WW^\top) - 4\,\mathrm{tr}\left(WW^\top\Sigma_0^{1/2}(\Sigma_0^{1/2}WW^\top\Sigma_0^{1/2})^{-1/2}\Sigma_0^{1/2}\right)$$
$$- 4\,\mathrm{tr}\left(\Sigma_0^{1/2}(\Sigma_0^{1/2}WW^\top\Sigma_0^{1/2})^{-1/2}\Sigma_0^{1/2}WW^\top\right) + 4\,\mathrm{tr}(\Sigma_0).$$

$$(27)$$

Note that the middle two terms above are the same, and they can be further simplified as

$$\mathrm{tr}\left(WW^\top\Sigma_0^{1/2}(\Sigma_0^{1/2}WW^\top\Sigma_0^{1/2})^{-1/2}\Sigma_0^{1/2}\right) = \mathrm{tr}\left(\Sigma_0^{1/2}(\Sigma_0^{1/2}WW^\top\Sigma_0^{1/2})^{-1/2}\Sigma_0^{1/2}WW^\top\right)$$
$$= \mathrm{tr}\left((\Sigma_0^{1/2}WW^\top\Sigma_0^{1/2})^{1/2}\right).$$

$$(28)$$

Combining all the terms together, we get the equality (25). $\qquad\square$

In the case of a general, twice differentiable loss $L^1$ and the parametrization $L^N = L^1 \circ \mu$, one can express the second-order differential structures of the loss.

**Lemma C.3** (Second-order differential). *Let $(\overrightarrow{U}, \overrightarrow{V}) \in \Theta \times \Theta$ be two parameters, $\overrightarrow{U} = (U_1, \ldots, U_N)$, $\overrightarrow{V} = (V_1, \ldots, V_N)$. The second-order differential of the loss $L^N$ at $\overrightarrow{W} = (W_1, \ldots, W_N) \in \Theta$ is*

$$\mathrm{d}^2 L^N(\overrightarrow{W})[\overrightarrow{U}, \overrightarrow{V}] = \sum_{i=1}^N \sum_{j \neq i} \langle U_i, W_{i+1}^\top \cdots V_j^\top \cdots W_N^\top \nabla L^1(W) W_1^\top \cdots W_{i-1}^\top \rangle$$
$$+ \sum_{i=1}^N \sum_{j=1}^N \mathrm{vec}(U_i)^\top \left(W_{i-1:1} \otimes (W_{N:i+1})^\top \cdot \nabla^2 L^1(W) \cdot (W_{j-1:1})^\top \otimes (W_{N:j+1})\right)\mathrm{vec}(V_j),$$

$$(29)$$

*where $\nabla^2 L^1(W) \in \mathbb{R}^{n^2 \times n^2}$ is the matrix such that, $\forall (U, V) \in (\mathbb{R}^{n \times n})^2$, $\mathrm{d}^2 L^1(W)[U, V] = \mathrm{vec}(U)^\top \nabla^2 L^1(W) \mathrm{vec}(V)$.*

**Corollary C.4** (Hessian of the Loss). *The Hessian of $L^N$, $\nabla^2 L^N(\theta)$, can be represented as a $d_\theta^2 \times d_\theta^2$ matrix. It is a block matrix, the blocks corresponding to different layers. Each block $\nabla^2_{W_i, W_j} L^N(\overrightarrow{W})$ has dimension $d_i d_{i-1} \times d_j d_{j-1}$, and corresponds to the differential $\mathrm{d}^2 L^N(\overrightarrow{W})[\overrightarrow{U}_i, \overrightarrow{U}_j]$, where $\overrightarrow{U}_i = (0, \ldots, 0, U_i, 0, \ldots, 0)$. The block diagonals elements are*

$$\nabla^2_{W_i} L^N(\overrightarrow{W}) = (W_{i-1:1} \otimes (W_{N:i+1})^\top) \cdot \nabla^2 L^1(W) \cdot (W_{i-1:1})^\top \otimes (W_{N:i+1}), \qquad (30)$$

*the off-diagonal terms are*

$$\nabla^2_{W_i, W_j} L^N(\overrightarrow{W}) = (W_{i-1:1} \otimes (W_{N:i+1})^\top) \cdot \nabla^2 L^1(W) \cdot ((W_{j-1:1})^\top \otimes W_{N:j+1})$$
$$+ \left[(W_{i-1}\cdots W_1 \nabla L^1(W)^\top W_N \cdots W_{j+1}) \otimes (W_{i+1}^\top \ldots W_{j-1}^\top)\right] K_{d_j d_{j-1}},$$

$$(31)$$

*where $K_{pq}$ is the pq-commutation matrix (for $X \in \mathbb{R}^{p \times q}$, $K_{pq} \mathrm{vec}\, X = \mathrm{vec}\, X^\top$).*

The invariance property on the gradient flow (GF) (Lemma C.1.3) is key in numerous analyses. Another useful property of the gradient flow (GF) is its convergence, under mild assumption on the loss $L^1$, to a critical point of $L^N$. Namely, if the trajectory $t \mapsto \overrightarrow{W}(t)$ remains bounded for all $t \geqslant 0$, and if $L^1$ is an analytic function (i.e. locally given by a power series), then (GF) converges to a critical point of $L^N$, i.e., a point $\theta^*$ so that $\nabla L^N(\theta^*) = 0$. This is stated in the next theorem.

**Theorem C.5** (Gradient flow converges to a critical point of $L^N$). *Let $L^1$ be analytic, such that the trajectory $t \mapsto \mu(\theta(t))$ remains bounded under the gradient flow evolution $\dot{\theta} = -\nabla [L^1 \circ \mu](\theta)$. Then, the flows of $W_i(t)$ given by* (GF) *and of $W(t)$ given by* (22) *are defined and bounded for all $t \geqslant 0$ and $(W_1, \ldots, W_N)$ converges to a critical point of $L^N = L^1 \circ \mu$ as $t \to \infty$.*

This result relies on the Łojasiewicz' argument for the convergence of gradient flows (Absil et al., 2005). Bah et al. (2021) show how to bound each of the different $\{W_i\}_{i=1}^N$ once the end-to-end product $W_{N:1}$ is bounded. The boundedness of $\|W\|$ can be showed depending on the loss that is considered. For example, it holds for the regularized loss $L^1_\tau$. In Appendix C we collect further general results for linear networks.

In the case of the perturbative loss introduced in (5), on can bound the norm of $W$ throughout training. Since the loss $L^1_\tau$ is analytic, one immediately gets the following result.

We give a simple test to show the boundedness of a trajectory under (GF). This is allowed by the decrease of the loss along training.

**Lemma C.6.** *Assume that, for a given loss $L^1$, there exists there exists an increasing function $f$ such that, for any $t \geqslant 0$, $\|W(t)\| \leqslant f(L^1(W(t)))$. Then, the trajectory $t \mapsto W(t)$ under* (GF) *is bounded.*

*Proof.* Under gradient flow, for any $t \geqslant 0$, $L^1(W(t)) \leqslant L^1(W(0))$. Indeed, writing the chain rule and the gradient flow (22),

$$\frac{\mathrm{d}}{\mathrm{d}t} L^1(W(t)) = \sum_j D_{W_j} L^N(W_1(t), \ldots, W_N(t)) \frac{\mathrm{d}W_j(t)}{\mathrm{d}t}$$

$$= -\sum_j \|\nabla_{W_j} L^N(W_1, \ldots, W_N)\|_F^2 \leqslant 0.$$

Therefore, for any $t \geqslant 0$, $L^1(W(t)) \leqslant L^1(W(0))$.

Now, let $f \colon \mathbb{R} \to \mathbb{R}$ be an increasing function, so that $f(L^1(W(t))) \leqslant f(L^1(W(0)))$. Therefore, if for any $t \geqslant 0$, $\|W(t)\| \leqslant f(L^1(W(t)))$, then $\|W(t)\| \leqslant f(L^1(W(t))) \leqslant f(L^1(W(0)))$ is bounded. $\qquad\square$

The assumption of Lemma C.6 is satisfied for a couple of losses, including the square loss (Bah et al., 2021) and the $L^1_\tau$ loss, as shown in Lemma C.8. It allows to consider losses that "grow with the weights", so that the end-to-end matrix is bounded when the loss converges to zero.

We now show the boundedness of the weights when considering the Bures-Wasserstein loss (4).

**Lemma C.7** (Boundedness for the BW loss $\tilde{L}$). *The loss $\tilde{L}(\Sigma)$ can be lower-bounded by the quantity $\frac{1}{2} \operatorname{tr} \Sigma - \operatorname{tr} \Sigma_0$.*

*Proof.* By definition of dual expression of the Wasserstein distance (13), $\tilde{L}(\Sigma) = \mathcal{W}_2^2(\nu_0, \nu_\theta) = \sup_{f \in \mathcal{L}^1(\nu_\theta)} \int f(x) \, \mathrm{d}\nu_\theta + \int f^{\|\cdot\|^2}(y) \, \mathrm{d}\nu_0(y)$, with $\nu_\theta = \mathcal{N}(0, \Sigma)$, $\nu_0 = \mathcal{N}(0, \Sigma_0)$ and $f^{\|\cdot\|^2}$ the $\|\cdot\|^2$-transform of $f$ defined as $\forall y \in \mathbb{R}^d$, $f^{\|\cdot\|^2}(y) = \inf_{x \in \mathbb{R}^d} \|x - y\|^2 - f(x)$.

With $\tilde{f} \colon x \mapsto \frac{1}{2}\|x\|^2$, the $\|\cdot\|^2$-transform of $\tilde{f}$ is $\tilde{f}^{\|\cdot\|^2} \colon y \mapsto -\|y\|^2$, and we get

$$\tilde{L}(\Sigma) = \mathcal{W}_2^2(\nu_0, \nu_\theta) \geqslant \frac{1}{2} \int \|x\|^2 \, \mathrm{d}\nu_\theta(x) - \int \|y\|^2 \, \mathrm{d}\nu_0(y) = \frac{1}{2} \operatorname{tr} \Sigma - \operatorname{tr} \Sigma_0. \qquad (32)$$

as claimed. $\qquad\square$

**Lemma C.8** (Boundedness for the loss $L^1_\tau$). *The norm of the end-to-end matrix $W$ is upper-bounded when using the loss $L^1_\tau$ defined in* (5).

*Proof.* With $\varphi_\tau(\Sigma) = \Sigma + \tau I_n =: \Sigma_\tau$, the loss $L^1_\tau$ satisfies

$$L^1_\tau(W) = \tilde{L}(\varphi_\tau(\pi(W))) \overset{(32)}{\geqslant} \frac{1}{2} \operatorname{tr} \Sigma_\tau - \operatorname{tr} \Sigma_0 = \frac{1}{2} \operatorname{tr} WW^\top - \operatorname{tr} \Sigma_0 + \frac{n}{2}\tau \qquad (33)$$

$$\implies \quad \sqrt{2L^1_\tau(W) + 2\operatorname{tr} \Sigma_0 - n\tau} \geqslant \|W\|. \qquad (34)$$

Therefore, there exists an increasing function $f$ such that $\|W\| \leqslant f(L_\tau^1(W))$. Since the loss decreases under gradient flow, one has

$$\|W(t)\| \leqslant \sqrt{2L_\tau^1(W(0)) + 2\operatorname{tr}\Sigma_0 - n\tau}, \tag{35}$$

and the boundedness of $t \mapsto W(t)$ is shown. $\qquad\square$

**Corollary C.9.** *For the Bures-Wasserstein loss, if $WW^\top$ is positive definite, and loss is differentiable, and the norm of the weight throughout the training is uniformly bounded*

$$\|W\| \leq \sqrt{2L^1(W(0)) + 2\operatorname{tr}\Sigma_0}, \tag{36}$$

*by using similar arguments as in Lemma C.8.*

**Lemma C.10.** *The gradient flow* (GF) *on the perturbative loss* (5) *converges to a critical point $\theta^*$ of $L_\tau^N$ in the limit.*

This property of the gradient flow is necessary in order to prove the convergence of the training to a minimizer of $L_\tau^1$. At first glance, the critical points of $L_\tau^N$ do not correspond in general to critical points of $L_\tau^1$ since the parametrization $\mu$ also comes into play. This led Trager et al. (2020) to distinguish between the *pure* and *spurious* critical points; i.e., the points that are shared between $L^N$ and $L^1$, and those that are exclusive to $L^N$.

## D PROOFS OF SECTION 4

In this section, we provide the proofs for the critical points of $L^1|_{\mathcal{M}_k}$ and $L_\tau^1|_{\mathcal{M}_k}$.

### D.1 CRITICAL POINTS OF $L^1|_{\mathcal{M}_k}$

First, the loss $L^1$ is expressed on the manifolds $\mathcal{M}_k$ (Lemma D.2), where it is differentiable (Lemma D.3). Then, necessary conditions (Lemma D.5) on the critical points can be expressed, leading to the proof of Theorem 4.2. The second part of Theorem 4.2 is then proven by evaluating the loss at the critical points found, and ranking them.

Recall Definition 4.1. Computing the differential of the restriction $L^1|_{\mathcal{M}_k}$ will allow to characterize the different critical points.

**Definition D.1** (Gradient). Given an embedded manifold $\mathcal{M}$ and a function with a smooth restriction $f|_{\mathcal{M}}$, the gradient of $f|_{\mathcal{M}_k}$ at $x \in \mathcal{M}$ is the (unique) element of the tangent space $T_x\mathcal{M}$ such that, for all $v \in T_x\mathcal{M}$, $\mathrm{d}f|_{\mathcal{M}}(x)[v] = \langle \nabla f|_{\mathcal{M}}(x), v\rangle$.

We begin by expressing the loss $L^1|_{\mathcal{M}_k}$ with the Singular Value Decomposition (SVD) of $\Sigma_0^{1/2}W$.

**Lemma D.2.** *Let $USV^\top = \Sigma_0^{1/2}W$ be a thin SVD of $\Sigma_0^{1/2}W$, so that $U \in \mathbb{R}^{n\times k}$, $V \in \mathbb{R}^{m\times k}$, $U^\top U = V^\top V = I_k$, $S = \operatorname{diag}(s_1, \ldots, s_k) \in \mathbb{R}^{k\times k}$, where $k = \operatorname{rank}\Sigma_0^{1/2}W = \operatorname{rank}W$. The loss $L^1$ from* (4) *on $\mathcal{M}_k$ can be expressed as*

$$L^1|_{\mathcal{M}_k}(W) = \|W\|_F^2 + \|\Sigma_0^{1/2}\|_F^2 - 2\operatorname{tr}S. \tag{37}$$

*Proof.* If $USV^\top = \Sigma_0^{1/2}W$ is a thin SVD of $\Sigma_0^{1/2}W$, then $\left(\Sigma_0^{1/2}W\left(\Sigma_0^{1/2}W\right)^\top\right)^{1/2} = USU^\top$. Therefore, the expression of the loss $L^1$ given by (4) can be written as

$$L^1|_{\mathcal{M}_k}(W) = \operatorname{tr}(WW^\top) + \operatorname{tr}\Sigma_0 - 2\operatorname{tr}(USU^\top) = \|W\|_F^2 + \|\Sigma_0^{1/2}\|_F^2 - 2\operatorname{tr}S$$

as claimed. $\qquad\square$

We then give the gradient of $L^1|_{\mathcal{M}_k}$.

**Lemma D.3** (Gradient of $L^1|_{\mathcal{M}_k}$). *Let $(n, m) \in \mathbb{N}_*^2$, and let $k \leqslant \min\{n, m\}$. The loss $L^1|_{\mathcal{M}_k}$ (as given in* (37)) *is twice continuously differentiable on $\mathcal{M}_k$. With $W \in \mathcal{M}_k$ and $USV^\top = \Sigma_0^{1/2}W$ a thin SVD of $\Sigma_0^{1/2}W$, its gradient is*

$$\nabla L^1|_{\mathcal{M}_k}(W) = 2W - 2\Sigma_0^{1/2}UV^\top. \tag{38}$$

In order to derive this expression, the differential of the singular values is required. But first, a note on the differential notation used throughout the derivations.

**Notation** (Differential). The differential of a function $f$ can be written using different formalisms. Explicitly, $\mathrm{d}f(X)[H]$ is the differential of $f$ at $X$ in the direction $H$. Sometimes, with $Y = f(X)$, the shorthand notation $\mathrm{d}Y$ is preferred, where the same symbol is used for both the variable and the function. In this case, it is assumed that the direction $H$ is a small perturbation $\mathrm{d}X$ around $X$. For instance, if $Y = f(X) = XX^\top$, then $\mathrm{d}Y = \mathrm{d}XX^\top + X\,\mathrm{d}X^\top$, which would be written $\mathrm{d}f(X)[H] = HX^\top + XH^\top$ with the full notation.

**Lemma D.4** (Differential of the SVD). *Let $k \leqslant \min\{n, m\}$ and let $X \in \mathcal{M}_k$ be a matrix with $\operatorname{rank} X = k$. Let $USV^\top = X$ be a thin SVD of $X$, with $U \in \mathbb{R}^{n \times k}$, $S \in \mathbb{R}^{k \times k}$, $V \in \mathbb{R}^{m \times k}$, $S$ diagonal and $U^\top U = V^\top V = I_k$. Then, the differential $\mathrm{d}S$ is*

$$\mathrm{d}S = I_k \odot (U^\top \,\mathrm{d}XV),$$

*where $A \odot B$ denotes the Hadamard product between $A$ and $B$.*

*Proof.* Let $USV^\top = X$ be the decomposition as given in the lemma statement. The differential rules ensure that

$$\mathrm{d}X = \mathrm{d}USV^\top + U\,\mathrm{d}SV^\top + US\,\mathrm{d}V^\top.$$

This implies that

$$U^\top \,\mathrm{d}XV = U^\top \,\mathrm{d}USV^\top V + U^\top U\,\mathrm{d}SV^\top V + U^\top US\,\mathrm{d}V^\top V$$
$$= U^\top \,\mathrm{d}US + \mathrm{d}S + S\,\mathrm{d}V^\top V$$
$$\implies \quad \mathrm{d}S = U^\top \,\mathrm{d}XV - U^\top \,\mathrm{d}US - S\,\mathrm{d}V^\top V.$$

Since $U^\top U = I_k$, $\mathrm{d}U^\top U + U^\top \,\mathrm{d}U = 0$, and $A := U^\top \,\mathrm{d}U = -\mathrm{d}U^\top U = -A^\top$. Likewise, $B := V^\top \,\mathrm{d}V$ is also antisymmetric. The matrices $A$ and $B$ being antisymmetric, their diagonals are null; hence so are the diagonals of $AS$ and $SB$, i.e. $I_k \odot (AS) = I_k \odot (SB) = 0$. Since $S$ is constrained to be diagonal, $\mathrm{d}S$ must also be diagonal, i.e. $I_k \odot \mathrm{d}S = \mathrm{d}S$. Therefore,

$$\mathrm{d}S = I_k \odot (U^\top \,\mathrm{d}XV)$$

as was claimed. $\qquad\square$

*Proof of lemma D.3.* For $W \in \mathcal{M}_k$, let $USV^\top = \Sigma_0^{1/2}W$ be a thin SVD of $\Sigma_0^{1/2}W =: X$. Lemma D.2 ensures that

$$L^1|_{\mathcal{M}_k}(W) = \|W\|_F^2 + \|\Sigma_0^{1/2}\|_F^2 - 2\operatorname{tr} S. \tag{39}$$

According to Lemma D.4, the matrix $S$ is differentiable and has differential $\mathrm{d}S = I_k \odot (U^\top \,\mathrm{d}XV)$. Therefore, the loss $L^1|_{\mathcal{M}_k}$ is differentiable. With the fact that $\mathrm{d}\operatorname{tr} S = \operatorname{tr} \mathrm{d}S$ (see e.g. (Magnus & Neudecker, 2019, Chap. 8, Eq. 18)), we can compute

$$\mathrm{d}\operatorname{tr} S = \operatorname{tr} \mathrm{d}S = \operatorname{tr}\left(I_k \odot (U^\top \,\mathrm{d}XV)\right) = \operatorname{tr}\left(U^\top \,\mathrm{d}XV\right)$$
$$= \langle UV^\top, \mathrm{d}X\rangle = \langle UV^\top, \Sigma_0^{1/2}\,\mathrm{d}W\rangle = \langle \Sigma_0^{1/2}UV^\top, \mathrm{d}W\rangle.$$

Moreover, $\mathrm{d}\|W\|_F^2 = 2\langle W, \mathrm{d}W\rangle$, and so

$$\mathrm{d}L^1|_{\mathcal{M}_k}(W) = \mathrm{d}\|W\|_F^2 - 2\,\mathrm{d}\operatorname{tr} S = 2\langle W - \Sigma_0^{1/2}UV^\top, \mathrm{d}W\rangle,$$

and

$$\nabla L^1|_{\mathcal{M}_k}(W) = 2\left(W - \Sigma_0^{1/2}UV^\top\right).$$

Since matrices $(U, V)$ are continuously differentiable on $\mathcal{M}_k$, $\nabla L^1|_{\mathcal{M}_k}(W) = 2(W - \Sigma_0^{1/2}UV^\top)$ is again continuously differentiable, and $L^1|_{\mathcal{M}_k}$ is twice continuously differentiable. $\qquad\square$

We are now ready to give the proof of Theorem 4.2. We divide the proof into necessary and sufficient conditions for a point to be a critical point of $L^1|_{\mathcal{M}_k}$.

**Lemma D.5** (Necessary condition on the critical points of $L^1_{\mathcal{W}}|_{\mathcal{M}_k}$). *Assume $\Sigma_0$ has $n$ distinct eigenvalues. Let $W^* \in \mathcal{M}_k$ be a critical point of $L^1|_{\mathcal{M}_k}$. Then, with $U^* S^* V^{*\top} = \Sigma_0^{1/2} W^*$ a thin SVD of $\Sigma_0^{1/2} W^*$, and $\Omega \Lambda \Omega^\top = \Sigma_0$ an spectral decomposition of $\Sigma_0$ (i.e. with $\Omega \in \mathcal{O}(n)$), there exists $\mathcal{J}_k \subseteq \{1, \ldots, n\}$, such that $S^* = \bar{\Lambda}_{\mathcal{J}_k}$ and $U^* = \Omega_{\mathcal{J}_k}$.*

*Proof.* Since $W^* \in \mathcal{M}_k$, and $U^* S^* V^{*\top} = \Sigma_0^{1/2} W^*$ is a thin SVD of $\Sigma_0^{1/2} W^*$, this means that $S^* \in \mathbb{R}^{k \times k}$. Then,

$$\nabla L^1(W^*) = 0 \implies W^* = \Sigma_0^{1/2} U^* V^{*\top}, \qquad \text{by (38)}$$
$$\implies \Sigma_0^{1/2} W^* = \Sigma_0 U^* V^{*\top}$$
$$\implies U^* S^* V^{*\top} = \Sigma_0 U^* V^{*\top}$$
$$\implies S^* = U^{*\top} \Sigma_0 U^*, \qquad U^{*\top} U^* = I_k, \ V^{*\top} V^* = I_k.$$

Therefore, $U^{*\top} \Sigma_0 U^*$ must be diagonal; and since $U^*$ is semi-orthogonal, this is the case if and only if the vectors in $U^*$ are eigenvectors for $\Sigma_0$, by uniqueness of the spectral decomposition of $\Sigma_0$. Therefore, there exist $j_1, \ldots, j_k$ indices between 1 and $n$ such that $U^* = \begin{pmatrix} \omega_{j_1} & \cdots & \omega_{j_k} \end{pmatrix} = \Omega_{\mathcal{J}_k}$, in which case

$$S^* = \Omega_{\mathcal{J}_k}{}^\top \Sigma_0 \Omega_{\mathcal{J}_k} = \begin{pmatrix} \lambda_{j_1} & & \\ & \ddots & \\ & & \lambda_{j_k} \end{pmatrix} = \bar{\Lambda}_{\mathcal{J}_k}.$$

$\square$

Now we are ready to prove the first part of Theorem 4.2.

*Proof of Theorem 4.2, first part.* Consider the expression for the gradient of $L^1|_{\mathcal{M}_k}$ given in (38). The necessary condition follows from Lemma D.5, since

$$\nabla L^1|_{\mathcal{M}_k}(W^*) = 0 \implies \Sigma_0^{1/2} W^* = \Omega_{\mathcal{J}_k} \bar{\Lambda}_{\mathcal{J}_k} V^\top$$
$$\implies W^* = \Sigma_0^{-1/2} \Omega_{\mathcal{J}_k} \bar{\Lambda}_{\mathcal{J}_k} V^\top$$
$$= \Omega \Lambda^{-1/2} \Omega^\top \Omega_{\mathcal{J}_k} \bar{\Lambda}_{\mathcal{J}_k} V^\top$$
$$= \Omega \Lambda^{-1/2} \Lambda_{\mathcal{J}_k} V^\top$$
$$= \Omega \Lambda^{1/2} P_{\mathcal{J}_k} V^\top$$
$$= \Omega P_{\mathcal{J}_k} \bar{\Lambda}_{\mathcal{J}_k}^{1/2} V^\top$$
$$= \Omega_{\mathcal{J}_k} \bar{\Lambda}_{\mathcal{J}_k}^{1/2} V^\top,$$

which corresponds to the necessary condition in Theorem 4.2.

The sufficient condition can be verified as follows. With $W^* = \Omega_{\mathcal{J}_k} \bar{\Lambda}_{\mathcal{J}_k}^{1/2} V^\top$, one has $\Sigma_0^{1/2} W^* = \Omega \Lambda^{1/2} \Omega^\top \Omega_{\mathcal{J}_k} \bar{\Lambda}_{\mathcal{J}_k}^{1/2} V^\top = \Omega_{\mathcal{J}_k} \bar{\Lambda}_{\mathcal{J}_k} V^\top$, and, as this is a correct thin SVD of $\Sigma_0^{1/2} W^*$, Lemma D.3 gives

$$\nabla L^1|_{\mathcal{M}_k}(W^*) = 2(W^* - \Sigma_0^{1/2} \Omega_{\mathcal{J}_k} V^\top).$$

Further,

$$\Sigma_0^{1/2} \Omega_{\mathcal{J}_k} = \Omega \Lambda^{1/2} \Omega^\top \Omega_{\mathcal{J}_k}$$
$$= \Omega \Lambda^{1/2} P_{\mathcal{J}_k}$$
$$= \Omega P_{\mathcal{J}_k} \bar{\Lambda}_{\mathcal{J}_k}^{1/2}$$
$$= \Omega_{\mathcal{J}_k} \bar{\Lambda}_{\mathcal{J}_k}^{1/2}.$$

Hence
$$\nabla L^1|_{\mathcal{M}_k}(W^*) = 2(W^* - \Sigma_0^{1/2}\Omega_{\mathcal{J}_k}V^\top) = 2(\Omega_{\mathcal{J}_k}\bar{\Lambda}_{\mathcal{J}_k}^{1/2}V^\top - \Omega_{\mathcal{J}_k}\bar{\Lambda}_{\mathcal{J}_k}^{1/2}V^\top) = 0,$$
and the sufficient condition is verified. $\qquad\square$

Now, the loss can be evaluated at the critical points in order to statute on its minimizers.

**Corollary D.6** (Value of $L^1$ at the critical points). *The value of the loss $L^1$ at a critical point $W^* = \Omega_{\mathcal{J}_k}\bar{\Lambda}_{\mathcal{J}_k}^{1/2}V^\top$ is $L^1(W^*) = \operatorname{tr}\Lambda - \operatorname{tr}\bar{\Lambda}_{\mathcal{J}_k} = \sum_{i\notin\mathcal{J}_k}\lambda_i$.*

*Proof.* For $k \geqslant 0$, let $W^*$ be a critical point of $L^1|_{\mathcal{M}_k}$. From Theorem 4.2, with $\Sigma_0 = \Omega\Lambda\Omega^\top$ a spectral decomposition of $\Sigma_0$, there exists a set $\mathcal{J}_k$ and a semi-orthogonal matrix $V \in \mathbb{R}^{n\times k}$ such that $W^* = \Omega_{\mathcal{J}_k}\bar{\Lambda}_{\mathcal{J}_k}^{1/2}V^\top$. One can then compute the value of the loss at $W^*$:

$$
\begin{aligned}
L^1(W^*) &= \operatorname{tr} W^*W^{*\top} + \operatorname{tr}\Sigma_0 - 2\operatorname{tr}\left((\Sigma_0^{1/2}W^*)(\Sigma_0^{1/2}W^*)^\top\right)^{1/2} \\
&= \operatorname{tr}\Omega_{\mathcal{J}_k}\bar{\Lambda}_{\mathcal{J}_k}\Omega_{\mathcal{J}_k} + \operatorname{tr}\Lambda - 2\operatorname{tr}\left(\Omega_{\mathcal{J}_k}\bar{\Lambda}_{\mathcal{J}_k}^2\Omega_{\mathcal{J}_k}^\top\right)^{1/2} \\
&= \operatorname{tr}\bar{\Lambda}_{\mathcal{J}_k} + \operatorname{tr}\Lambda - 2\operatorname{tr}\bar{\Lambda}_{\mathcal{J}_k} \\
&= \operatorname{tr}\Lambda - \operatorname{tr}\bar{\Lambda}_{\mathcal{J}_k}.
\end{aligned}
$$

$\qquad\square$

We now have all we need to prove the second part of Theorem 4.2.

*Proof of Theorem 4.2, second part.* The first part of the statement is readily implied by Corollary D.6, as the eigenvalues are in decreasing order. The second part is implied by the fact that the minimum $L^1|_{\mathcal{M}_k}$ is indeed achieved for any $k \leqslant n$ (by selecting the $k$ largest eigenvalues of $\Sigma_0$) and the optimal value of the loss $L_k^*$ is smaller when considering more eigenvalues, i.e. $\min_{\mathcal{M}_k} L^1 \leqslant \min_{\mathcal{M}_{<k}} L^1$. $\qquad\square$

Moreover, it can be shown that only one point per set $\mathcal{M}_k$ is a minimizer of the loss $L^1|_{\mathcal{M}_k}$; all other points are (strict) saddle points. We recall the definition of a strict saddle point: a point where there exist a descent direction.

**Definition D.7** (Strict saddle point). *A critical point $x$ of a function $f$ is said to be a strict saddle point if the Hessian of $f$ at $x$ has a strict negative eigenvalue. If all critical points of $f$ are either a strict saddle point or the global minimizer, the we say that $f$ satisfies the strict saddle point property.*

If the gradient flow can be expressed on a manifold, with a Riemannian gradient corresponding to a given metric, there is an equivalent definition of those saddle points, which will be handy to use. See (Bah et al., 2021, §6.1) for the details.

**Proposition D.8.** *The loss $L^1|_{\mathcal{M}_k}$ satisfies the strict saddle point property.*

*Proof.* Let $\Sigma_0 = U\Lambda U^\top$ be the spectral decomposition of $\Sigma_0$ with decreasing eigenvalues. For $k \in \mathbb{N}$, according to Theorem 4.2, $W^*$ is a critical point of $L^1|_{\mathcal{M}_k}$ if and only if there exists $\mathcal{J}_k \subset \{1,\dots,n\}$, such that $W^* = U_{\mathcal{J}_k}\Lambda_{\mathcal{J}_k}^{1/2}V^\top$, with any $V \in \mathbb{R}^{m\times k}$ so that $V^\top V = I_k$. If $\mathcal{J}_k = \{1,\dots,k\}$, $W^*$ is a global minimum of $L^1|_{\mathcal{M}_k}$, as shown in Corollary D.6, and the proposition holds.

Assume $\mathcal{J}_k \neq \{1,\dots,k\}$, then there exists $j_0 \in \mathcal{J}_k$ such that $\lambda_{j_0} < \lambda_k$, and there exists $j_1 \notin \mathcal{J}_k$ but $j_1 \in \{1,2,\dots,k\}$ such that $\lambda_{j_1} > \lambda_{j_0}$. We will show that $W^*$ is a strict saddle point of $L^1|_{\mathcal{M}_k}$.

The critical point $W^*$ can equivalently be expressed as

$$W^* = \Sigma_0^{-1/2}\sum_{i\in\mathcal{J}_k}\lambda_i u_i v_i^\top, \tag{40}$$

where $u_i, v_i$ are corresponding orthogonal uni-vectors in $U$ and $V$, and $\lambda_i$ are eigenvalues in $\Lambda$.

The key is, by the following perturbation, for $t \in (-1, 1)$, we define

$$u_{j_0}(t) = tu_{j_1} + \sqrt{1 - t^2} u_{j_0}$$

and the curve $\gamma : (-1, 1) \mapsto \mathcal{M}_k$. We look at the perturbative matrix

$$\gamma(t) = \Sigma_0^{-1/2}\Big(\lambda_{j_0} u_{j_0}(t) v_{j_0}^\top + \sum_{i \in \mathcal{J} \backslash \{j_0\}} \lambda_i u_i v_i^\top\Big).$$

Note that $\gamma(0) = W$. Recall $L^1(W) = \mathrm{tr}\left(WW^\top + \Sigma_0 - 2\left(\Sigma_0^{1/2} WW^\top \Sigma_0^{1/2}\right)^{1/2}\right)$. It is enough to show that (Bah et al., 2021, §6.1)):

$$\frac{d^2}{dt^2} L^1(\gamma(t))\Big|_{t=0} < 0.$$

We check it term by term,

$$\begin{aligned}
\mathrm{tr}\left(\gamma(t)\gamma(t)^\top\right) &= \mathrm{tr}\left(\Sigma_0^{-1/2}\big(\lambda_{j_0} u_{j_0}(t) v_{j_0}^\top + \sum_{i \in \mathcal{J} \backslash \{j_0\}} \lambda_i u_i v_i^\top\big)\big(\lambda_{j_0} u_{j_0}(t) v_{j_0}^\top + \sum_{i \in \mathcal{J} \backslash \{j_0\}} \lambda_i u_i v_i^\top\big)^\top \Sigma_0^{-1/2}\right) \\
&= \mathrm{tr}\left(\Sigma_0^{-1}\big(\lambda_{j_0}^2 u_{j_0}(t) u_{j_0}(t)^\top + \sum_{i \in \mathcal{J} \backslash \{j_0\}} \lambda_i^2 u_i u_i^\top\big)\right) \\
&= \mathrm{tr}\left(\big(\sum_{1 \le i \le n} \lambda_i^{-1} u_i u_i^\top\big)\big(\lambda_{j_0}^2 u_{j_0}(t) u_{j_0}(t)^\top + \sum_{i \in \mathcal{J} \backslash \{j_0\}} \lambda_i^2 u_i u_i^\top\big)\right) \\
&= \frac{\lambda_{j_0}^2}{\lambda_{j_1}} t^2 + \lambda_{j_0}(1 - t^2) + \sum_{i \in \mathcal{J} \backslash \{j_0\}} \lambda_i^2,
\end{aligned}$$

and

$$\begin{aligned}
\mathrm{tr}&\left(\big(\Sigma_0^{1/2}\gamma(t)\gamma(t)^\top\Sigma_0^{1/2}\big)^{1/2}\right) \\
&= \mathrm{tr}\left(\big(\big(\lambda_{j_0} u_{j_0}(t) v_{j_0}^\top + \sum_{i \in \mathcal{J} \backslash \{j_0\}} \lambda_i u_i v_i^\top\big)\big(\lambda_{j_0} u_{j_0}(t) v_{j_0}^\top + \sum_{i \in \mathcal{J} \backslash \{j_0\}} \lambda_i u_i v_i^\top\big)^\top\big)^{1/2}\right) \\
&= \mathrm{tr}\left(\big(\lambda_{j_0}^2 u_{j_0}(t) u_{j_0}(t)^\top + \sum_{i \in \mathcal{J} \backslash \{j_0\}} \lambda_i^2 u_i u_i^\top\big)^{1/2}\right) \\
&= \mathrm{tr}\left(\big(t^2 \lambda_{j_0}^2 u_{j_1} u_{j_1}^\top + (1 - t^2)\lambda_{j_0}^2 u_{j_0} u_{j_0}^\top + \sum_{i \in \mathcal{J} \backslash \{j_0\}} \lambda_i^2 u_i u_i^\top\big)^{1/2}\right) \\
&= t|\lambda_{j_0}| + \sqrt{1 - t^2}|\lambda_{j_0}| + \sum_{i \in \mathcal{J} \backslash \{j_0\}} |\lambda_i|.
\end{aligned}$$

Thus, since $\lambda_{j_1} > \lambda_{j_0}$,

$$\frac{d^2}{dt^2} L^1(\gamma(t))\Big|_{t=0} = 2(\lambda_{j_0}^2 \lambda_{j_1}^{-1} - \lambda_{j_0}) - |\lambda_{j_0}| < 0.$$

This completes the proof. □

The loss $L_\tau^1$ satisfies the strict-saddle point property in a similar fashion.

**Lemma D.9.** *The loss $L_\tau^1|_{\mathcal{M}_k}$ satisfies the strict saddle point property.*

*Proof of Lemma D.9.* The proof of Proposition D.8 can be adapted, with the expression of the critical points as, if $\Sigma_0 = \Omega\Lambda\Omega^\top$, and with $V \in \mathbb{R}^{n \times k}$ any semi-orthogonal matrix, $W^* = (\Sigma_0 - \tau I_n)^{-1/2} \sum_{j=1}^n (\lambda_i - \tau)\omega_i v_i^\top$. □

### D.2 CRITICAL POINTS OF THE PERTURBATIVE LOSS

In this section, we provide the different derivations for Section 4.2. The structure is similar to Theorem 4.2; first the gradient of $L_\tau^1$ is computed, then the critical points are characterized and ranked.

**Lemma D.10** (Differential of $\tilde{L}$). *The differential of $\tilde{L}$ on $\mathcal{S}_{++}(n)$ is*

$$\forall\, \Sigma \in \mathcal{S}_{++}(n),\ X \in \mathcal{S}(n),\quad \mathrm{d}\tilde{L}(\Sigma)[X] = \mathrm{tr}\,(X - \Sigma_0^{1/2}[\Sigma_0^{1/2}\Sigma\Sigma_0^{1/2}]^{-1/2}\Sigma_0^{1/2}X).$$

**Corollary D.11** (Gradient of $\tilde{L}$). *The gradient of $\tilde{L}$ on $\mathcal{S}_{++}(n)$ is*

$$\forall\, \Sigma \in \mathcal{S}_{++}(n),\quad \nabla\tilde{L}(\Sigma) = I - \Sigma_0^{1/2}[\Sigma_0^{1/2}\Sigma\Sigma_0^{1/2}]^{-1/2}\Sigma_0^{1/2}.$$

**Lemma D.12** (Gradient of $L_\tau^1$). *The loss $L_\tau^1$ has the following gradient*

$$\forall\, W \in \mathbb{R}^{n\times m},\quad \nabla L_\tau^1(W) = 2\big(W - \Sigma_0^{1/2}\big[\Sigma_0^{1/2}(WW^\top + \tau I_n)\Sigma_0^{1/2}\big]^{-1/2}\Sigma_0^{1/2}W\big). \tag{41}$$

*Proof.* This results comes from the chain rule for the loss $L_\tau^1(W) = \tilde{L}\circ\varphi_\tau\circ\pi(W)$. With $\Sigma = \pi(W) = WW^\top$ and $\Sigma_\tau = \varphi_\tau(\Sigma) = \Sigma + \tau I_n$, and since $\mathrm{d}\pi(W)[Z] = WZ^\top + ZW^\top$ and $\mathrm{d}\varphi_\tau(\Sigma) = \mathrm{id}$, one has

$$\mathrm{d}L_\tau^1(W)[Z] = \mathrm{d}(\tilde{L}\circ\varphi_\tau\circ\pi)(W)[Z]$$
$$= \mathrm{d}\tilde{L}(\Sigma_\tau)\Big[\mathrm{d}\varphi_\tau(\Sigma)\Big[\mathrm{d}\pi(W)[Z]\Big]\Big]$$
$$= \mathrm{d}\tilde{L}(\Sigma_\tau)[WZ^\top + ZW^\top]$$
$$\langle\nabla L_\tau^1(W), Z\rangle = \langle\nabla\tilde{L}(\Sigma_\tau), WZ^\top + ZW^\top\rangle$$
$$\iff \quad \nabla L_\tau^1(W) = (\nabla\tilde{L}(\Sigma_\tau) + \nabla\tilde{L}(\Sigma_\tau)^\top)W$$
$$= 2(W - \Sigma_0^{1/2}[\Sigma_0^{1/2}\Sigma_\tau\Sigma_0^{1/2}]^{-1/2}\Sigma_0^{1/2}W).$$

$\square$

*Proof of Theorem 4.4.* The eigenvectors of $WW^\top + \tau$ are the same as $WW^\tau$, and the eigenvalues are shifted by $\tau$. Therefore, the expression of the critical points in the original loss can be adapted, so that the modified critical points have the same left singular vectors and shifted singular values. This leads to having $W^* = \Omega_{\mathcal{J}_k}(\bar{\Lambda}_{\mathcal{J}_k} - \tau I_k)^{1/2}V^\top = \Omega\begin{pmatrix}(\bar{\Lambda}_{\mathcal{J}_k} - \tau I_k)^{1/2} & \\ & \mathbf{0}_{n-k\times n-k}\end{pmatrix}\begin{pmatrix}V & \hat{V}\end{pmatrix}^\top$,

with $V \in \mathbb{R}^{m\times k}$ such that $V^\top V = I_k$. One checks that $\nabla L_\tau^1(W^*) = 0$.

The value at such a critical point $W^* = \Omega_{\mathcal{J}_k}(\bar{\Lambda}_{\mathcal{J}_k} - \tau I_k)^{1/2}V^\top$ is $L_\tau^1(W^*) = \sum_{j\notin\mathcal{J}_r}\lambda_j - 2\sqrt{\tau\lambda_j}$, which is uniquely minimized for $\mathcal{J}_r = [k]$ when the eigenvalues of $\Sigma_0$ are distinct and in descending order. $\square$

### D.3 PROOF OF PROPOSITION 4.5

We state here the proof on Proposition 4.5. We will transfer the results obtained on the space of linear maps $\mathcal{M}_{\leqslant k}$ to the space of covariance matrices $\mathcal{S}_+(k, n)$. Borrowing the terminology from Levin et al. (2022), we introduce the following notations and definitions. Let $\mathcal{M}$ be any smooth manifold, $\mathcal{E}$ a linear space, $\varphi\colon \mathcal{M} \to \mathcal{E}$ a smooth (over)parametrization (or lift) of the search space $\mathcal{X} = \varphi(\mathcal{M}) \subseteq \mathcal{E}$. The following problems are considered

$$\min_{x\in\mathcal{X}} f(x) \tag{P}$$
$$\min_{y\in\mathcal{M}} f\circ\varphi(y), \tag{Q}$$

where we assume that $f\colon \mathcal{E} \to \mathbb{R}$ is smooth, and hence so is $g := f\circ\varphi$. The following property is relevant for us.

**Definition D.13** (Levin et al. 2022, Definition 2.7)**.** The lift $\varphi\colon \mathcal{M} \to \mathcal{X}$ satisfies the "1 $\Rightarrow$ 1" property at $y$ if for all differentiable $f\colon \mathcal{X} \to \mathbb{R}$, if $y$ is a critical point for (Q), then $x = \varphi(y)$ is a critical point for (P).

Recall that

$$\mathcal{S}_+(k, n) = \{\Sigma \in \mathcal{S}(n)\colon \Sigma \succcurlyeq 0, \operatorname{rank}(\Sigma) = k\}, \tag{42}$$

and let

$$\mathcal{S}_+(\leqslant k, n) = \{\Sigma \in \mathcal{S}(n)\colon \Sigma \succcurlyeq 0, \operatorname{rank}(\Sigma) \leqslant k\}. \tag{43}$$

We will make use of the following result from Levin et al. (2022).

**Proposition D.14** (Levin et al. 2022, Proposition 3.4)**.** *Let $k \leqslant n$, and let $\varphi\colon \mathbb{R}^{n\times k} \to \mathcal{S}_+(\leqslant k, n)$ be the parametrization $\varphi(R) = RR^\top$. Then, $\varphi$ satisfies the "1 $\Rightarrow$ 1" property at $R \in \mathbb{R}^{n\times k}$ if and only $\operatorname{rank} R = k$.*

Said differently, the condition at which a critical point on the space $\mathbb{R}^{n\times k}$ is such that its image through $\varphi$ is also a critical point on $\mathcal{S}_+(k, n)$ are exactly the points of full rank $\mathbb{R}_*^{n\times k}$. The image of the parametrization $\mu$ is $\mathbb{R}_{\leqslant k}^{n\times m}$. Therefore, we need to adapt Proposition D.14 in order to work on $\mathbb{R}_{\leqslant k}^{n\times k}$ — which is not smooth — instead of $\mathbb{R}^{n\times k}$. This is performed in the next proposition.

**Proposition D.15.** *Let $k \leqslant \min(n, m)$, and let $\pi\colon \mathbb{R}_{\leqslant k}^{n\times m} \to \mathcal{S}_+(\leqslant k, n)$ be the parametrization $\pi(W) = WW^\top$. Then, $\pi$ satisfies the "1 $\Rightarrow$ 1" property at $W \in \mathbb{R}_{\leqslant k}^{n\times m}$ if $\operatorname{rank} W = k$.*

*Proof.* Let $\varphi\colon \mathbb{R}^{n\times k} \to \mathcal{S}_+(\leqslant k, n)$, $R \mapsto RR^\top$ and $\pi\colon \mathbb{R}_{\leqslant k}^{n\times m} \to \mathcal{S}_+(\leqslant k, n)$, $W \mapsto WW^\top$ be the covariance parametrizations. Since the manifold $\mathbb{R}_{\leqslant k}^{n\times m}$ is not smooth, we will focus on the smooth manifold $\mathbb{R}_k^{n\times m}$. Therefore, let $\varphi_*\colon \mathbb{R}_*^{n\times k} \to \mathcal{S}_+(k, n)$ and $\pi_*\colon \mathbb{R}_k^{n\times m} \to \mathcal{S}_+(k, n)$ be the parametrization $\varphi, \pi$ defined on matrices of rank exactly $k$. We know that $\varphi_*$ satisfies the "1 $\Rightarrow$ 1" property, and want to show that $\pi_*$ also satisfies it. The idea of the proof is the pass through given quotient spaces on which the functions are equivalent.

Let $\mathcal{O}_k$ be the set of $k \times k$ orthogonal matrices, with the dimension omitted when inferred from context. Consider the equivalent relation on $\mathbb{R}_k^{n\times m}$ (or $\mathbb{R}_*^{n\times k}$) such that $X_1 \sim X_2 \iff X_1 X_1^\top = X_2 X_2^\top$. From (Massart & Absil, 2020, Proposition 2.1), we know that $X_1 \sim X_2 \iff \exists Q \in \mathcal{O}, X_1 = X_2 Q$. Denote the equivalent class $[X] = \{XQ\colon Q \in \mathcal{O}\} = X\mathcal{O}$.

Let $p\colon \mathbb{R}_k^{n\times m} \to \mathbb{R}_k^{n\times m}/\mathcal{O}_m$, $W \mapsto [W]$ be the quotient map on $\mathbb{R}_k^{n\times m}$, and let

$$\begin{array}{rcl} \Pi \quad : \quad \mathbb{R}_k^{n\times m}/\mathcal{O}_m & \longrightarrow & \mathcal{S}_+(k, n) \\ W\mathcal{O}_m & \longmapsto & WW^\top \end{array} \tag{44}$$

be the map on the quotient space, so that $\pi_* = \Pi \circ p$.

Likewise, let $q\colon \mathbb{R}_*^{n\times k} \to \mathbb{R}_*^{n\times k}/\mathcal{O}_k$, $R \mapsto [R]$ be the quotient map on $\mathbb{R}_*^{n\times k}$, and let

$$\begin{array}{rcl} \Phi \quad : \quad \mathbb{R}_*^{n\times k}/\mathcal{O}_k & \longrightarrow & \mathcal{S}_+(k, n) \\ R\mathcal{O}_k & \longmapsto & RR^\top \end{array} \tag{45}$$

be the map on the quotient space, so that $\varphi_* = \Phi \circ q$.

The map $\Phi$ is an diffeomorphism (Massart & Absil, 2020, Proposition A.7), and therefore satisfies the "1 $\Rightarrow$ 1" property.

For $W \in \mathbb{R}_k^{n\times m}$, we can find $R \in \mathbb{R}_*^{n\times k}$ such that $WW^\top = RR^\top$. It is unique up to an orthogonal matrix. Therefore, let

$$\begin{array}{rcl} \iota \quad : \quad \mathbb{R}_k^{n\times m}/\mathcal{O}_m & \longrightarrow & \mathbb{R}_*^{n\times k}/\mathcal{O}_k \\ [W] & \longmapsto & [R] \end{array}, \tag{46}$$

be the identification map between the quotient spaces.

With the next two lemma, we will be able to finish the proof of Proposition D.15.

**Lemma D.16.** *The map $\iota\colon \mathbb{R}_k^{n\times m}/\mathcal{O}_m \to \mathbb{R}_*^{n\times k}/\mathcal{O}_k$, $[W] \mapsto [R]$ is a diffeomorphism.*

**Lemma D.17.** *The map $\iota \circ p$ is a submersion from $\mathbb{R}_k^{n \times m}$ onto $\mathbb{R}_*^{n \times k}/\mathcal{O}_k$.*

So now, to conclude the proof of Proposition D.15, the map $\pi_*$ can be written $\pi_* = \Pi \circ p = \Phi \circ \iota \circ p$. Since we know that $\Phi$, being a diffeomorphism, satisfies the "1 $\Rightarrow$ 1" property, and since $\iota \circ q$ is a submersion (Lemma D.17), by (Levin et al., 2022, Proposition 2.42 (b)), the map $\Phi \circ \iota \circ p = \pi_*$ satisfies the "1 $\Rightarrow$ 1" property. □

It remains to show Lemmas D.16 and D.17.

*Proof of Lemma D.16.* From (Massart & Absil, 2020, Proposition A.7), the mapping $\Phi \colon \mathbb{R}_*^{n \times k} \to \mathcal{S}_+(k, n)$, $R \mapsto RR^\top$ is a diffeomorphism. Likewise, the mapping $\Pi \colon \mathbb{R}_k^{n \times m} \to \mathcal{S}_+(k, n)$ is also a diffeomorphism, since both $\pi_*$ and $p$ are submersions. Then, since $\Phi([R]) = \Pi([W])$, we have $[R] = \Phi^{-1} \circ \Pi([W]) =: \iota([W])$, and $\iota$ is a diffeomorphism. □

*Proof of Lemma D.17.* The map $p$ is a submersion (Massart & Absil, 2020, Proposition A.5) and the map $\iota$ is a diffeomorphism (Lemma D.16), hence a submersion. Therefore, the composition $\iota \circ p$ is a submersion. □

We are now ready to proof Proposition 4.5.

*Proof of Proposition 4.5.* From (Trager et al., 2020, Proposition 5), we know that a critical point in the parameter space $\overrightarrow{W}$ with rank $\text{rank}(\overrightarrow{W}) = k$ will be a critical point for $L_\tau^1|_{\mathcal{M}_{\leqslant k}}$. Now, from Proposition D.15, a critical point $W^*$ for $L_\tau^1|_{\mathcal{M}_{\leqslant k}}$ with rank $W^* = k$ is such that $\pi(W^*)$ is a critical point for $\tilde{L}|_{\mathcal{S}_+(k,n)}$, and the first part of the proposition is proved. For the second part, assume that $k = \underline{d} = \min_i\{d_i\}$. Then, according to (Trager et al., 2020, Proposition 6), $\overrightarrow{W}$ is a local minimizer for $L^N$ if and only if $W = \mu(\overrightarrow{W})$ is a local minimizer for $L_\tau^1|_{\mathcal{M}_{\leqslant \underline{d}}}$. Since $W$ is a local minimizer for $L_\tau^1|_{\mathcal{M}_{\leqslant \underline{d}}}$, according to Theorem 4.4, there exists $V \in \mathcal{O}_m$ orthogonal, such that if $\Sigma_0 = \Omega\Lambda\Omega^\top$ is a spectral decomposition of $\Sigma_0$, we have $W^* = \Omega_{[\underline{d}]}(\Lambda - \tau I_{\underline{d}})^{1/2} V_{[\underline{d}]}^\top$, so that $\Sigma_\tau^* = W^* W^{*\top} + \tau I_n = \Omega \begin{pmatrix} \Lambda_{[\underline{d}]} & \\ & \tau \end{pmatrix} \Omega^\top$ is also a minimizer of $\tilde{L}_\tau|_{\mathcal{S}_+(\underline{d},n)}$. □

# E  PROOFS OF SECTION 5

## E.1  BOUNDS ON THE HESSIAN

In this section, we provide bounds on the Hessian of the perturbative loss $L_\tau^1$. We first express the loss as a function of the covariance matrix, in which case the Hessian is known (Kroshnin et al., 2021). Then, a simple chain rule for the differential allows to express the Hessian in the case the loss is a function of the end-to-end matrix $W$.

**Lemma E.1** (Second-order differential of $\tilde{L}_\tau$, Kroshnin et al. 2021, Lemma A.2). *Let $W \in \mathbb{R}^{n \times m}$ and let $\tau > 0$. Define $\Sigma_\tau = WW^\top + \tau I_n$ to be the regularized covariance matrix. Given that $\Sigma_\tau \succ 0$, the loss 10 is twice continuously differentiable for any $W$. Let $\Gamma Q \Gamma^\top = \Sigma_0^{1/2} \Sigma_\tau \Sigma_0^{1/2}$ be a spectral decomposition of $\Sigma_0^{1/2} \Sigma_\tau \Sigma_0^{1/2}$, with $Q = \text{diag}(q_1, \ldots, q_n)$. For $Y \in \mathbb{R}^{n \times n}$, define $\Delta(Y) \in \mathbb{R}^{n \times n}$ to be the matrix with element $\Delta(Y)_{ij} = \left(\frac{(\Gamma^\top Y \Gamma)_{ij}}{\sqrt{q_i} + \sqrt{q_j}}\right)$. Let $\tilde{H}$ be the linear operator defined as*

$$\tilde{H}(Y) = \Sigma_0^{1/2} \Gamma Q^{-1/2} \Delta(Y) Q^{-1/2} \Gamma^\top \Sigma_0^{1/2}. \tag{47}$$

*Then, the second order differential of $\tilde{L}_\tau$ is given by*

$$\forall (X, Y) \in (\mathbb{R}^{n \times n})^2, \quad \mathrm{d}^2 \tilde{L}_\tau(\Sigma_\tau)[X, Y] = \langle X, \tilde{H}(Y) \rangle. \tag{48}$$

*Proof.* We begin by stating the first-order differential for the loss $\tilde{L}$ evaluated on the PD matrix $\Sigma_\tau$. This is given in lemma D.10

$$
\begin{aligned}
\mathrm{d}\tilde{L}(\Sigma_\tau)[X] &= \mathrm{tr}(X - \Sigma_0^{1/2}(\Sigma_0^{1/2}\Sigma_\tau\Sigma_0^{1/2})^{-1/2}\Sigma_0^{1/2}X) \\
&= \langle I - \Sigma_0^{1/2}(\Sigma_0^{1/2}\Sigma_\tau\Sigma_0^{1/2})^{-1/2}\Sigma_0^{1/2}, X\rangle.
\end{aligned}
$$

Let $\mathrm{GL}(n) = \{A \in \mathbb{R}^{n\times n} \mid \det A \neq 0\}$, and let $f\colon \mathrm{GL}(n) \ni F \mapsto F^{-1}$; then $f$ is differentiable with differential $\mathrm{d}f(F)[X] = -F^{-1}XF^{-1}$ (Magnus & Neudecker, 2019, Theorem 8.3). Let $g\colon \mathcal{S}_{++}^n \ni A \mapsto A^{1/2}$ be the matrix square root. The function $g$ is differentiable on $\mathcal{S}_{++}^n$, and its differential can be computed as follows (Kroshnin et al., 2021, Lemma A.1). Let $A \in \mathcal{S}_{++}^n$, and let $\Gamma Q \Gamma^\top$ be its spectral decomposition, with $Q = \mathrm{diag}\,(q_i)_{i=1}^n$. For $X \in \mathcal{S}^n$, define $\Delta(X) \in \mathbb{R}^{n\times n}$ to be the matrix with elements $\Delta(X)_{ij} = \frac{(\Gamma X \Gamma^\top)_{ij}}{\sqrt{q_i}+\sqrt{q_j}}$. Then, the differential of $g$ at $A$ in the direction $X$ is $\mathrm{d}g(A)[X] = \Gamma\Delta(X)\Gamma^\top$.

Therefore, the chain rule on the differentials gives

$$
\mathrm{d}(f\circ g)(A)[X] = \mathrm{d}f(g(A))[\mathrm{d}g(A)[X]] = -A^{-1/2}\,\mathrm{d}g(A)[X]A^{-1/2} = -A^{-1/2}\Gamma\Delta(X)\Gamma^\top A^{-1/2},
$$

and, with $A = \Sigma_0^{1/2}\Sigma_\tau\Sigma_0^{1/2}$,

$$
\begin{aligned}
\mathrm{d}^2\tilde{L}(\Sigma_\tau)[X,Y] &= \mathrm{d}(\Sigma_\tau \mapsto \mathrm{d}\tilde{L}(\Sigma_\tau)[X])[Y] \\
&= \mathrm{d}(\mathrm{tr}(X - (\Sigma_0^{1/2}(\Sigma_0^{1/2}\Sigma_\tau\Sigma_0^{1/2})^{-1/2}\Sigma_0^{1/2}X)))[Y] \\
&= -\mathrm{tr}(\Sigma_0^{1/2}(\mathrm{d}(\Sigma_0^{1/2}\Sigma_\tau\Sigma_0^{1/2})^{-1/2}[Y])\Sigma_0^{1/2}X) \\
&= -\mathrm{tr}(\Sigma_0^{1/2}(-A^{-1/2}\Gamma\Delta(Y)\Gamma^\top A^{-1/2})\Sigma_0^{1/2}X) \\
&= \mathrm{tr}(\Sigma_0^{1/2}\Gamma Q^{-1/2}\Delta(Y)Q^{-1/2}\Gamma^\top\Sigma_0^{1/2}X) \\
&= \langle X, \tilde{H}(Y)\rangle
\end{aligned}
$$

with $\tilde{H}(Y) = \Sigma_0^{1/2}\Gamma Q^{-1/2}\Delta(Y)Q^{-1/2}\Gamma^\top\Sigma_0^{1/2}$. $\qquad\square$

In order to express the Hessian of the loss as a function of the end-to-end matrix $W$, we need the chain rule for the second-order differential. We first recall the chain rule for the second-order differential.

**Lemma E.2** (Chain rule for second-order differential, Magnus & Neudecker 2019, Theorem 6.9)**.** *Let $f\colon R \to S$ and $g\colon S \to T$ be two differentiable functions on open sets, such that $h = g\circ f\colon R \to T$ is always well defined. Then, given two directions $u, v$, the second-order differential of $h$ at $c$ is*

$$
\mathrm{d}^2 h(c)[u,v] = \mathrm{d}^2 g(f(c))\big[\,\mathrm{d}f(c)[u], \mathrm{d}f(c)[v]\big] + \mathrm{d}g(f(c))[\mathrm{d}^2 f(c)[u,v]]. \tag{49}
$$

With this computation rule, we are able to give the second-order differential of $L_\tau^1 = \tilde{L}_\tau \circ \pi$.

**Lemma E.3** (Second-order differential of $L_\tau^1$)**.** *Let $W \in \mathbb{R}^{n\times m}$. For any $U, V \in \mathbb{R}^{n\times m}$, the second order differential of $L_\tau^1$ at $W$ in the directions $U, V$ is*

$$
\mathrm{d}^2 L_\tau^1(W)[U,V] = \langle U, H(V)\rangle, \tag{50}
$$

*where*

$$
H(V) = 2(\tilde{H}(VW^\top + WV^\top)W + (I - \Sigma_0^{1/2}(\Sigma_0^{1/2}\Sigma_\tau\Sigma_0^{1/2})^{-1/2}\Sigma_0^{1/2})V), \tag{51}
$$

*and $\tilde{H}$ is defined as in* (47).

*Proof.* Applying the formula (49) to $L_\tau^1 = \tilde{L}_\tau \circ \pi$ gives, with $\Sigma = \pi(W)$ and $d^2\pi(W)[U, V] = UV^\top + VU^\top$,

$$
\begin{aligned}
d^2 L_\tau^1(W)[U, V] &= d^2\tilde{L}_\tau(\Sigma)[d\pi(W)[U], d\pi(W)[V]] + d\tilde{L}_\tau(\Sigma)[d^2\pi(W)[U, V]] \\
&= \langle UW^\top + WU^\top, \tilde{H}(VW^\top + WV^\top)\rangle + \operatorname{tr}(UV^\top + VU^\top) \\
&\quad - \operatorname{tr} \Sigma_0^{1/2}(\Sigma_0^{1/2}\Sigma_\tau\Sigma_0^{1/2})^{-1}\Sigma_0^{1/2}(UV^\top + VU^\top) \\
&= 2\langle U, \tilde{H}(VW^\top + WV^\top)W + V - \Sigma_0^{1/2}(\Sigma_0^{1/2}\Sigma_\tau\Sigma_0^{1/2})^{-1}\Sigma_0^{1/2}V\rangle \\
&= \langle U, H(V)\rangle.
\end{aligned}
$$

where we used the symmetry of $\Sigma_0^{1/2}(\Sigma_0^{1/2}\Sigma_\tau\Sigma_0^{1/2})^{-1/2}\Sigma_0$ to simplify the expression. $\square$

The maximal eigenvalue of $H$ will then be computed as $\lambda_{\max}(H) = \sup_{U:\|U\|_F=1}\langle U, H(U)\rangle$ in Lemma E.6.

## E.2 Lipschitz-smoothness of $L_\tau^1$

One can use the bounds of Kroshnin et al. (2021, Lemma A.3) to bound the Hessian of the loss.

**Lemma E.4** (Bounds on the second-order differential, Kroshnin et al. 2021, Lemma A.3). *Let $\tilde{H}(X)$ be defined as in (47). The second-order differential of $\tilde{L}_\tau$ respects the following bounds*

$$
\langle X, \tilde{H}(X)\rangle \leqslant \frac{\lambda_{\max}^{1/2}(\Sigma_0^{1/2}\Sigma_\tau\Sigma_0^{1/2})}{2}\|\Sigma_\tau^{-1/2}X\Sigma_\tau^{-1/2}\|_F^2, \tag{52a}
$$

$$
\langle X, \tilde{H}(X)\rangle \geqslant \frac{\lambda_{\min}^{1/2}(\Sigma_0^{1/2}\Sigma_\tau\Sigma_0^{1/2})}{2}\|\Sigma_\tau^{-1/2}X\Sigma_\tau^{-1/2}\|_F^2. \tag{52b}
$$

*Those in turn bound the extremal eigenvalues of the Hessian, defined as $\lambda_{\max}(\tilde{H}) = \sup_{X\neq 0}\frac{\langle X, \tilde{H}(X)\rangle}{\|X\|_F}$ and $\lambda_{\min}(\tilde{H}) = \inf_{X\neq 0}\frac{\langle X, \tilde{H}(X)\rangle}{\|X\|_F}$.*

**Lemma E.5** (Bounds on the Hessian $\tilde{H}$). *Let $\tilde{H}$ be defined as in (47). Then, the extremal eigenvalues of $\tilde{H}$ are bounded as*

$$
\lambda_{\max}(\tilde{H}) \leqslant \frac{\sqrt{C_0\lambda_{\max}(\Sigma_0)}}{2\tau^2}, \qquad \lambda_{\min}(\tilde{H}) \geqslant \frac{\sqrt{\tau\lambda_{\min}(\Sigma_0)}}{2C_0^2}, \tag{53}
$$

*where $C_0 = 2(\tilde{L}(\Sigma_\tau(0)) + \operatorname{tr}(\Sigma_0))$ is initialization-dependent.*

*In particular, the loss $\tilde{L}_\tau$ is strongly convex, with parameter $K = \frac{\sqrt{\tau\lambda_{\min}(\Sigma_0)}}{2C_0^2}$.*

*Proof.* We first provide the proof for the maximal eigenvalue.

The maximal eigenvalue of the Hessian is defined as

$$
\lambda_{\max}(\tilde{H}) = \sup_{X:\|X\|_F=1}\langle X, \tilde{H}(X)\rangle.
$$

From the upper-bound of $\langle X, \tilde{H}(X)\rangle$ in (52a), one has

$$
\begin{aligned}
\sup_{X:\|X\|_F=1}\langle X, \tilde{H}(X)\rangle &\leqslant \sup_{X:\|X\|_F=1}\frac{\lambda_{\max}^{1/2}(\Sigma_0^{1/2}\Sigma_\tau\Sigma_0^{1/2})}{2}\|\Sigma_\tau^{-1/2}X\Sigma_\tau^{-1/2}\|_F^2 \\
&= \frac{\lambda_{\max}^{1/2}(\Sigma_0^{1/2}\Sigma_\tau\Sigma_0^{1/2})}{2}\sup_{X:\|X\|_F=1}\|\Sigma_\tau^{-1/2}X\Sigma_\tau^{-1/2}\|_F^2 \\
&= \frac{\lambda_{\max}^{1/2}(\Sigma_0^{1/2}\Sigma_\tau\Sigma_0^{1/2})}{2}\sup_{X:\|X\|_F=1}\|\Sigma_\tau^{-1}X\|_F^2 \\
&= \frac{\lambda_{\max}^{1/2}(\Sigma_0^{1/2}\Sigma_\tau\Sigma_0^{1/2})}{2}\lambda_{\max}^2(\Sigma_\tau^{-1}) \\
&\leqslant \frac{\lambda_{\max}^{1/2}(\Sigma_0^{1/2}\Sigma_\tau\Sigma_0^{1/2})}{2\tau^2}.
\end{aligned}
$$

The last inequality comes from the definition of $\Sigma_\tau$; if $\lambda_1 \geqslant \lambda_2 \geqslant \cdots \geqslant \lambda_k > 0$ are the positive eigenvalues of $WW^\top$, then $\Sigma_\tau^{-1} = (WW^\top + \tau I_n)^{-1}$ has eigenvalues $\underbrace{\tau^{-1} = \cdots = \tau^{-1}}_{n-k \text{ times}} >$
$(\lambda_k + \tau)^{-1} \geqslant \cdots \geqslant (\lambda_1 + \tau)^{-1}$.

For any positive definite matrices $A, B \in \mathcal{S}_{++}(n)$ with increasing eigenvalues, and for any $k \in [n]$, we know that
$$\lambda_k(A)\lambda_1(B) \leqslant \lambda_k(AB) = \lambda_k(A^{1/2}BA^{1/2}) \leqslant \lambda_k(A)\lambda_n(B).$$

Therefore, we have the bound $\lambda_{\max}^{1/2}(\Sigma_0^{1/2}\Sigma_\tau\Sigma_0^{1/2}) \leqslant \lambda_{\max}^{1/2}(\Sigma_0)\lambda_{\max}^{1/2}(\Sigma_\tau)$. Moreover, $\lambda_{\max}(\Sigma_\tau) \leqslant \operatorname{tr}\Sigma_\tau$, and from Lemma C.7, we know that $\operatorname{tr}\Sigma_\tau \leqslant 2(\tilde{L}(\Sigma_\tau) - \tilde{L}(\Sigma_0)) =: C_0$. Therefore, we obtain
$$\lambda_{\max}(\tilde{H}) \leqslant \frac{\sqrt{C_0\lambda_{\max}(\Sigma_0)}}{2\tau^2}.$$

The proof for the minimal eigenvalue is similar and follows from the bound (52b). In this case, the term $\lambda_{\min}^{1/2}(\Sigma_0^{1/2}\Sigma_\tau\Sigma_0^{1/2})$ can be lower bounded by $\sqrt{\tau\lambda_{\min}(\Sigma_0)}$. $\qquad\square$

We now turn to the Hessian of $L_\tau^1$.

**Lemma E.6** (Spectral bound of $H$). *Let $H$ be defined as in* (51). *The maximal eigenvalue for the Hessian of $L_\tau^1$ respects the following bound*
$$\lambda_{\max}(H) \leqslant \lambda_{\max}^{1/2}(\Sigma_0^{1/2}\Sigma_\tau\Sigma_0^{1/2})\frac{2C^2}{\tau^2} + 2(1 - \lambda_{\min}(\Sigma_0^{1/2}(\Sigma_0^{1/2}\Sigma_\tau\Sigma_0^{1/2})^{-1/2}\Sigma_0^{1/2})) \qquad (54)$$

*Proof.* From (52a), one has for any $X \in \mathcal{S}^n$,
$$\langle X, \tilde{H}(X)\rangle \leqslant \frac{\lambda_{\max}^{1/2}(\Sigma_0^{1/2}\Sigma_\tau\Sigma_0^{1/2})}{2}\|\Sigma_\tau^{-1/2}X\Sigma_\tau^{-1/2}\|_F^2.$$

Let $U \in \mathbb{R}^{n\times m}$. With $X(U) = UW^\top + WU^\top$, the bound becomes
$$\langle UW^\top + WU^\top, \tilde{H}(X(U))\rangle \leqslant \frac{\lambda_{\max}^{1/2}(\Sigma_0^{1/2}\Sigma_\tau\Sigma_0^{1/2})}{2}\|\Sigma_\tau^{-1/2}X(U)\Sigma_\tau^{-1/2}\|_F^2$$
$$\iff \qquad 2\langle UW^\top, \tilde{H}(X(U))\rangle \leqslant \frac{\lambda_{\max}^{1/2}(\Sigma_0^{1/2}\Sigma_\tau\Sigma_0^{1/2})}{2}\|\Sigma_\tau^{-1/2}X(U)\Sigma_\tau^{-1/2}\|_F^2$$
$$\iff \qquad 2\langle U, \tilde{H}(X(U))W\rangle \leqslant \frac{\lambda_{\max}^{1/2}(\Sigma_0^{1/2}\Sigma_\tau\Sigma_0^{1/2})}{2}\|\Sigma_\tau^{-1/2}X(U)\Sigma_\tau^{-1/2}\|_F^2.$$

Therefore,
$$\langle U, H(U)\rangle = 2\langle U, \tilde{H}(X(U))W + (I - \Sigma_0^{1/2}(\Sigma_0^{1/2}\Sigma_\tau\Sigma_0^{1/2})^{-1/2}\Sigma_0^{1/2})U\rangle$$
$$\leqslant \frac{\lambda_{\max}^{1/2}(\Sigma_0^{1/2}\Sigma_\tau\Sigma_0^{1/2})}{2}\|\Sigma_\tau^{-1/2}X(U)\Sigma_\tau^{-1/2}\|_F^2 + 2\langle U, (I - \Sigma_0^{1/2}(\Sigma_0^{1/2}\Sigma_\tau\Sigma_0^{1/2})^{-1/2}\Sigma_0^{1/2})U\rangle. \tag{55}$$

We proceed by bounding each of the summands.

First consider the term $\|\Sigma_\tau^{-1/2}X(U)\Sigma_\tau^{-1/2}\|_F^2 = \|\Sigma_\tau^{-1}X(U)\|_F^2$. If $U$ is such that $\|U\|_F = 1$, then $\|X(U)\|_F^2 = \|UW^\top + WU^\top\|^2 \leqslant 4\|W\|_F^2$. We know that $\|W\|_F \leqslant C$ for some constant $C$, c.f. (35). Therefore, $\|U\|_F = 1 \implies \|X(U)\| \leqslant 2C$ and
$$\sup_{U:\, \|U\|_F=1}\|\Sigma_\tau^{-1}X(U)\|_F^2 \leqslant \sup_{X:\, \|X\|_F\leqslant 2C}\|\Sigma_\tau^{-1}X(U)\|_F^2$$
$$= \sup_{X:\, \|X\|=1}4C^2\|\Sigma_\tau^{-1}X\|_F^2$$
$$= 4C^2\lambda_{\max}^2(\Sigma_\tau^{-1}) = \frac{4C^2}{\tau^2}.$$

Therefore,

$$\sup_{U:\,\|U\|_F=1} \frac{\lambda_{\max}^{1/2}(\Sigma_0^{1/2}\Sigma_\tau\Sigma_0^{1/2})}{2}\|\Sigma_\tau^{-1/2}X(U)\Sigma_\tau^{-1/2}\|_F^2 \leqslant \lambda_{\max}^{1/2}(\Sigma_0^{1/2}\Sigma_\tau\Sigma_0^{1/2})\frac{2C^2}{\tau^2}.$$

The second summation in (55) can be bounded as

$$\sup_{U:\,\|U\|_F=1} 2\langle U, (I-\Sigma_0^{1/2}(\Sigma_0^{1/2}\Sigma_\tau\Sigma_0^{1/2})^{-1/2}\Sigma_0^{1/2})U\rangle$$

$$= 2\lambda_{\max}(I - \Sigma_0^{1/2}(\Sigma_0^{1/2}\Sigma_\tau\Sigma_0^{1/2})^{-1/2}\Sigma_0^{1/2})$$

$$= 2(1 - \lambda_{\min}(\Sigma_0^{1/2}(\Sigma_0^{1/2}\Sigma_\tau\Sigma_0^{1/2})^{-1/2}\Sigma_0^{1/2})).$$

$\square$

**Lemma E.7** (Lipshitz-smoothness of $L_\tau^1$). *For $\tau > 0$, the loss $W \mapsto L_\tau^1(W)$ is Lipschitz smooth.*

*Proof.* This directly follows from the boundedness of the Hessian showed previously and the convexity of $L_\tau^1$ using Taylor approximation. $\square$

Once the Lipschitz-smoothness of the loss has been proven, one can turn to showing that the rank is preserved under balanced initial conditions.

**Proposition E.8** (Bah et al. 2021, Proposition 4.4). *Let $\mathcal{L}^1\colon \mathbb{R}^{n\times m} \to \mathbb{R}$ be a Lipschitz smooth function (i.e., a differentiable function with Lipschitz gradient). Suppose that $W_1(t), \ldots, W_N(t)$ are solutions of the gradient flow* (GF) *of $L^N$ with balanced initial values $W_j(0)$ and define the product $W(t) = \phi(\theta(t)) = W_N(t)\cdots W_1(t)$. If $W(0)$ is contained in $\mathcal{M}_k$ for some $k \in \mathbb{N}$, then $W(t)$ is contained in $\mathcal{M}_k$ for all $t \geqslant 0$.*

*Proof.* Let $P(t) = W_1(t)^\top W_1(t) = \left(W(t)^\top W(t)\right)^{1/N}$ and $Q(t) = W_N(t)W_N(t)^\top = \left(W(t)W(t)^\top\right)^{1/N}$. The proof follows **if** the gradient flow is locally Lipschitz continuous in $P, Q, W$, so that the curves $P, Q, W$ are uniquely determined by an initial datum $P(0), Q(0), W(0)$. From Equations (GF) and (21),

$$\dot{P} = -W^\top\nabla\mathcal{L}^1(W) - \nabla\mathcal{L}^1(W)^\top W,$$

$$\dot{Q} = -\nabla\mathcal{L}^1(W)W^\top - W\nabla\mathcal{L}^1(W)^\top,$$

$$\dot{W} = -\sum_{j=1}^{N} Q^{N-j}\nabla\mathcal{L}^1(W)P^{j-1}$$

Now, with the assumption of Lipschitz continuity of the flow, a given solution is uniquely determined by the initial data $P_0, Q_0, W_0$, and the proof tools of Bah et al. (2021, Proposition 4.4) can be used here as well. $\square$

**Remark E.9.** The loss $L_\tau^1$ satisfies the conditions of Proposition E.8; therefore, the flow on $L_\tau^1$ remains in the manifold $\mathcal{M}_k$ if $W(t_0) \in \mathcal{M}_k$ for some $t_0$.

### E.3 PROOFS OF GRADIENT FLOW CONVERGENCE

*Proof of Theorem 5.6.* The idea of the proof is to transfer the strong convexity property from $\tilde{L}_\tau$ to the evolution of the parameters. Let us start by the inequality which holds due to strong convexity

$$\tilde{L}(\Sigma_\tau) - \tilde{L}(\Sigma_\tau^*) \leq \frac{1}{2K}\|\nabla\tilde{L}(\Sigma_\tau)\|^2,$$

where $K$ is the constant from Lemma (E.5. Rearranging the terms in the above equation, we have

$$-\|\nabla\tilde{L}(\Sigma_\tau)\|^2 \leq -2K\left(\tilde{L}(\Sigma_\tau) - \tilde{L}(\Sigma_\tau^*)\right). \tag{56}$$

On the covariance space, for the perturbative loss, the gradient flow is written

$$\begin{aligned}
\frac{\mathrm{d}\tilde{L}_\tau(\Sigma)}{\mathrm{d}t} &= \langle\nabla\tilde{L}_\tau(\Sigma), \frac{\mathrm{d}}{\mathrm{d}t}\Sigma(t)\rangle \\
&= \langle\nabla_\Sigma\tilde{L}_\tau(\Sigma), W\frac{\mathrm{d}W}{\mathrm{d}t}^\top + \frac{\mathrm{d}W}{\mathrm{d}t}W^\top\rangle \\
&= 2\langle\nabla_\Sigma\tilde{L}_\tau(\Sigma)W, \frac{\mathrm{d}W}{\mathrm{d}t}\rangle
\end{aligned}$$

The expression of $\frac{\mathrm{d}W}{\mathrm{d}t}$ is given in Lemma C.1.2:

$$\frac{\mathrm{d}W}{\mathrm{d}t} = -\sum_{\ell=1}^N W_{N:j+1}W_{N:j+1}^\top\nabla L^1(W)W_{\ell-1:1}^\top W_{\ell-1:1}.$$

Since $\nabla L^1(W) = 2\nabla\tilde{L}(\Sigma)W$, and from the balancedness assumption we have $W_{N:j+1}W_{N:j+1}^\top = (WW^\top)^{\frac{N-\ell}{N}}$ and $W_{\ell-1:1}^\top W_{\ell-1:1} = (W^\top W)^{\frac{\ell-1}{N}}$, we get

$$\frac{\mathrm{d}\tilde{L}(\Sigma(t))}{\mathrm{d}t} = -4\sum_{\ell=1}^N\langle\nabla\tilde{L}(\Sigma)W, (WW^\top)^{\frac{N-\ell}{N}}\nabla\tilde{L}(\Sigma)W(W^\top W)^{\ell-1}N\rangle$$

Now, let $USV^\top = W$ be a (thin) SVD of $W$, so that $WW^\top = US^2U^\top$ and $W^\top W = VS^2V^\top$. For one layer $\ell \in [N]$, we then have

$$\begin{aligned}
\langle\nabla\tilde{L}(\Sigma)W, (WW^\top)^{\frac{N-\ell}{N}}\nabla\tilde{L}(\Sigma)W(W^\top W)^{\frac{\ell-1}{N}}\rangle &= \mathrm{tr}\left(\nabla\tilde{L}(\Sigma)W(W^\top W)^{\frac{\ell-1}{N}}W^\top\nabla\tilde{L}(\Sigma)(WW^\top)^{\frac{N-\ell}{N}}\right) \\
&= \mathrm{tr}\left(\nabla\tilde{L}(\Sigma)USV^\top VS^{\frac{2(\ell-1)}{N}}V^\top VSU^\top\nabla\tilde{L}(\Sigma)US^{\frac{2(N-\ell)}{N}}U^\top\right) \\
&= \mathrm{tr}\left(U^\top\nabla\tilde{L}(\Sigma)US^{\frac{2(N+\ell-1)}{N}}U^\top\nabla\tilde{L}(\Sigma)US^{\frac{2(N-\ell)}{N}}\right) \\
&= \langle U^\top\nabla\tilde{L}(\Sigma)US^{\frac{2(N+\ell-1)}{N}}, S^{\frac{2(N-\ell)}{N}}U^\top\nabla\tilde{L}(\Sigma)U\rangle
\end{aligned}$$

Let $X = U^\top\nabla\tilde{L}(\Sigma)U$, $R = S^{\frac{2(N+\ell-1)}{N}}$, and $L = S^{\frac{2(N-\ell)}{N}}$. We evaluate $\langle XR, LX\rangle$ for diagonals $R, L$ as

$$\langle XR, LX\rangle = \sum_{i,j}X_{i,j}R_jX_{i,j}L_i = \sum_{i,j}L_iR_jX_{i,\ell}^2$$

Since $L_i = s_i^{\frac{2(N-\ell)}{N}}$ and $R_j = s_j^{\frac{2(N+\ell-1)}{N}}$, and due to the uniform margin deficiency assumption, for all $(i,j) \in [k]^2$, we have $L_i \geqslant c^{\frac{2(N-\ell)}{N}}$ and $R_j \geqslant c^{\frac{2(N+\ell-1)}{N}}$, so that

$$\langle XR, LX\rangle \geqslant c^{\frac{2(2N-1)}{N}}\sum_{i,j}X_{i,j}^2 = c^{\frac{2(2N-1)}{N}}\|X\|_F^2.$$

Since $X = U^\top\nabla\tilde{L}(\Sigma)U^\top$, we have that $\|X\|_F^2 = \|\tilde{L}(\Sigma)\|_F^2$, so that in total

$$\frac{\mathrm{d}\tilde{L}(\Sigma(t))}{\mathrm{d}t} \leqslant -4\sum_{\ell=1}^N c^{\frac{2(2N-1)}{N}}\|\nabla\tilde{L}(\Sigma)\|_F^2 = -4Nc^{\frac{2(2N-1)}{N}}\|\nabla\tilde{L}(\Sigma)\|_F^2. \tag{57}$$

From the strong convexity of $\tilde{L}$ (56), we get the bound

$$\frac{\mathrm{d}\tilde{L}(\Sigma(t))}{\mathrm{d}t} \leqslant -8Nc^{\frac{2(2N-1)}{N}}K(\tilde{L}(\Sigma) - \tilde{L}(\Sigma^*))$$

$$\implies \frac{1}{\tilde{L}(\Sigma(t) - \tilde{L}(\Sigma^*))}\frac{\mathrm{d}(\tilde{L}(\Sigma(t)) - \tilde{L}(\Sigma^*))}{\mathrm{d}t} \leqslant -8Nc^{\frac{2(2N-1)}{N}}K$$

Now, by integrating both sides from $0$ to $t$,

$$\ln\left(\frac{\tilde{L}(\Sigma_\tau(t)) - \tilde{L}(\Sigma_\tau^*)}{\tilde{L}(\Sigma_\tau(0)) - \tilde{L}(\Sigma_\tau^*)}\right) \le -8Nc^{\frac{2(2N-1)}{N}}Kt. \tag{58}$$

Let $\Delta_0^* = \Sigma_\tau(0) - \Sigma_\tau^*$ which is the distance to optimality from the initialization. Finally we get the desired exponential rate

$$\tilde{L}(\Sigma_\tau(t)) - \tilde{L}(\Sigma_\tau^*) \le e^{-8Nc^{\frac{2(2N-1)}{N}}Kt}\Delta_0^*,$$

which concludes the proof. $\qquad\square$

### E.4  PROOF OF GRADIENT DESCENT CONVERGENCE

We start by proving Lemma 5.3 so that with the uniform margin deficiency assumption on the initial weights, $WW^\top$ does not degenerate along the gradient descent training algorithms.

*Proof of Lemma 5.3.* Let $\bar{U}(k) := \arg\min_{U \in U(n)}\|\sqrt{W(k)W(k)^\top} - \Sigma_0^{1/2}U\|_F^2$ for each $k$, then as $L^1(W(k)) \le L^1(W(0))$ for all $k \ge 0$, we have

$$
\begin{aligned}
\sigma_{\min}\left(\sqrt{W(k)W(k)^\top}\right) &= \sigma_{\min}\left(\sqrt{W(k)W(k)^\top} - \Sigma_0^{1/2}\bar{U}(k) + \Sigma_0^{1/2}\bar{U}(k)\right) \\
&\ge \sigma_{\min}\left(\Sigma_0^{1/2}\bar{U}(k)\right) - \sigma_{\max}\left(\sqrt{W(k)W(k)^\top} - \Sigma_0^{1/2}\bar{U}(k)\right) \\
&\ge \sigma_{\min}\left(\Sigma_0^{1/2}\bar{U}(k)\right) - \|\sqrt{W(k)W(k)^\top} - \Sigma_0^{1/2}\bar{U}(k)\|_F \\
&= \sigma_{\min}\left(\Sigma_0^{1/2}\bar{U}(k)\right) - \sqrt{L^1(W(k))} \\
&\ge \sigma_{\min}\left(\Sigma_0^{1/2}\bar{U}(k)\right) - \sqrt{L^1(W(0))} \\
&= \sigma_{\min}\left(\Sigma_0^{1/2}\bar{U}(k)\right) - \|\sqrt{W(0)W(0)^\top} - \Sigma_0^{1/2}\bar{U}(0)\|_F \\
&\ge \sigma_{\min}\left(\Sigma_0^{1/2}\bar{U}(k)\right) - \sigma_{\min}\left(\Sigma_0^{1/2}\right) + c = c.
\end{aligned}
\tag{59}
$$

The cancellation in the last equality works due to the fact that the multiplication with an arbitrary unitary matrix does not change singular values. $\qquad\square$

Now we are ready to prove the finite step size gradient descent convergence of the BW loss. We consider the perfect balancedness of initial values $W_i(0), 1 \le i \le N$ in the remaining proof. The approximation balancedness case can also be carried out but require more complicated auxiliary estimates. We leave the approximate balancedness assumption as a future direction.

*Proof of Theorem 5.7.* Let us start from the gradient descent of the loss with respect to each layer

$$
\begin{aligned}
W_j(k+1) &= W_j(k) - \eta\nabla_{W_j}L^N(W_1(k), \cdots W_n(k)) \\
&= W_j(k) - \eta W_{j+1:N}(k)^\top \nabla_W L^1(W(k))W_{1:j-1}(k)^\top, \quad 1 \le j \le N,
\end{aligned}
\tag{60}
$$

with the boundary conditions $W_{1:0}(k) = I_{d_0}$ and $W_{N+1:N}(k) = I_{d_N}$ for all $k \ge 0$.

With the notations $\overrightarrow{W} = (W_1, W_2, \cdots, W_N)$ and

$$\nabla L^N(\overrightarrow{W}) = \begin{pmatrix} \nabla_{W_1}L^N(\overrightarrow{W}) \\ \vdots \\ \nabla_{W_N}L^N(\overrightarrow{W}) \end{pmatrix},$$

we consider to write the Taylor expansion in the form

$$L^N(\overrightarrow{W}(k+1)) = L^N(\overrightarrow{W}(k)) + \left\langle \nabla L^N(\overrightarrow{W}(k)), \overrightarrow{W}(k+1) - \overrightarrow{W}(k) \right\rangle$$
$$+ \frac{1}{2}\left\langle (\overrightarrow{W}(k+1) - \overrightarrow{W}(k))^\top \nabla^2 L^N(\overrightarrow{A_\xi}(k)), \overrightarrow{W}(k+1) - \overrightarrow{W}(k) \right\rangle, \tag{61}$$

with

$$\overrightarrow{A_\xi}(k) = \overrightarrow{W}(k) + \xi(\overrightarrow{W}(k+1) - \overrightarrow{W}(k)), \quad \text{for some } \xi \in [0,1].$$

Recall the relation (21), for $1 \le j \le N$,

$$\nabla_{W_j} L^N(W_1, \ldots, W_N) = W_{j+1}^\top \cdots W_N^\top \nabla_W L^1(W) W_1^\top \cdots W_{j-1}^\top,$$

then the first order term in (61), under (60), can be written as

$$\left\langle \nabla L^N(\overrightarrow{W}(k)), \overrightarrow{W}(k+1) - \overrightarrow{W}(k) \right\rangle = \sum_{j=1}^N \nabla_{W_j} L^N(\overrightarrow{W}(k))^\top (W_j(k+1) - W_j(k))$$

$$= -\eta \sum_{j=1}^N W_{j-1} \cdots W_1 \nabla_W L^1(W(k))^\top W_N \cdots W_{j+1} W_{j+1}^\top \cdots W_N^\top \nabla_W L^1(W(k)) W_1^\top \cdots W_{j-1}^\top$$

$$= -\eta \sum_{j=1}^N W_{j-1} \cdots W_1 \nabla_W L^1(W(k))^\top (W_N W_N^\top)^{N-j} \nabla_W L^1(W(k)) W_1^\top \cdots W_{j-1}^\top$$

$$\le -\eta \sum_{j=1}^N \sigma_{\min}\left((W_N W_N^\top)^{N-j}\right) \sigma_{\min}\left((W_1^\top W_1)^{j-1}\right) \|\nabla_W L^1(W(k))\|_F^2$$

$$\tag{62}$$

Throughout the computation above, $W_i = W_i(k)$ for all $1 \le i \le N$. Moreover, we use the balancedness $W_j W_j^\top = W_{j+1}^\top W_{j+1}$ for all $1 \le i \le N-1$ so that, in the symmetric structure above,

$$W_N \cdots W_{j+1} W_{j+1}^\top \cdots W_N^\top = (W_N W_N^\top)^{N-j}$$
$$W_1^\top \cdots W_{j-1}^\top W_{j-1} \cdots W_1 = (W_1^\top W_1)^{j-1}.$$

Therefore, thanks to Lemma 5.3,

$$\sigma_{\min}\left((W_N(k)W_N(k)^\top)^N\right) = \sigma_{\min}\left((W_1(k)^\top W_1(k))^N\right) = \sigma_{\min}\left(W(k)W(k)^\top\right) \ge c^2,$$

from which we get

$$\left\langle \nabla L^N(\overrightarrow{W}(k)), \overrightarrow{W}(k+1) - \overrightarrow{W}(k) \right\rangle \le -\eta N c^{\frac{2(N-1)}{N}} \|\nabla_W L^1(W(k))\|_F^2. \tag{63}$$

Let us mention that Arora et al. (2018, Theorem 1 and Claim 1) provide rigorous derivations about the equalities above. The second order term in (61) is more complicated to handle, as we have

$$\nabla^2 L^N(\overrightarrow{W})[\overrightarrow{X}, \overrightarrow{X}] = \sum_{j=1}^N \left\langle X_j, \frac{\partial^2 L^N(\overrightarrow{W})}{\partial W_j^2} X_j \right\rangle + \sum_{j=1}^N \sum_{i=1, i\neq j}^N \left\langle X_j, \frac{\partial^2 L^N(\overrightarrow{W})}{\partial W_i \partial W_j} X_i \right\rangle. \tag{64}$$

Thanks to Corollary C.4, we have expressions of $\frac{\partial^2 L^N(\overrightarrow{W})}{\partial W_j^2}$ and $\frac{\partial^2 L^N(\overrightarrow{W})}{\partial W_i \partial W_j}$ ready.

Note that we have the boundedness (C.9)

$$\|W\|_F \le \sqrt{2\left(L^1(W) + \|\Sigma_0^{1/2}\|_F^2\right)} \le \sqrt{2\left(L^1(W(0)) + \|\Sigma_0^{1/2}\|_F^2\right)} =: M, \tag{65}$$

and it is straightforward to see that

$$\|W_i\|_F^2 \le \|W\|_F^{2/N}, \quad \text{for all } 1 \le i \le N. \tag{66}$$

Moreover, for all $1 \le i \le N$, since $\xi \in [0,1]$,

$$A_{\xi,i}(k) = W_i(k) + \xi(W_i(k+1) - W_i(k)) = (1-\xi)W_i(k) + \xi W_i(k+1),$$

we then have the uniform upper bound for all $k \geq 0$,

$$\|A_{\xi,i}(k)\|_F \leq (1-\xi)\|W_i(k)\|_F + \xi\|W_i(k+1)\|_F \leq M^{1/N}. \tag{67}$$

Using $A_{\xi,i}(k) = W_i(k) - \xi\eta W_{j+1:N}(k)^\top \nabla_W L^1(W(k)) W_{1:j-1}(k)^\top$, we can obtain a lower bound in terms of the minimum singular value,

$$
\begin{aligned}
&\sigma_{\min}\left(A_{\xi,i}(k)A_{\xi,i}(k)^\top\right) \\
&\quad \geq \sigma_{\min}\left(W_i(k)W_i(k)^\top\right) - 2\xi\eta\|W_i(k)\|_F\|W_{j+1:N}(k)\|_F\|W_{1:j-1}(k)\|_F\|\nabla_W L^1(W(k))\|_F \\
&\quad \geq c^2 - 4\eta M\sqrt{L^1(W(k))} \geq c^2 - 4\eta M\sqrt{L^1(W(0))},
\end{aligned}
\tag{68}
$$

where we utilize (25), (66) and (65), as well as non-increment of $L^1(W)$ throughout the training. We denote $X_j = -\eta W_{j+1:N}(k)^\top \nabla_W L^1(W(k)) W_{1:j-1}(k)^\top$. We may choose

$$\eta \leq \frac{c^2}{8M\sqrt{L^1(W(0))}},$$

so that for all $k \geq 0$,

$$\sigma_{\min}\left(A_{\xi,i}(k)A_{\xi,i}(k)^\top\right) \geq \frac{c^2}{2}, \quad \text{and} \quad \sigma_{\min}\left(A_\xi(k)A_\xi(k)^\top\right) \geq \frac{c^{2N}}{2^N}. \tag{69}$$

Then combining all estimates above, we have

$$
\begin{aligned}
&\left|\left\langle(\overrightarrow{W}(k+1) - \overrightarrow{W}(k))^\top \nabla^2 L^N(\overrightarrow{A_\xi}(k)), \overrightarrow{W}(k+1) - \overrightarrow{W}(k)\right\rangle\right| \\
&\quad \leq \sum_{j=1}^N \left|\left\langle X_j, \frac{\partial^2 L^N(\overrightarrow{A_\xi}(k))}{\partial W_j^2}X_j\right\rangle\right| + \sum_{j=1}^N \sum_{i=1,i\neq j}^N \left|\left\langle X_j, \frac{\partial^2 L^N(\overrightarrow{A_\xi}(k))}{\partial W_i \partial W_j}X_i\right\rangle\right| \\
&\quad \leq \sum_{j=1}^N \frac{\lambda_{\max}^{1/2}(\Sigma_0^{1/2}A_\xi(k)A_\xi(k)^\top\Sigma_0^{1/2})}{2}\|X_j(A_\xi(k)A_\xi(k)^\top)^{-1}\|_F^2 M^{2(N-1)/N} \\
&\qquad + \sum_{j=1}^N \sum_{i=1,i\neq j}^N M^{(N-2)/N}\|X_i\|_F\|X_j\|_F\|\nabla_W L^1(A_\xi(k))\|_F \\
&\qquad + \sum_{j=1}^N \sum_{i=1,i\neq j}^N \Big(\frac{\lambda_{\max}^{1/2}(\Sigma_0^{1/2}A_\xi(k)A_\xi(k)^\top\Sigma_0^{1/2})}{2}\|X_j(A_\xi(k)A_\xi(k)^\top)^{-1}\|_F \\
&\qquad\qquad \times \|X_i(A_\xi(k)A_\xi(k)^\top)^{-1}\|_F M^{2(N-1)/N}\Big),
\end{aligned}
$$

by using (67), (E.4) and applying the Cauchy-Schwarz inequality for the last term. Notice that $\|X_i\|_F \leq 2\eta M^{(N-1)/N}\|\nabla_W L^1(W(k))\|_F$. Now combining all the bounds we obtained previously, in addition to (69), we get that

$$
\begin{aligned}
&\left|\left\langle(\overrightarrow{W}(k+1) - \overrightarrow{W}(k))^\top \nabla^2 L^N(\overrightarrow{A_\xi}(k)), \overrightarrow{W}(k+1) - \overrightarrow{W}(k)\right\rangle\right| \\
&\quad \leq 2\eta^2 N^2\|A_\xi(k)\|_F \lambda_{\max}^{1/2}(\Sigma_0)\frac{M^{4(N-1)/N}}{\sigma_{\min}(A_\xi(k)A_\xi(k)^\top)}\|\nabla_W L^1(W(k))\|_F^2 \\
&\qquad + 4\eta^2 N(N-1)M^{(3N-4)/N}\|\nabla_W L^1(A_\xi(k))\|_F\|\nabla_W L^1(W(k))\|_F^2.
\end{aligned}
\tag{70}
$$

Moreover, we can use (7), (25) again to get

$$
\begin{aligned}
\|\nabla_W L^1(A_\xi(k))\|_F &= 2\sqrt{L^1(A_\xi(k))} \leq 2\|(A_\xi(k)A_\xi(k)^\top)^{1/2} - \Sigma_0^{1/2}U\|_F \\
&\leq 2\|(A_\xi(k)A_\xi(k)^\top)^{1/2}\|_F + 2\|\Sigma_0^{1/2}\|_F \leq 2M^{1/N} + 2\|\Sigma_0^{1/2}\|_F.
\end{aligned}
$$

Thus, we conclude the estimate for the second order term by

$$\left| \left\langle (\overrightarrow{W}(k+1) - \overrightarrow{W}(k))^\top \nabla^2 L^N(\overrightarrow{A_\xi}(k)), \overrightarrow{W}(k+1) - \overrightarrow{W}(k) \right\rangle \right|$$
$$\leq \eta^2 \|\nabla_W L^1(W(k))\|_F^2 \left( \frac{2^{N+1}}{c^{2N}} N^2 M^{(4N-3)/N} \lambda_{\max}^{1/2}(\Sigma_0) \right.$$
$$\left. + 8N(N-1)M^{(3N-4)/N}\left( M^{1/N} + \|\Sigma_0^{1/2}\|_F \right) \right).$$

Let us denote the constant

$$\Delta := \frac{2^{N+1}}{c^{2N}} N^2 M^{(4N-3)/N} \lambda_{\max}^{1/2}(\Sigma_0) + 8N(N-1)M^{(3N-4)/N}\left( M^{1/N} + \|\Sigma_0^{1/2}\|_F \right),$$

then, with we can write the iteration as

$$L^N(\overrightarrow{W}(k+1)) = \left( 1 - 4Nc^{\frac{2(N-1)}{N}}\eta + 4\Delta\eta^2 \right) L^N(\overrightarrow{W}(k)).$$

If we choose

$$\eta \leq \frac{Nc^{\frac{2(N-1)}{N}}}{2\Delta},$$

then we have

$$L^N(\overrightarrow{W}(k)) \leq \left( 1 - 2\eta Nc^{\frac{2(N-1)}{N}} \right)^k L^N(\overrightarrow{W}(0)).$$

For $\eta$ being sufficiently small, we have $1 - 2\eta Nc^{\frac{2(N-1)}{N}} \leq \exp\left( -2\eta Nc^{\frac{2(N-1)}{N}} \right)$. Thus, to achieve $\epsilon$-error for the loss,

$$k \geq \frac{1}{2\eta Nc^{\frac{2(N-1)}{N}}} \log\left( \frac{L^1(W(0))}{\epsilon} \right).$$

$\square$

