# OpenReview forum: "Convergence of Generative Deep Linear Networks Trained with Bures-Wasserstein Loss"
_ICLR.cc/2023/Conference — Submitted to ICLR 2023_

### Official Review · Reviewer_5WMe · 2022-10-17

**Confidence:** 5
**Correctness:** 3
**Technical Novelty And Significance:** 2
**Empirical Novelty And Significance:** Not applicable
**Recommendation:** 5

**Clarity, Quality, Novelty And Reproducibility:**

The paper is mostly clear. One assumption is not justified. The results are not very significant compared to prior work.

**Strength And Weaknesses:**

Strength: Although this paper considers a deep linear network, the problem setting is relatively new as previous work primarily studies square loss or logistic/exponential loss. The writings are mostly clear.

Weakness: I have several concerns about the quality and significance of the results. Specifically:
1. The discussion of critical points of BW loss and its smooth version is less relevant to the rest of the paper because the results are shown for the space of $W$, the end-to-end matrix. If one is concerned about the landscape of deep linear networks, he should study the critical points of BW loss in the weight space $W_1, \cdots, W_N$, as those in [1]. If this paper is mainly about convergence, I don't know why Section 4 is presented.
2. The convergence requires a uniform deficiency margin at initialization. Although this is very related to the condition proposed in [2], this UDM assumption is much more restrictive than the one in [2]. In fact, it is highly nontrivial to see whether this UDM assumption can be satisfied, but the UDM assumption is essential to the convergence proof in the paper. The author needs to justify the UDM assumption formally by showing there exists an initial $W(0)$ that satisfies UDM.
3. Regarding the contribution, the new thing is the BW loss. Except for the loss, the convergence analysis relies on a balanced assumption together with a deficiency margin (with minor modification), which is mostly based on the analysis in [2]. Moreover, even the balanced assumption is restrictive and recent work has shown convergence of GF under non-balanced initialization [3,4]. Unless there is a significant difference when analyzing the convergence under the BW loss, the contribution of this paper is minor, in my opinion.

Minor comments:
1. It's unclear from Theorem 5.6 whether the loss has $\mathcal{O}(t^{-1})$ convergence or not because $||\Delta_t^* ||$ depends on time $t$

Reference:

[1] Kenji Kawaguchi. Deep learning without poor local minima, Advances in Neural Information Processing Systems, volume 29. Curran Associates, Inc., 2016.

[2] Sanjeev Arora, Nadav Cohen, Noah Golowich, and Wei Hu. A convergence analysis of gradient descent for deep linear neural networks. ICLR, 2018

[3] Chulhee Yun, Shankar Krishnan, and Hossein Mobahi. A unifying view on implicit bias in training linear neural networks. ICLR, 2020.

[4] Hancheng Min, Salma Tarmoun, René Vidal, and Enrique Mallada. On the explicit role of initialization on the convergence and implicit bias of overparametrized linear networks. ICML, 2021

**Summary Of The Paper:**

This paper shows the convergence of specific first-order gradient-based algorithms on deep linear networks with Bures-Wasserstein loss, under the assumption that the initial weights are balanced. The author studies both the original non-smooth Bures-Wasserstein loss and a smooth perturbative one. They characterize the critical points of those two losses over the rank-constrained matrices and they show the convergence of GF on smooth perturbative BW loss, and GD on BW loss, under the assumption that the initial weights are balanced, and has a uniform deficiency margin.



**Summary Of The Review:**

One assumption is not justified in the paper;

The results are not very significant compared to prior work.

---

> ### Author Response · Authors · 2022-11-18
> **Response to Reviewer 5WMe**
>
> We thank the reviewer for their feedback and questions.
>
> 1.  In response to
>
>     > The discussion of critical points of BW loss and its smooth
>     > version ...
>
>    * This is a good point. In the initial manuscript we did not
>     sufficiently emphasize the connection between the discussion of
>     critical points and the convergence analysis. We have improved the
>     presentation in the revision to make the connection clearer and have
>     included new results to this end.
>
>    * The characterization of the critical points for the smooth
>     Bures-Wasserstein loss is necessary in order to study the possible
>     solutions provided by gradient optimization algorithms.
>     Specifically, Therorem 4.4 describes the nature of the critical
>     points in the function space. The reviewer is right, a connection to
>     the parameter space was missing. We have now added a discussion
>     about how a critical point in the parameter space corresponds to a
>     critical point in the function space. This result is based on the
>     work of [TKB], which we now discuss in
>     Proposition 4.5. This allows us to draw conclusions about
>     the nature of the functions that we obtain as the result of
>     optimizing the parameters by a gradient descent algorithm, which are
>     stationary points of the optimization problem in parameter space. We
>     have added a discussion on this fact after Proposition 4.5.
>
>    * For the original non-smooth loss, we can observe that the critical
>     points in function space are critical points in parameter space and
>     hence we have a partial description of the possible solutions in
>     function space. We recognize however that we are still missing a
>     complete description of the correspondence between critical points
>     in parameter space and critical points in function space. If one
>     uses the original loss separately over each $\mathcal{M}_k$ (over
>     each of these sets it is differentiable), the convergence in
>     function space would be to one of the minimizers computed in Theorem
>     4.2. This is matched experimentally, but the theory establishing
>     these links in full detail will necessitate additional work. We are
>     running computer experiments and will add them if time permits.
>
> 2.  In response to
>
>     > The convergence requires a uniform deficiency margin at
>     > initialization ...
>
>     In the initial submission we had stated an incorrect condition in
>     Remark 5.4. We apologize for the confusion created. We have removed
>     the Remark 5.4, which, we agree, was misleading. In the updated
>     manuscript we have relaxed the previous formulation of the uniform
>     deficiency margin. Please find now the new version in Definition
>     5.2. Now the bound for the modified uniform deficiency margin is
>     satisfied in case of initialization close to the target. Also, we
>     added the remark that the uniform margin deficiency is a sufficient
>     condition for the gradient flow convergence analysis but not a
>     necessary condition. The necessary condition is stated in the Lemma
>     5.3 on the boundedness of the eigenvalues of $WW^\top$, which holds
>     for instance when the weights are balanced and initialized with
>     full-rank.
>
>
>
> Reference
>
> [TKB] M. Trager, K. Kohn, and J. Bruna. Pure and spurious critical points: a geometric study of linear networks.  ICML, 2020.

---

> > ### Author Response · Authors · 2022-11-18
> > **Response to Reviewer 5WMe (Part 2/2)**
> >
> > 3.  In response to
> >
> >     > Regarding the contribution, the new thing...
> >
> >    * We point out that a main contribution of our work is to characterize
> >     the critical points of the Bures-Wasserstein loss on the space of
> >     functions parametrized by a deep linear network. This important
> >     aspect was omitted in the Reviewer's assessment of novelty.
> >
> >    * We have now also added a new result (Proposition 4.5) connecting the
> >     critical points in function space and the stationary points in
> >     parameter space found by gradient flow (in the maximal-rank case).
> >     The convergence analysis is a second main contribution from our work
> >     that complements the geometrical analysis of the optimization
> >     landscape. To further develop the connection to parameter space, we
> >     have also added the observation that for the smooth
> >     Bures-Wasserstein loss, the critical points in parameter space
> >     correspond to critical points in function space (over the space of
> >     realizable linear functions) and in covariance space with the
> >     corresponding rank constraints (see discussion in Section 4.2).
> >
> >    * We would like to point out that we have two types of convergence results: results for gradient descent and results for gradient flow.
> >     While the proofs of convergence for gradient descent are indeed
> >     based on previous works, those works focused on the MSE loss.
> >     Importantly, for the BW loss that we consider, the parametrization
> >     of the covariance matrix makes the loss non-convex on the space of
> >     linear functions parametrized by the neural network, so that in our
> >     case the analysis is more involved.
> >    * Moreover, our proof strategy for gradient flow convergence requires
> >     a bound on the Hessian of the loss over the space of covariance
> >     matrices in order to conclude on its strong convexity. This analysis
> >     is new and it required significant efforts (see Appendix D). Kindly
> >     note that the work of Arora et al. (2019) does not study gradient
> >     flow. In fact one of the reasons we consider the BW loss an
> >     interesting case are the challenges that are involved in studying
> >     the convergence of gradient flow for this loss.
> >
> >    * To adequately advertise the analysis we are conducting we have
> >     improved the presentation and included a new relevant result Lemma
> >     5.5. We are not aware of any other work studying exponential rates
> >     of convergence in a comparable setting as ours. We think that the
> >     techniques we have developed can be relevant for any loss that is
> >     convex in some space in the sense that it can be factored as
> >     $\ell\circ \pi \circ \mu(\theta)$ with a convex $\ell$ even if it is
> >     not convex in function space.
> >
> >    * We thank the reviewer for the references, some of which were already
> >     cited in our manuscript.
> >
> >      -   The paper [YKM] studies a deep linear neural network under a tensor framework, focusing on the
> >         implicit bias that the gradient flow solution will benefit from.
> >         The rate of convergence is missing from their analysis. They use
> >         a type of relaxation of the balancedness assumption on the
> >         initial weight matrices. As we mentioned in Remark 2.2, we
> >         expect that our analysis goes through for the case of delta
> >         balancedness. An difference however is that in the referenced
> >         work the condition does not require boundedness. We think this
> >         is an interesting possibility and have added a comment in Remark
> >         2.2. We focus on the investigation of the BW loss but ultimately
> >         the extensions to cover such initializations would be
> >         interesting too.
> >
> >      -   In the paper [MTVM], the authors introduced an imbalance assumption for a linear neural network
> >         with one hidden layer. In their work the analysis requires that
> >         the level of imbalance is controlled in so that it will not
> >         increase too much from its initial value. This is ensured as
> >         long as the error is decreasing exponential fast. In the same
> >         paper the authors consider an initialization which is similar to
> >         the NTK regime. The biggest difference to our focus is that they
> >         consider shallow networks whereas we are considering deep
> >         networks.
> >
> >   *  In response to your minor comment, indeed the term in $\Delta_t^*$
> >     was not precisely taken care of in Theorem 5.6. We have improved the
> >     proof and statement for the convergence rate so that now there is no
> >     dependency on the $\Delta_t^*$ term.
> >
> >   *  Thanks once more for your valuable feedback. We have made important
> >     efforts to address all your concerns and believe that the paper has
> >     much improved as a consequence of this.
> >
> > References
> >
> > [YKM] C. Yun, S. Krishnan and H. Mobahi. A unifying view on implicit bias in training linear neural networks. ICML 2021
> >
> > [MTVM] H. Min, S. Tarmoun, R. Vidal and E. Mallada. On the Explicit Role of Initialization on the Convergence and Implicit Bias of Overparametrized Linear Networks. ICML 2021

---

> ### Author Response · Authors · 2022-11-24
> **Comments in regard to our response and updated manuscript**
>
> We made systematic efforts to address all your initial comments. We believe the manuscript has improved considerably. Please let us know if there are any remaining questions. We look forward to your comments in regard to our response and updated manuscript. Regardless, thank you for your work so far.

---

> > ### Comment · Reviewer_5WMe · 2022-12-05
> > **Thanks for the response**
> >
> > Thank the author for the response, the updated manuscript addresses most of my concerns, thus I updated my score from 3 to 5.
> >
> > I think the paper still lacks some coherence: the study of both critical points and convergence should be clearly motivated in the introduction and preliminary part, and a connection, if any, between the two should be drawn. It will be a good submission in another venue once the presentation is improved.

---

### Official Review · Reviewer_LN7P · 2022-10-17

**Confidence:** 3
**Correctness:** 3
**Technical Novelty And Significance:** 2
**Empirical Novelty And Significance:** Not applicable
**Recommendation:** 3

**Clarity, Quality, Novelty And Reproducibility:**

As said before, clarity and quality are low.

That being said, I truly believe that, once clarified, the paper can be strong in terms of results. I strongly encourage the authors to take a serious second look at the clarity of exposition and the correctness of the statements.

**Strength And Weaknesses:**

### **Strength**

The paper tries to derive results on what can be rebaptized the Bures-Wasserstein loss that may appear for generative purposes.

-----------------------------------------------


### **Weaknesses**

It is hard to really understand the purpose of the paper:
- Is it to derive properties of the Bures-Wasserstein distance function restricted to manifolds of constant rank as a theoretical object? In this perspective, not only ICLR may not be a target for such a paper, but also, the results do not seem very conclusive with respect to this precise literature.
- Is this to theoretically ground generative models? Here again, the overall motivation is not clearly depicted and furthermore, even if this motivation appears in the introduction and the title, there is no comment and/or simulation of this. Which, once again, is clearly out of target when considering this is a ICLR submission.
- Finally, the mathematical writing as well as the story behind the paper is really hard to follow. Notations are very messy even-though all the reparametrisations used necessitate a careful writing. As a striking example, the beginning of the Section 3  concerning the notations is really hard to follow: $\pi(W)$ is introduced but is in fact $\Sigma$, $\mu$ appears for no reel reason sometimes, and there is no consistency from one line to the other. I advise the authors to ***carefully and explicitly*** write any parameter dependence on the notations used, otherwise it is impossible to follow. Examples: $\Sigma \rightarrow \Sigma(W)$,  $W^* \rightarrow W^*_{J_k} $, etc...
- The theorems, the assumptions and the Lemma are not commented. The setup of the Theorems are not recalled. For example, Theorem 5.6 is on the gradient flow in terms of $W$ or $\Sigma$? Furthermore Theorem 5.6 cannot be really considered as a convergence result as $\|\Delta_t^*\|$ appears in the right term. Finally why do we have a Theorem 5.7 as a linear convergence result whereas Therem 5.6 is polynomial ?

**Summary Of The Paper:**

The paper studies the minimization of the Bures-Wassertein loss on positive symmetric matrices. Seen as a function over matrices of constant ranks, they give a characterisation of its critical points and of its gradient flow when added an overparametrized multiplicative structure.

**Summary Of The Review:**

Under this state, the paper is inappropriate for publication.

---

> ### Author Response · Authors · 2022-11-18
> **Response to Reviewer LN7P**
>
> We thank the reviewer for their valuable feedback. Below are our answers
> to their concerns.
>
> 1.  In response to
>
>     > Is it to derive properties of the Bures-Wasserstein distance \...?
>     > Is this to theoretically ground generative models?
>
>    * The purpose of our work is hybrid. Given the Bures-Wasserstein loss,
>     we aim at studying the optimization landscape and the convergence
>     parameter optimization procedures to minimizers.
>
>    * The motivation behind this particular loss stems from generative
>     modelling defined with Wasserstein-type losses. In order to gain
>     insight into this topic, we leverage a closed-form solution to the
>     distance computation which is available when comparing two Gaussian
>     distributions. We agree that the motivation was not made
>     sufficiently clear in the initial version of our paper and we have
>     now made several modifications to improve this in particular added
>     comments in the introduction and reorganized the Sections 2 and 3.
>     Please see also our response to pMGB first item and response to 8S82
>     motivation.
>
>    * While we don't claim to study a real world GAN, we believe that in
>     order to understand the problem in full generality it is instructive
>     to first understand the simplified instance that we consider in our
>     paper. It is striking indeed that even this case is as challenging
>     and that theoretical studies or proofs of convergence for
>     optimization procedures have so far been lacking.
>
>    * We are aware that a typical Wasserstein GAN does not exactly compute
>     a Wasserstein distance due to the representation of the
>     discriminator in terms of a neural network as well as the inexact
>     computation of the discriminator in practice even over the set of
>     realizable choices. Moreover, in practice one often works with mini
>     batches both for the real data generating distribution and the model
>     distribution. These are several different aspects that one would
>     need to take into consideration for fully understanding GANs. In our
>     work we seek to advance on understanding just some of the components
>     of the problem, namely the non-linear parametrization and the loss
>     for the generator.
>
> 2.  In response to
>
>     > Mathematical writing\...
>
>     We agree that the writing and notation was not optimal. We thank the
>     reviewer for their specific suggestions. We have worked on making
>     the dependency on the different variables clearer. The notation for
>     the parametrizations aims at separating the variables from the
>     actual map that performs the transformation. Unfortunately we could
>     not directly implement your suggested notation since this would
>     clash with other notations we are using. Nonetheless, we will make
>     sure to carefully work through the document to improve the notation
>     wherever possible and make all the dependencies clear.
>
> 3.  In response to
>
>     > The theorems, the assumptions and the Lemma are not commented\...
>
>     We thank the reviewer for pointing this out. We have updated the
>     different theorems with a more explicit statement of their
>     assumptions and settings, notably by adding a clear rule of update
>     for the gradient descent in equation (13). The Theorem 5.6 has be
>     rewritten in order to remove the dependency on the factor
>     $\Delta_t^*$, and in order to match the rate of convergence given in
>     Theorem 5.7. Please see also our response to pMGB.
>
> We worked on addressing all your concerns. We believe that the
> manuscript has improved significantly as a result of this. We hope you
> will find the revised manuscript more convincing. Thank you again for
> your constructive feedback.

---

> > ### Comment · Reviewer_LN7P · 2022-11-22
> > **Final thought**
> >
> > I thank the authors for having taken into account the remarks that all reviewers asked. However, I feel that the first submission was below the bar of admission. I encourage the authors to resubmit to another venue: their paper will be strong when clearer!

---

> > > ### Author Response · Authors · 2022-11-23
> > > **Re: Final thought**
> > >
> > > Thank you for your response. We made systematic efforts to address your comments in the revised manuscript and believe this has much improved. We understand that ACs and reviewers reserve the right to ignore changes that are significantly different from the original paper. However, while the changes to the manuscript provide clarifications and improve definitions and results, they do not change the nature of the main results or the key components of the work. Hence we would appreciate it very much indeed if you could still take a look and share your thoughts about the revision. Regardless, thank you for your feedback.

---

### Official Review · Reviewer_pMGB · 2022-10-20

**Confidence:** 4
**Correctness:** 3
**Technical Novelty And Significance:** 2
**Empirical Novelty And Significance:** Not applicable
**Recommendation:** 5

**Clarity, Quality, Novelty And Reproducibility:**

In my opinion, if the authors can address my concerns above, then the novelty of the math part is significant enough to meet the bar for acceptance. What is lacking, however, is a sound motivation for the considered setting.

**Strength And Weaknesses:**

Strength:
- Many properties of the Bures-Wasserstein loss in the context of deep linear networks are proved, which might be interesting for future works.


Weaknesses:
- The motivation for using the Bures-Wasserstein loss is not clear. In other words, while I find the math interesting, I do not understand why we should study the problem presented in the paper. Can the authors provide further arguments other than "the Bures-Wasserstein distance has also attracted much interest"? Please note that each of the cited literature in the introduction, unlike this work, has its own motivation for the Bures-Wasserstein distance.

- The characterization of the critical points of (non-smooth) Bures-Wasserstein losses is quite disconnected from the rest of the paper.

- Why does the gradient descent (Theorem 5.7) converge faster than the gradient flow, ie exponential versus $O(t^{-1})$? Moreover, the proof of Theorem 5.7 relies on a contraction argument. Why does this not apply to GF, as an "infinitesimal step-size GD"?


- The bound in Lemma 3.2 does not decrease to 0. In fact, quite unintuitively, it grows with $r$, the rank constraint. These facts suggest that the relation between the smooth and non-smooth Bures-Wasserstein losses is not captured correctly.

- The uniform margin condition in Definition 5.2 is exceedingly strong as it is required to hold for *all* unitary matrix $U$. This is different from Arora et al. 2019a where the "over all $U$" part is not present. For instance, let $\Sigma_\theta = \Sigma_0 = I_n$. Then the $\max_U$ over the left-hand side of (7) would grow with $n$, which is not a property of the deficiency margin condition in (Arora et al. 2019a).
In this light, Remark 5.4 seems quite misleading as it suggests that the uniform deficiency margin condition can be easily met with high probability by choosing a specific initialization as in Arora et al. 2019a.


- The authors claimed to have proved $O(t^{-1})$ convergence for GF. However, inspecting Theorem 5.6, there is an $\Delta_t^*$ term in (10) which might increase over $t$ (at least the authors did not provide a rigorous proof that $\Delta_t^*$ remains bounded). The proof therefore seems incomplete to me.

**Summary Of The Paper:**

This paper studies the convergence of deep linear networks under the so-called Bures-Wasserstein loss, a loss function on the space of positive semi-definite matrices. Such losses arise as the 2-Wasserstein distance between gaussian distributions, and hence can be considered as approximating a gaussian target in a generative setting.

The authors characterized the critical points of the loss. Since the Bures-Wasserstein loss is non-smooth, the authors introduced a smooth version of the loss and prove the convergence rates of the gradient flow under an initial condition and a uniform margin condition. Extention to gradient descent with small step-sizes was considered.

**Summary Of The Review:**

This paper studies a mathematically interesting problem and provides a detailed analysis. My two major concerns are:

1. The various gaps in the derivation, and
2. Motivation.

I suggest the authors to improve the paper based in these two directions.

---

> ### Author Response · Authors · 2022-11-18
> **Response to Reviewer pMGB (Part 1/2)**
>
> We thank the anonymous Reviewer for the careful reading, constructive
> and helpful comments. We have systematically worked on addressing all
> issues with the derivations and implementing the suggestions. This led
> to a substantially updated and much improved revised manuscript. Below,
> please find a point-by-point reply to the reported comments.
>
> 1.  In response to
>
>     > The motivation for using the Bures-Wasserstein loss\...
>
>     Thanks for bringing this up. A similar comment appeard in the review
>     of 8S82.
>
>   *  A main motivation to study the Bures-Wasserstein loss comes from the
>     study of generative adversarial neural networks (GANs), which are
>     successful generative models that have attracted enormous interest
>     but for which the convergence properties are not yet sufficiently
>     well understood. The connection between Wasserstein GANs and the
>     Bures-Wasserstein loss is explained in Section 3.2. The BW loss over
>     deep linear networks is a special setting that we study with the aim
>     to develop the theoretical understanding of these models. Instead of
>     studying the minimax problem directly, in our case the inner
>     maximization problem has a closed-form optimal solution. Indeed, in
>     the case of centered Gaussian distributions, the problem boils down
>     to the study of Bures-Wasserstein distance. We posit that if we want
>     to have any hope at understanding the optimization problem in
>     Wasserstein GANs, we should understand the Bures-Wasserstein loss
>     over deep linear networks, which is the topic that we pursue in our
>     work. To better convey this motivation we have now added a few
>     sentences in the introduction.
>
>
>    * Mathematically speaking, the Bures-Wasserstein distance is a very
>     interesting quantity to investigate especially in the case of a deep
>     linear networks, since it presents new challenges and allows us to
>     move beyond cases that are already well understood such as the
>     square loss. In particular, the Bures-Wasserstein distance is not
>     convex on the space of linear functions parametrized by a deep
>     linear network. This leads to a rather challenging problem to study
>     in terms of the optimization landscape i.e., the study of critical
>     points and the convergence of gradient flow/ descent. In the
>     literature of deep linear networks, the L2-loss is overly exploited.
>     The Bures-Wasserstein loss offers a novel direction to the
>     optimization of deep linear networks. We think that the strategies
>     and methods developed by studying this problem are relevant for more
>     general classes of losses as well, as mentioned in the previous
>     item.
>
>
>    * Finally, we should point out that in the literature of optimal
>     transport, the Bures-Wasserstein distance is a very important
>     quantity and the Bures-Wasserstein gradient flow is used for low
>     rank approximation of Gaussian processes, e.g.
>     [LCBR].
>
>
>   *  From our perspective a main interest in studying deep linear
>     networks is that they serve as a platform to investigate
>     optimization of non-linearly parametrized models. Indeed deep linear
>     networks can also offer interesting possibilities in terms of
>     implicit biases, a topic that is most relevant in the case that
>     there are multiple minima in function space, such as in the case of
>     classification or linear regression in a high-dimensional feature
>     space.
>
>
>    * In our analysis of deep neural networks trained with the
>     Bures-Wasserstein loss, we focused on the maximal-rank case with
>     initialization bounded away from zero. Our Theorems 5.6 and 5.7 show
>     that the depth of the network accelerates the convergence of the
>     gradient algorithms. This could be a motivation to employ this type
>     of parametrization. From the perspective of implicit biases, a
>     closed-form expression for the evolution of singular values of the
>     end-to-end matrix is unfortunately missing, which prevents us from
>     concluding on an effective low-rank implicit bias type of
>     convergence for the network. This is a very interesting avenue to
>     consider in future work.
>
>
> Reference
>
> [LCBR] M. Lambert, S. Chewi, F. Bach, S. Bonnabel, and P. Rigollet. Variational inference via Wasserstein gradient flows. NeurIPS 2022

---

> > ### Author Response · Authors · 2022-11-18
> > **Response to the Reviewer pMGB (Part 2/2)**
> >
> > 2.  In response to
> >  > The characterization of the critical points ...
> >
> >     This is a good point. In the initial manuscript we did not
> >     sufficiently emphasize the connection between the discussion of
> >     critical points and the convergence analysis. We have improved the
> >     presentation in the revision to make the connection clearer and have
> >     included new results to this end.
> >
> >    * The characterization of the critical points for the smooth
> >     Bures-Wasserstein loss is necessary in order to study the possible
> >     solutions provided by gradient optimization algorithms.
> >     Specifically, Theorem 4.4 describes the nature of the critical
> >     points in the function space. The reviewer is right, a connection to
> >     the parameter space was missing. We have now added a discussion
> >     about how a critical point in the parameter space corresponds to a
> >     critical point in the function space. This result is based on the
> >     work of [TKB], which we now discuss in
> >     Proposition 4.5. This allows us to draw conclusions about
> >     the nature of the functions that we obtain as the result of
> >     optimizing the parameters by a gradient descent algorithm, which are
> >     stationary points of the optimization problem in parameter space. We
> >     have added a discussion on this fact after Proposition 4.5.
> >
> >    * Concerning the non-smooth loss, we point out that this is the
> >     original loss, and the smooth loss is derived as an approximation of
> >     this one. Hence it is natural to ask what are the critical points of
> >     both objectives and how similar they are. Our analysis provides such
> >     comparison and shows that the distance of the critical points in
> >     function space is bounded in terms of $\tau$ as explained in the
> >     updated and improved Lemma 3.3.
> >
> >    * We point out that we are indeed investigating the convergence of
> >     gradient descent for the original non-smooth loss in Theorem 5.7.
> >     Here, similar to the above discussion it is of interest to
> >     characterize the possible solutions in function space. We can
> >     observe that the critical points in function space are critical
> >     points in parameter space and hence we have a partial description of
> >     the possible solutions in function space. We recognize however that
> >     we are still missing a complete description of the correspondence
> >     between critical points in parameter space and critical points in
> >     function space.
> >
> >    * In addition, if one uses the original loss separately over each
> >     $\mathcal{M}_k$ (over each of these sets it is differentiable), the
> >     convergence in function space would be to one of the minimizers
> >     computed in Theorem 4.2. This is matched experimentally, but the
> >     theory establishing these links in full detail will necessitate
> >     additional work.
> >
> >    * We believe that the revised manuscript has much improved in response
> >     to your initial comments. We hope that you will agree that the
> >     revision gives a clearer connection the two main contributions on
> >     critical points and convergence analysis.
> > 3.  In response to
> >
> >     > Why does the gradient descent converge faster...?
> >
> >     Thank you for bringing this up! Your observation is correct. The
> >     bound previously stated in Theorem 5.6 for the gradient flow could
> >     be improved to show an exponential rate of convergence. This is due
> >     to the lower-bound of the Hessian of the loss (in the covariance
> >     space) that can be leveraged for the convergence of the flow in the
> >     parameter space. We have updated the Theorem 5.6 and its proof
> >     accordingly.
> >
> > 4.  In response to
> >
> >     > The bound in Lemma 3.2 ...
> >
> >     We thank the reviewer for this observation. Indeed the bound that we
> >     had previously was not tight. We worked out an improvement of the
> >     bound now presented in the updated Lemma 3.2, which now reflects the
> >     equivalence of the two losses as $\tau$ tends to 0.
> >
> > 5.  In response to
> >
> >     > The uniform margin condition ...
> >
> >     You are right, the notion of margin deficiency we had proposed was
> >     too strong. Our analysis is possible under a weaker condition that
> >     we have updated in Definition 5.2. In this new definition, only the
> >     minimum over all possible unitary matrices $U$ is relevant (which is
> >     nothing else than the Bures-Wasserstein distance between $\Sigma$
> >     and $\Sigma_0$), and thus does not suffer from the pathology that
> >     you had pointed out for the previous condition. While the new
> >     condition is still restrictive, it can be fulfilled by any $\Sigma$
> >     that is close enough to $\Sigma_0$.
> >
> > Reference
> >
> > [TKB] M. Trager, K. Kohn, and J. Bruna. Pure and spurious critical points: a geometric study of linear networks. ICML, 2020.

---

> > > ### Author Response · Authors · 2022-11-18
> > > **Response to Reviewer pMGB (Part 3/3)**
> > >
> > > 6.  In response to
> > >
> > >     > The authors claimed to have proved $O(t^{-1})$ convergence ...
> > >
> > >     The term in $\Delta_t^*$ was indeed not well taken care of. We have
> > >     improved the proof and the statement of the Theorem 5.6 which now do
> > >     not need this term.
> > >
> > > We have made systematic efforts to address all the concerns from the initial
> > > review. We believe that the updated manuscript has significantly
> > > improved as a consequence of this. We hope to have answered all of your
> > > concerns and that you will agree that the paper is much improved. Thank
> > > you for your comments and help in doing so.

---

> ### Author Response · Authors · 2022-11-24
> **Comments in regard to our response and updated manuscript**
>
> We made systematic efforts to address all your initial comments. We believe the manuscript has improved considerably. Please let us know if there are any remaining questions. We look forward to your comments in regard to our response and updated manuscript. Regardless, thank you for your work so far.

---

> > ### Comment · Reviewer_pMGB · 2022-12-05
> > **Thanks for the detailed rebuttal**
> >
> > I want to thank the authors for the revision which addresses many of my initial concerns. I have thus updated my score from 3 to 5.
> >
> > My remaining concerns are the motivation and the margin condition, which, although weakened in the revision, is still quite restrictive. I encourage the authors to improve upon these issues and resubmit in the future.

---

### Official Review · Reviewer_8S82 · 2022-10-21

**Confidence:** 3
**Correctness:** 4
**Technical Novelty And Significance:** 2
**Empirical Novelty And Significance:** Not applicable
**Recommendation:** 5

**Clarity, Quality, Novelty And Reproducibility:**

The paper is well written and clear.

The novelty of this paper only lies in adapting the proof of convergence of (Arora et al. 2020) to the perturbative Bures-Wasserstein loss. The result of this paper requires the same two very strong assumptions required for the proof of (Arora et al. 2020), suggesting that there is no significant improvement to the proof techniques.

I do not understand why the paper focuses on such a specific loss as the Bures-Wasserstein loss. Furthermore it is unclear why there would be an advantage in optimizing the Bures-Wasserstein loss with a deep linear network instead of with a matrix directly, since both seem to converge to the same global minimum. My impression is that deep linear networks are mainly useful in settings such as matrix completion where their implicit bias towards low-rank solution help generalization. Is there such an advantage in this setting?

**Strength And Weaknesses:**

The proof of convergence is strong, but I have issues with its novelty (see next section) and with the assumptions:
 - Though the balanced assumption is common due to its power, I find it quite restrictive. The main reason why I am interested in the convergence of deep linear networks is as a stepping stone to later prove convergence of nonlinear networks. And these kind of balanced arguments break down in the non-linear case. I think it is important to focus on non-balanced linear networks.
 - The uniform deficiency margin assumption is very strong and the discussion of this assumption is misleading: (Arora et al., 202) show that a random network at initialization will have a positive deficiency margin with prob. close to 0.5 only when when the input and outputs have dimension 1. In Remark 5.4 they cite the above paper but omit the dimension assumption which is very strong (balanced dimension 1 network are equivalent to scalar networks whose convergence with prob. 0.5 is obvious). In Remark 5.4 the authors write "in our setting, the uniform deficiency margin can also be satisfied using a similar result" but I couldn't find the corresponding argument anywhere in the appendix.


**Summary Of The Paper:**

The paper studies the convergence of a deep linear network trained with a Bures-Wasserstein loss.

The authors first give a description of the critical point of the Bures-Wasserstein loss (and its perturbation) when restricted to the set of rank $k$ matrices $\mathcal{M}_k$. If I understand correctly, the critical points of the loss as a function of the parameters of the network are not studied.

Convergence of gradient flow and gradient descent on the parameters of the network (with the perturbative Bures-Wasserstein loss) are then showed for initialization that are balanced and have a uniform deficiency margin.

**Summary Of The Review:**

The paper adapts the proof of convergence of (Arora et al. 2020) to the Bures-Wasserstein loss. The same strong assumptions are required for their convergence proof.

---

> ### Author Response · Authors · 2022-11-18
> **Response the Reviewer 8S82 (Part 1/2)**
>
> We are grateful to the Reviewer for the careful reading, constructive
> and helpful comments. The suggestions have been incorporated into a
> substantially revised version. Let us first start with some general
> comments regarding the Reviewer's concerns about novelty and the
> motivation behind using the Bures-Wasserstein distance.
>
> **Novelty** In response to the comment
> > The novelty of this paper only lies in adapting the proof of
> > convergence of Arora et al. (2019) to the perturbative
> > Bures-Wasserstein loss
>
> * We point out that a main contribution of our work is to characterize the
> critical points of the Bures-Wasserstein loss on the space of functions
> parametrized by a deep linear network. This important aspect was omitted
> in the Reviewer's assessment of novelty.
>
> * We have now also added a new result (Proposition 4.5) connecting the
> critical points in function space and the stationary points in parameter
> space found by gradient flow (in the maximal-rank case). The convergence
> analysis is a second main contribution from our work that complements
> the geometrical analysis of the optimization landscape. To further
> develop the connection to parameter space, we have also added the
> observation that for the smooth Bures-Wasserstein loss, the critical
> points in parameter space correspond to critical points in function
> space (over the space of realizable linear functions) and in covariance
> space with the corresponding rank constraints (see discussion in Section
> 4.2).
>
> * We would like to point out that we have two types of convergence
> results: results for gradient descent and results for gradient flow.
> While the proofs of convergence for gradient descent are indeed based on
> previous works, those works focused on the MSE loss. Importantly, for
> the BW loss that we consider, the parametrization of the covariance
> matrix makes the loss non-convex on the space of linear functions
> parametrized by the neural network, so that in our case the analysis is
> more involved.
>
> * Moreover, our proof strategy for gradient flow convergence requires a
> bound on the Hessian of the loss over the space of covariance matrices
> in order to conclude on its strong convexity. This analysis is new and
> it required significant efforts (see Appendix D). Kindly note that the
> work of Arora et al. (2019) does not study gradient flow. In fact one of
> the reasons we consider the BW loss an interesting case are the
> challenges that are involved in studying the convergence of gradient
> flow for this loss.
>
> * To adequately advertise the analysis we are conducting we have improved
> the presentation and included a new relevant result Lemma 5.5. We are
> not aware of any other work studying exponential rates of convergence in
> a comparable setting as ours. We think that the techniques we have
> developed can be relevant for any loss that is convex in some space in
> the sense that it can be factored as $\ell\circ \pi \circ \mu(\theta)$
> with a convex $\ell$ even if it is not convex in function space.
>
> **Motivations** In response to the comment
> > I do not understand why the paper focuses on such a specific loss as
> > the Bures-Wasserstein loss...
>
>
> A main motivation to study the Bures-Wasserstein loss comes from the
> study of generative adversarial neural networks (GANs), which are
> successful generative models that have attracted enormous interest but
> for which the convergence properties are not yet sufficiently well
> understood. The connection between Wasserstein GANs and the
> Bures-Wasserstein loss is explained in Section 3.2. The BW loss over
> deep linear networks is a special setting that we study with the aim to
> develop the theoretical understanding of these models. Instead of
> studying the minimax problem directly, in our case the inner
> maximization problem has a closed-form optimal solution. Indeed, in the
> case of centered Gaussian distributions, the problem boils down to the
> study of Bures-Wasserstein distance. We posit that if we want to have
> any hope at understanding the optimization problem in Wasserstein GANs,
> we should understand the Bures-Wasserstein loss over deep linear
> networks, which is the topic that we pursue in our work. To better
> convey this motivation we have now added a few sentences in the
> introduction.

---

> > ### Author Response · Authors · 2022-11-18
> > **Response to Reviewer 8S82 (Part 2/2)**
> >
> > Mathematically speaking, the Bures-Wasserstein distance is a very
> > interesting quantity to investigate especially in the case of a deep
> > linear networks, since it presents new challenges and allows us to move
> > beyond cases that are already well understood such as the square loss.
> > In particular, the Bures-Wasserstein distance is not convex on the space
> > of linear functions parametrized by a deep linear network. This leads to
> > a rather challenging problem to study in terms of the optimization
> > landscape i.e., the study of critical points and the convergence of
> > gradient flow/ descent. In the literature of deep linear networks, the
> > L2-loss is overly exploited. The Bures-Wasserstein loss offers a novel
> > direction to the optimization of deep linear networks. We think that the
> > strategies and methods developed by studying this problem are relevant
> > for more general classes of losses as well, as mentioned in the previous
> > item.
> >
> > Finally, we should point out that in the literature of optimal
> > transport, the Bures-Wasserstein distance is a very important quantity
> > and the Bures-Wasserstein gradient flow is used for low rank
> > approximation of Gaussian processes, e.g. in [LCBR].
> >
> > For your comment
> >
> > > My impression is that deep linear networks...
> >
> > From our perspective a main interest in studying deep linear networks is
> > that they serve as a platform to investigate optimization of
> > non-linearly parametrized models. Indeed deep linear networks can also
> > offer interesting possibilities in terms of implicit biases, a topic
> > that is most relevant in the case that there are multiple minima in
> > function space, such as in the case of classification or linear
> > regression in a high-dimensional feature space.
> >
> > In our analysis of deep neural networks trained with the
> > Bures-Wasserstein loss, we focused on the maximal-rank case with
> > initialization bounded away from zero. Our Theorems 5.6 and 5.7 show
> > that the depth of the network accelerates the convergence of the
> > gradient algorithms. This could be a motivation to employ this type of
> > parametrization. From the perspective of implicit biases, a closed-form
> > expression for the evolution of singular values of the end-to-end matrix
> > is unfortunately missing, which prevents us from concluding on an
> > effective low-rank implicit bias type of convergence for the network.
> > This is a very interesting avenue to consider in future work.
> >
> > Below, please find a point-by-point reply to the specific comments.
> >
> > 1.  In response to
> >
> >     > Though the balanced assumption\...
> >
> >     We agree that the balancedness assumption is restrictive, and might
> >     not extend to broader classes of neural networks. Nevertheless, it
> >     remains a standard assumption in the context of deep linear neural
> >     networks. Some authors (such as [MTVM] or
> >     [TFHV]) investigate
> >     unbalancedness settings but do not consider deep networks. Future
> >     work should indeed consider extending the studies of the unbalanced
> >     regime to the deep case. In our work we have focused on addressing
> >     another notable limitation of many previous works on linear networks
> >     which is the type of loss that is considered. This entails
> >     challenges beyond those that are addressed by assuming balancedness,
> >     advancing on which we believe is also an important endeavor.
> >
> > 2.  In response to
> >
> >     > The uniform deficiency margin assumption
> >
> >     In the initial submission we had stated an incorrect condition in
> >     Remark 5.4. We apologize for the confusion created. We have removed
> >     the Remark 5.4, which, we agree, was misleading. In the updated
> >     manuscript we have relaxed the previous formulation of the uniform
> >     deficiency margin. Please find now the new version in Definition
> >     5.2. Now the bound for the modified uniform deficiency margin is
> >     satisfied in case of initialization close to the target. Also, we
> >     added the remark that the uniform margin deficiency is a sufficient
> >     condition for the gradient flow convergence analysis but not a
> >     necessary condition. The necessary condition is stated in the Lemma
> >     5.3 on the boundedness of the eigenvalues of $WW^\top$, which holds
> >     for instance when the weights are balanced and initialized with
> >     full-rank.
> >
> > References
> >
> > [LCBR] M. Lambert, S. Chewi, F. Bach, S. Bonnabel, and P. Rigollet. Variational inference via Wasserstein gradient flows. NeurIPS 2022
> >
> > [MTVM] H. Min, S. Tarmoun, R. Vidal and E. Mallada. On the Explicit Role of Initialization on the Convergence and Implicit Bias of Overparametrized Linear Networks. ICML 2021
> >
> > [TFHV] S. Tarmoun, G. Franca, B. D. Haeffele, and R. Vidal. Understanding the Dynamics of Gradient Flow in Overparameterized Linear models. ICML 2021

---

> > > ### Comment · Reviewer_8S82 · 2022-11-24
> > > **Response**
> > >
> > > Thanks for the update and clarifications, I now better understand the motivation behind the Bures-Wasserstein loss.
> > >
> > > I still think that the novelty is marginal over (Arora et al. 2019) since this work does not improve on the two restrictive assumption of balancedness and initialization close to the solution. In the mean time there has been results that study the training dynamics of gradient flow for deep unbalanced linear networks for general losses (Jacot et al. 2021).
> > >
> > > I have updated my score to a `weak reject'.
> > >
> > > Arthur Jacot, François Ged, Berfin Şimşek, Clément Hongler, Franck Gabriel, Saddle-to-Saddle Dynamics in Deep Linear Networks: Small Initialization Training, Symmetry, and Sparsity, 2021

---

> > > > ### Author Response · Authors · 2022-11-30
> > > > **Response to the Reviewer 8S82**
> > > >
> > > > Thank you for sharing the reference to the work of Jacot et al, we were aware of this reference. The idea of this paper is to move from saddle to saddle while the loss is decreasing and ideally goes to zero and we learn the target. However, this is only a conjecture as it is clearly mentioned in the abstract of the paper (see conjecture 3 in the paper). Whereas we use as an assumption balanced weights at initialization, in that paper they assume an initialization close to zero, and it is assumed to be a saddle point. Then the authors use a proper scaling in which the trajectories will enter an escape cone. In that cone the trajectories follow the largest eigenvalue. The time horizon on the escape cone is bounded by radius $r$, and the trajectories should not hit the boundary of this radius. The reason is what is happening after the trajectories escape this cone is not clear in the paper. In addition in the statement of their main results (Proposition 20 and 21) is mentioned that it works for any t but they prove for $t >t_0$ and bounded. Moreover their main result is only for the first path starting from the initialization which is a saddle point.
> > > > On the other hand we used the balanced weights assumption at initialization, we study the form of critical points, and prove exponential convergence of the trajectories for a deep linear network using Bures-Wasserstein as a loss. In view of these facts, it is not the same regime as in the paper by Jacot et al.

---

> ### Author Response · Authors · 2022-11-24
> **Comments in regard to our response and updated manuscript**
>
> We made systematic efforts to address all your initial comments. We believe the manuscript has improved considerably. Please let us know if there are any remaining questions. We look forward to your comments in regard to our response and updated manuscript. Regardless, thank you for your work so far.

---

### Author Response · Authors · 2022-11-19
**Manuscript updated**

Dear Area Chair and Reviewers,

We have revised our manuscript according to the reviewers' comments and suggestions. In particular, in the revised version, we have

1. The title has been slightly changed to emphasize that  a main contribution of our work is to characterize the critical points of the Bures-Wasserstein loss on the space of functions parametrized by a deep linear network. We also added a new result (Proposition 4.5) connecting the critical points in function space and the stationary points in parameter space found by gradient flow (in the maximal-rank case).

2. We have changed the assumption "uniform deficiency margin" to be "modified deficiency margin" in Definition 5.2. so that it becomes more accessible. The proof of gradient descent convergence works well under this new assumption.

3. We have improved the proof and the statement of the Theorem 5.6, so that the gradient flow has exponential convergence, and it is now consistent with the convergence result of gradient descent (Theorem 5.7).

4. Lemma 3.3 has been improved. Now as $\tau$ goes to zero, the difference between perturbative loss and original loss goes down to zero as well.



We highlighted the revised parts in blue color in the main text.

---

### Decision · Program_Chairs · 2023-01-20

**Decision:**

Reject

**Justification For Why Not Higher Score:**

the reviews do not merit a higher score

**Justification For Why Not Lower Score:**

N/A

**Metareview: Summary, Strengths And Weaknesses:**

The authors consider a deep matrix factorization model of covariance matrices trained with the Bures-Wasserstein distance. Unlike most recent publication that focus on discriminative settings and the square loss, the authors' model considers another interesting type of loss that connects with the generative setting. The authors characterize the critical points and minimizers of the Bures-Wasserstein distance over the space of rank-bounded matrices. The authors establish convergence results for gradient flow using a smooth perturbative version of the loss and convergence results for finite step size gradient descent under certain assumptions on the initial weights.  The reviewers thought the convergence proof is interesting. The reviewers however found the balanced assumption too restrictive raised due to lack of generalization to nonlinear cases and also thought the uniform deficiency margin is also restrictive and somewhat misleading, lack of novelty in the proof compared to Arora 2022, and lack of clarity on focusing on Bures-Wasserstein loss. The authors provided a thorough response but their response did not fully assuage reviewer concerns. I agree with the reviewer assessments, while the paper has some interesting ideas more work is required to relax the stringent assumptions. I recommend submission to a future ML venue after a thorough revision but can not recommend acceptance of the paper in its current version.